# Memristor-based adaptive neuromorphic perception in unstructured environments

Shengbo Wang [1,9], Shuo Gao [1,9] ✉, Chenyu Tang [2], Edoardo Occhipinti [3], Cong Li[1], Shurui Wang[1], Jiaqi Wang[1], Hubin Zhao [4], Guohua Hu [5], Arokia Nathan[6,7], Ravinder Dahiya [8] & Luigi Giuseppe Occhipinti [2] ✉

Efficient operation of control systems in robotics or autonomous driving targeting real-world navigation scenarios requires perception methods that allow them to understand and adapt to unstructured environments with good accuracy, adaptation, and generality, similar to humans. To address this need, we present a memristor-based differential neuromorphic computing, perceptual signal processing, and online adaptation method providing neuromorphic style adaptation to external sensory stimuli. The adaptation ability and generality of this method are confirmed in two application scenarios: object grasping and autonomous driving. In the former, a robot hand realizes safe and stable grasping through fast (~1 ms) adaptation based on the tactile object features with a single memristor. In the latter, decision-making information of 10 unstructured environments in autonomous driving is extracted with an accuracy of 94% with a 40×25 memristor array. By mimicking human low-level perception mechanisms, the electronic neuromorphic circuit-based method achieves real-time adaptation and high-level reactions to unstructured environments.

Understanding sensory data efficiently to achieve human-like perception of the real world is pivotal for robotics[1–6]. With such capabilities, robotics could truly transit from controlled environments such as factories and laboratories into unstructured environments of home and businesses that entail considerable 'variability' (Supplementary Discussion 1 and 2). Traditionally, the adeptness of organisms within unstructured environments has been attributed to the integration of diverse types of physical information[7–9]. However, recent studies in life sciences have revealed that the most common mechanism humans use to understand unstructured environments is differential processing of sensory information[10–14]. For a given stimulus, multiple types of receptors and subsequent sensory neurons located in sensory ganglia participate in differential processing, and they adjust their structure and synaptic weight on the features of external stimuli (Supplementary Discussion 3). Specifically, by extracting the main stimulus features embedded in the signal properties, these receptors and neurons can rapidly form a complex set of intricate network-based perception functions, such as environmental mapping, motion control tasks, associative memory, etc. Memristors, a kind of neuromorphic device[15–20], not only integrate storage and computation capabilities but also have the property of changing their information transmission efficiency. These capabilities, often referred to as synapse-like characteristics[19–24], underscore the remarkable similarity between memristors and synapses, i.e., the basic computational unit in biology.

[1]School of Instrumentation and Optoelectronic Engineering, Beihang University, Beijing, China. [2]Department of Engineering, University of Cambridge, Cambridge, UK. [3]UKRI Centre for Doctoral Training in AI for Healthcare, Department of Computing, Imperial College London, London, UK. [4]HUB of Intelligent Neuro-engineering (HUBIN), CREATe, Division of Surgery and Interventional Science, UCL, HA7 4LP Stanmore, UK. [5]Department of Electronic Engineering, The Chinese University of Hong Kong, Shatin, N. T., Hong Kong S. A. R., China. [6]Darwin College, University of Cambridge, Cambridge, UK. [7]School of Information Science and Engineering, Shandong University, Qingdao 266237, China. [8]Bendable Electronics and Sustainable Technologies (BEST) Group, Department of Electrical and Computer Engineering, Northeastern University, Boston, MA 02115, USA. [9]These authors contributed equally: Shengbo Wang, Shuo Gao. ✉e-mail: shuo_gao@buaa.edu.cn; lgo23@cam.ac.uk

Such similarities make memristors the ideal fundamental device for realizing human-like perception functionalities in robotics. To this end, herein we present a memristor-based differential neuromorphic computing, perceptual signal processing and online adaptation method for robotics. Specifically, in this article we choose to term 'differential neuromorphic computing' any data manipulation method involving multi-branch functions that emulate biological sensory processing, where sensory stimuli are selectively perceived by receptors and undergo different processing (pathways) supported by diverse groups of neurons. The use of the word differential in this context is not to be confused with differential (delta) encoding of sensory data as employed say in dynamic vision sensors.

Unstructured information encompasses multidimensional and multimodal features, which undergo the perceptual processing of diverse receptors in biology[6,25,26]. This type of processing indicates the need for designing different modulation methods for neuromorphic

computing. However, existing methods focus on keeping a memristor to a fixed receptor (Fig. 1a, c and Supplementary Discussion 11), e.g., the memristor-based pressure nociceptor[27-30] only processes pressure inputs above a pre-set threshold. This kind of design omits useful information in pressure amplitudes below this threshold, failing to utilize the full spectrum of pressure data for understanding dynamic inputs in a manner akin to human skin. A potential solution is to operate multiple memristors to process pressure data across a broader spectrum. However, given the high similarity between the memristor and biological synapse, the memristor has the capability to replicate diverse synaptic plasticity, thus effectively emulating the intrinsic characteristics of different receptors and sensory neurons. Furthermore, distinct features of sensory data exhibit time domain independence, implying that a memristor can adaptively switch to an appropriate state to process each feature at a time slot. Drawing inspiration from the sensory information processing model, our

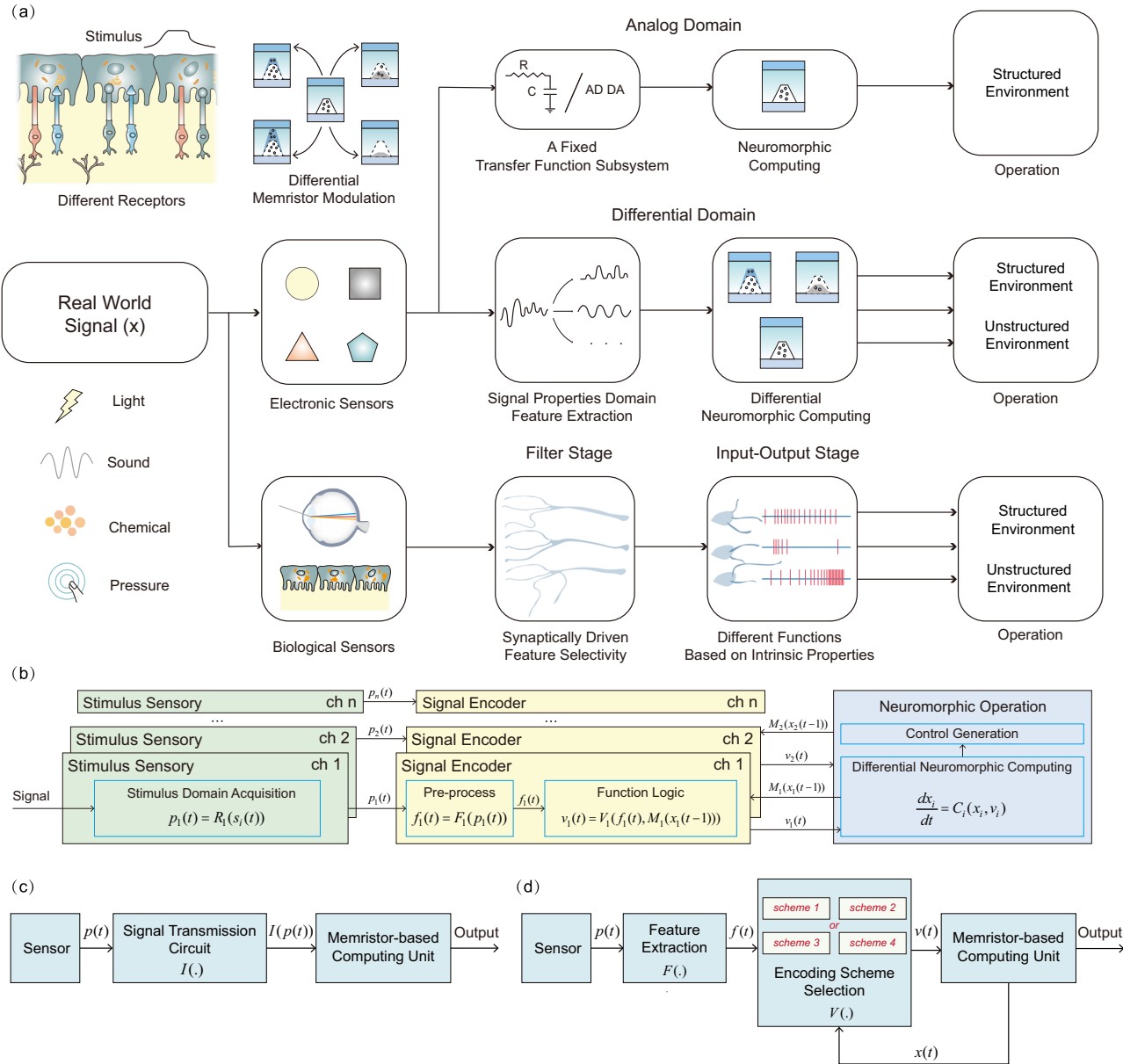

**Fig. 1 | The proposed differential neuromorphic computing method.**
**a** Comparison of our proposed differential processing method with biological sensory processing methods and current neuromorphic processing methods.
**b** The specific implementation of the memristor-based differential processing method. **c** The schematic of current methods. **d** The schematic of the differential neuromorphic computing method, distinguishing our approach from existing techniques.

proposed differential neuromorphic computing method utilizes memristors' intrinsic multistate properties. It extracts features from unstructured data and modulates the memristor state (Fig. 1d), enhancing the adaptability of robotic systems for operation in unstructured environments (Supplementary Discussion 4 to 6). This approach exploits the bifurcations of the nonlinear state-space of the memristor for providing neuromorphic style sensory adaptation to environmental stimuli. We apply this method in two complex environments. Firstly, we address robot control for object grasping, and secondly, we focus on object detection and reaction in autonomous driving. In the former, we utilized a single Self-Directed Channel (SDC) memristor, a type of chalcogenide-based electrochemical metallization (ECM) device, to emulate nociceptors by achieving amplification ( > 720%) of hazardous stimuli, and adapting receptors by achieving regulation ( < 50%) of mild stimuli, which play a key role in grasping unknown objects. In the latter scenario, we process differentially encoded visual motion information with a $40 \times 25$ memristor matrix and achieve a commendable accuracy of 94% in extracting critical information within each of 10 autonomous driving scenes, after comparison with human labeled results. These two experiments demonstrate that the proposed method is general enough to work with different types or number of memristors. Moreover, this method shows the potential to assist robotics with a comprehensive perceptual capability akin to that of living organisms, enabling them to better comprehend environmental information.

## Results

### Tactile perception

Realizing safe and stable manipulation of unknown objects by robotic hands is tricky but highly needed[31–35]. Unlike lab and factory settings, an unknown object may exhibit sharp or slippery attributes. The former potentially damages the contacting end, and the latter places a heavy burden on the sensing and computing modules when keeping the object balanced between stabilization and deformation. To address this issue, the proposed memristor-based differential neuromorphic computing method has been implemented into the sensing and control system of a robotic hand to highlight its non-linear adaptation feature as required to achieve intricate and nuanced tactile perception, illustrated in Fig. 2a. Specifically, this method is utilized to emulate multiple essential tactile receptors and sensory neurons in tactile stimuli perception, including nociceptors, fast-adapting, and slow-adapting receptors, along with their respective neural pathways.

Here, a piezoresistive force sensing architecture (more details in Fig. S1) is assembled for receiving pressure amplitudes, and a self-directed channel effect-based multilayered nonvolatile memristor (KNOWM Inc.) is selected for differential neuromorphic computing (as depicted in Fig. S2). The structure, hysteresis curves, and electrical characteristics under pulse testing of the memristor are given in Fig. 2b–e, respectively. The piezoresistive layers and the memristor offer short-term and long-term force information, respectively, based on which the attributes of an object are first extracted by a status acquisition block and then utilized to generate the corresponding modulation schemes for the memristor. The sensory feature extraction and modulation scheme selection are both conducted in a field programmable gate array (FPGA) platform, as shown in Fig. 2a. Based on the aims of modulation methods, the schemes can be further classified into 3 groups, allowing the memristor to stay at high ( > 250k), middle ( ~ 170k) and low ( < 100k) resistance levels. This arrangement facilitates the adaptive, normalization, and nociception perception of external stimuli, aligning with 3 types of biological perception behaviors, i.e., adaptation, recovery and nociception (Fig. S6, S7 and Supplementary Discussion 7). The corresponding memristor conductance change (adaptation, recovery and nociception) can be found in Fig. 2a. The nociception refers to the

amplification processing of strong pressure, the adaptation refers to the adaptation processing of mild pressure (the behavioral adaptation of the tactile system analogous to mechanoreceptors), and the recovery indicates resetting the memristor to its initial state, akin to the sensitivity reset of the biological receptor sensitivity after removing external stimuli. The detailed mapping relationship between attributes and modulation methods is given in the Supplementary Table 1.

The three mimicked biological functionalities are experimentally validated, by studying the change in force induced electrical signals with and without neuromorphic computing. Here, the neuromorphic computing module tunes the adjustment factor of a voltage amplifier block, whose input is the status of the piezoresistive film. After the magnitude of a force is measured, an associated modulation scheme is selected and then applied to the memristor. The resistance of the memristor is then changed and used as an indicator of the amplification factor. The final yielded voltage outputs illustrate the biological perception behaviors. The experimental results in Fig. 2f demonstrate the responses of this tactile system to hazardous stimuli, with a more than 170% amplification of the hazard signal through applying positive encoding pulses to the memristor. In contrast, when perceiving mild tactile stimuli (Fig. 2h), the memristor is modulated by the negative encoding pulses, achieving a more than 50% attenuation of the mild signal. The curve trends in Fig. 2f, h are similar to the biological response strengths in Fig. 2g. For biological nociceptors, the sensitivity to external sensory stimuli increases when exposed to dangerous stimuli, resulting in a continuously increasing response strength; in contrast, the adapting receptors adapt to external tactile stimuli by gradually reducing their sensitivity when the feature of stimuli is mild, resulting in a decreasing response strength.

When considering the unstructured information processing in the real world, it often becomes necessary to perform multiple processing iterations of the aforementioned functions. For better adaptation to dynamic changes in hazardous scenarios, we mimic the time window processing mechanism in organisms, thereby distinguishing between sudden threats and persistent threats[29,36]. Specifically, during the encoding process, we additionally consider the state of the memristor. When its state falls a predefined threshold (100 kΩ in this case), the switch in the modulation scheme occurs (Fig. S24). This threshold value 100 kΩ is pre-programmed into the FPGA-based logic circuit, serving as a trigger for modulation scheme adjustments. Consequently, there is an increase in both the amplitude and the pulse duty cycle of the positive pulses (as shown in Fig. 2i), achieving the functionality of sensitized processing for external stimulus information. Further details about the threshold value selection are provided in Supplementary Table 1. In our work, a > 720% amplification of tactile stimuli is offered (Fig. 2j), exceeding the state-of-the-art works[27,30,37]. Regarding the adaptation function, the adaptive speed (the magnitude of the attenuation of response strength over time) is of significance[38–40]. Here, we implement two adaptation speeds by adjusting the pulse duty cycle and amplitude of encoding pulses as shown in Fig. 2k, l. The developed method is then used in object grasping tasks, to extract and learn the main characteristics of objects for safe and stable manipulation (conceptually depicted in Fig. 3a).

In task 1, an irregular object consisting of a cube and a cone is 3D printed. During the grasping process, the robot might encounter the sharp point that leads to painful pressures and poses a potential danger. It needs the help of a nociceptor to quickly perceive the hazard stimulus caused by the sharp point and then changes the grasping strategy (Supplementary Discussion 10), as conceptually depicted in Fig. 3b, in which the resistance information of the piezoresistive film and the memristor during the entire grasping task is shown. At 2.2 s, the piezoresistive sensor attached to the robotic hand experiences a significant force due to contact with the sharp point, resulting in decreased resistance. In this case, the pressure conforms to hazardous

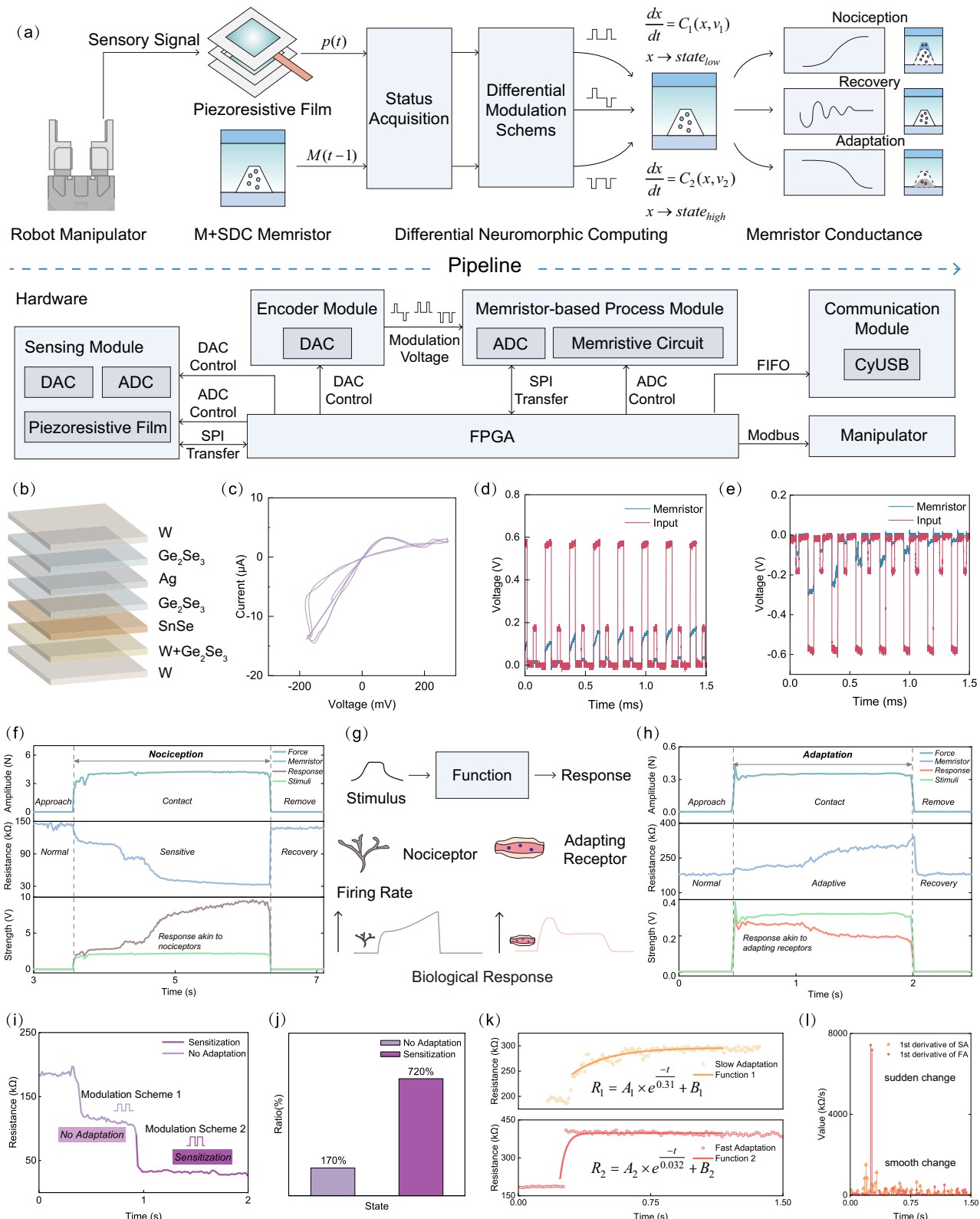

characteristics, and the memristor is modulated into the amplification state (low resistance). At 8.2 s, the memristor resistance falls below the threshold of 35 kΩ, and the pressure stimulus is amplified by 500%. This indicates that the robot has been in contact with the sharp point, triggering the pain reflex. Based on this pain experience, the robot changes its grasping position and hand gesture. Specifically, the grasping position of the robot is adjusted to an appropriate position to expect the subsequent safe contact between 8.2 and 14.4 s.

Subsequently, the robot hand moves again at 14.4 s and makes contact with the object at 16.1 s. By 17.4 s, the pressure perceived by the contact point stabilizes, and the memristor is modulated into the high-resistance state, achieving safe and stable grasping.

Similarly, in task 2, a soap is used to represent slippery objects. The instability of tactile information during the slipping process leads to spikes in piezoresistive film incorporated with the memristance resistance change, prompting the robot to increase the gripping force

**Fig. 2 | Utilizing differentiation processing method to process tactile information and mimic multiple receptors. a** Schematic diagram of the comprehensive pipeline designed to process tactile sensory stimuli. This approach initially captures immediate tactile interactions through the resistance changes in a piezoresistive film, employing Analog-to-Digital Converters (ADC) and Digital-to-Analog Converters (DAC) and the required information exchange interface. Concurrently, it archives historical sensory data from the memristor state, leveraging the memory properties of a memristor. Subsequently, sensory features are extracted from both the current state of the piezoresistive film and the memristor to select a memristive adaptive modulation scheme, driving the nociception, recovery, or adaptation process, based on the memristor conductance change. The entire sensory data acquisition, feature extraction, and modulation scheme selection process are managed through the FPGA control platform. Note that the memristor is detected and modulated under the cooperation of the encoder and memristor-based process modules within the hardware setup. Additionally, a communication module is established for transferring information to external PC devices alongside a specifically designed interface for manipulator control. **b** The device structure of the self-directed memristor. **c** The U-I test of memristors. In this test, a 500 mV peak-to-peak sine wave is applied to the memristor and the 10kΩ fixed-value resistor. **d** Electrical characterization of memristors using positive pulses. **e** Electrical characterization of memristors using negative pulses. **f** Amplification processing of hazardous stimuli. **g** Corresponding biological processing functions. **h** Adaptation processing of mild stimuli. **i** Sensitization processing achieved through further differentiation. **j** Quantitative evaluation of the sensitization function. **k** Realization of adaptation at different speeds. **l** The adaptive rate.

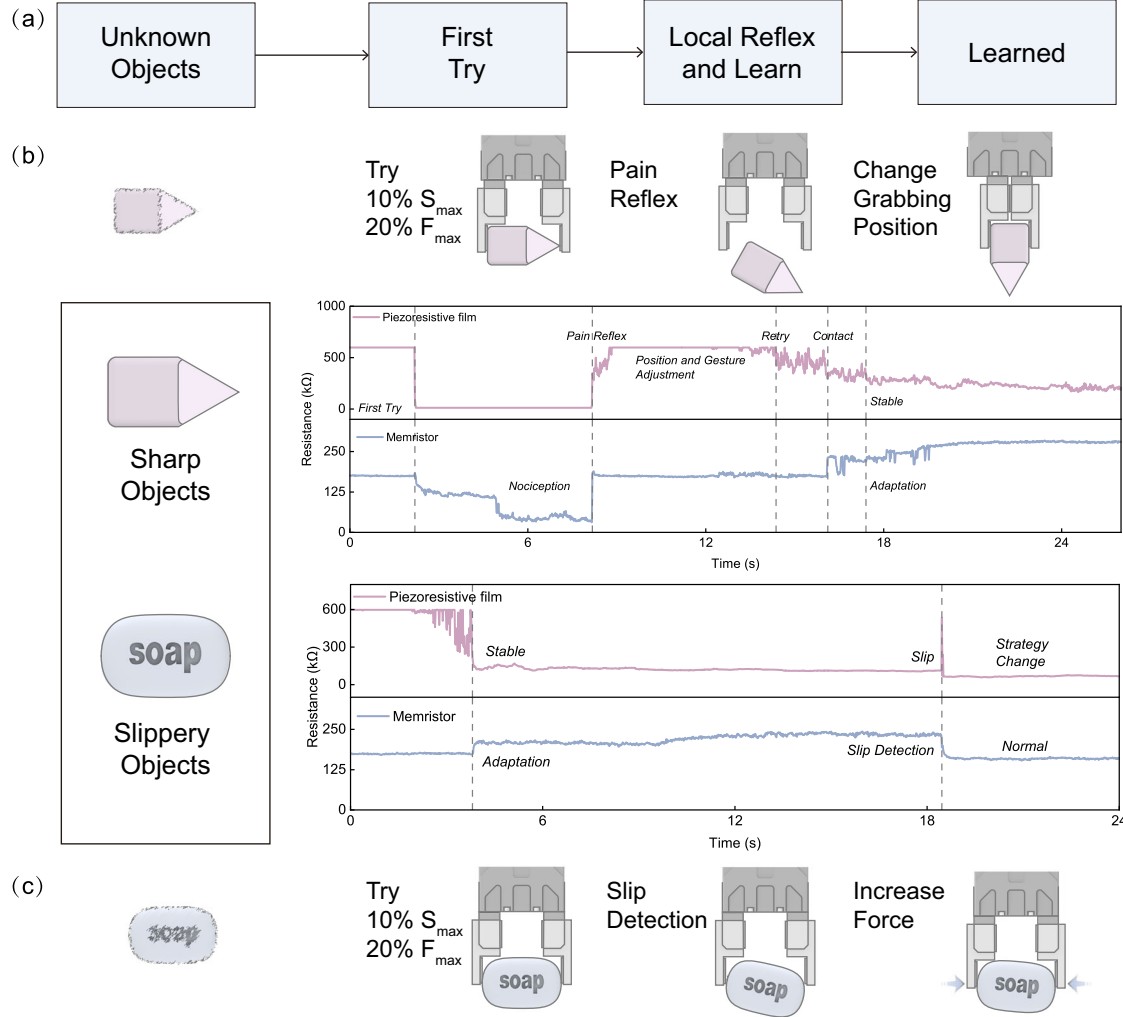

**Fig. 3 | Grasping diverse objects in unstructured environments. a** Overall logical flowchart of learning environmental information. **b** Learning the properties of sharp objects and ultimately achieving safe grasping. **c** Learning the properties of slippery objects and ultimately achieving stable grasping.

to ensure stability. As illustrated in Fig. 3c, during the grasp attempt of the soap, the gripping force stabilizes at 3.8 s, with the memristor switching to a high-resistance state. By 18 s, the adaptation level to external tactile information reaches 75%. At 18.5 s, an external inference causes the object to slip, leading to changes in gripping force and resulting in a spike in the memristor resistance. Upon detecting a slip event onset, the robot prompts an increase of the gripping force to prevent the object from slipping further. This timely increase in contact force prevents the object from falling, and the memristor is modulated into a normal perception state (stable middle resistance state), successfully achieving stable grasping of a slippery object. To our knowledge, this represents the first time that a memristor-based approach has been successfully employed for local slip detection and consequent adjustment of the actuator force in robotic grasping tasks.

At present, the processing and execution time of the proposed method is 1 ms, which can strongly support safe and stable operations for robots[41–44]. The operation time can be further improved when better memristive devices that respond in nanoseconds (ns) or microseconds (µs) are employed[45–48].

## Visual perception

In previous tactile information processing, attributes based on force strength were used to understand environmental knowledge. In contrast, visual frequency-based attributes are more important for autonomous driving, as they imply the relative position change of surrounding objects to the car[49–51] and are vital for real-time decision making[52–54]. For instance, the sudden appearance of pedestrians or vehicles can lead to life-threatening collisions. Thus, shortening the attribute extraction time of moving objects is important. In addition, the moving direction of the object can further help to make good decisions[55,56]. To this end, the proposed differential neuromorphic method is employed to acquire and process visual frequency-based attributes (Supplementary Discussion 13). Drawing an analog to the frequency processing difference between cone and rod cells in biological vision perception-where cone cells excel in capturing rapidly changing visual stimuli-this method retrieves fast information akin to cone cells, and produces neural excitation to maintain fast information for a while (Supplementary Discussion 8).

In the implementation, a driving recorder of $1920 \times 900$ resolution and a $25 \times 40$ memristor array are used. During the processing procedure shown in Fig. 4a, the gray CMOS image from the recorder is compressed by averaging a matrix of m×n into one pixel (in this set parameters, m and n equals 48 and 36, respectively) based on image spatial redundancy theory[57]. Next, filtering circuits for pre-process extracts the visual information changes within two adjacent frames. Then, the computing function select module, based on the analog computing circuit and the control switch, divides the light intensity changes into fast and slow categories based on a comparison to a preset threshold, resulting in the different encoding functions $E_i$. Specifically, if the change exceeds this threshold, the visual stimuli in this pixel are classified as fast; otherwise, it is considered slow. Subsequently, the fast and slow visual attributes are encoded into electrical pulses $V_f$ and $V_s$ to modulate the memristor into low- and high-resistance states, respectively. Here, slow visual information is taken to release the memristor from a low-resistance state to a high-resistance state. As the change in resistance is analog, its value within this period reflects the moving orientation of the object, achieving a neural excitation effect. In scenarios involving moving objects, the light intensity changes correlate to the relative moving speed of an object to the observing car. Therefore, the predefined threshold effectively establishes the speed boundary for categorizing the movement of an object as either relatively fast or slow, further modulating the memristor state differently.

In the experiments, the driving operations are divided into designed driving and free driving scenarios. In the former, slow driving takes place on a closed road segment, and a pedestrian runs across the road from different sides (left and right), distances (near, medium and far) and speeds (walking and running), aiming to examine the detection ability of the method in 3 widely occurring danger scenarios for autonomous driving as shown in Fig. 4b: 1. Pedestrian running across the road (moment 1); 2. Nearby pedestrian walks across the road (moment 2); 3. The walking pedestrian (moment 3) suddenly runs to another orientation (moment 4). Figure 4c displays 3 representative scenarios during a pedestrian's moving path given in Fig. 4b. The yellow box in Fig. 4c is the example m by n pixel area that is first compressed to a single pixel through aggregating the analog voltage outputs from the m×n pixel region, and the aggregated visual information of this compressed pixel is processed by a memristor in the 25×40 memristor array. The light intensity changes of the pixel are shown in Fig. 4d. Before moment 1, no pedestrian enters, and the slight light intensity change is solely due to the vehicle movements. Upon reaching moment 1, the pedestrian runs into the area from a medium distance, resulting in strong light intensity change, triggering the extraction and encoding module to generate a positive modulation voltage pulse, thus modulating the corresponding memristor into a low-resistance state. After the pedestrian leaves the area, the corresponding memristors enter the neural excitation status and finally return to the high resistance status because the scene information changes slowly. At moment 2, the pedestrian walks into the yellow box area, giving rise to a strong light intensity change again. Note that although the pedestrian is in a slow-motion mode, the distance between the vehicle and pedestrian is short, hence showing the same effect as a pedestrian running from far. At moment 3, the pedestrian enters the detection area from far, and the slow movement characteristics generate a negative voltage pulse that maintains the memristor in a high-resistance state. At moment 4, the sudden movement of the pedestrian strongly changes the light intensity, hence reducing the resistance value of the memristor. However, due to the distance of the pedestrian and the fact that he has already entered the area before the sudden run, the amplitude of the positive pulse generated by the high-frequency feature is smaller compared to the previous more dangerous moments (moments 1 and 2). For this reason, the resistance value of the memristor does not reach to its lowest resistance state. Overall, the 3 danger scenarios are successfully detected.

When the above processing method is applied to the whole image area, global attributes are gathered through the memristor array, and reflected in terms of the memristor array resistance. Figure 4e–g is 3 examples taken from daytime and evening, together with their corresponding differential neuromorphic computing results; the fast-running pedestrian is accurately captured, and afterimages are generated, implying the orientation. Observing the whole image, the memristor state information, even in a single frame, not only distinctly emphasizes rapid changes but also maintains historical change information (Fig. S25). These capabilities provide cleaner and more actionable data for further high-level processing, such as deducing the direction of movement through afterimages, predicting future locations, and other decision-making processes crucial for navigating dynamic environments.

In free-driving experiments, we collected over 100 h of video data in various lighting and weather conditions, as conceptually shown in the middle part of Fig. 5. Ten representative scenarios containing important decision-making associated information for autonomous driving, such as taillight, nearby cars and lane lines, are given as examples in Fig. 5. From the differential neurocomputing processed results, it is clear that the information is successfully retrieved. We then further invite 10 senior drivers to locate dangerous objects they believe are important to safe driving in 10 short videos containing the 10 images given in Fig. 5. Their results are offered as the yellow regions in the processed results, and we can learn that human decisions overlap with the proposed method. For all 10 videos, the overlap rate is more than 94%. The difference arises from the process of important information labeling. The judgment criterion of manual labeling, based on senior drivers' driving/riding experience, tends to prioritize targets (such as pedestrians and cars) that are likely to cause traffic accidents. In contrast, the criterion of our memristor-assisted visual perception method is based on objective motion situation, which is of greater significance for timely decision-making, assigning relatively stationary or slow-moving objects low priority on current driving. Therefore, when both empirical and objective moving targets are present, a person tends to label empirical targets. However, at times, such as when driving with a vehicle traveling at a relatively close speed nearby, the road marking line is more important and can reflect the yawing condition of driving (Fig. 5b). In most cases the empirical and objective moving targets are the same object, but there are exceptions such as in Fig. 5h, leading to a 6% bias in the detection results. The detailed working performance can be found in Fig. S13 to S19.

Compared with dynamic vision camera, our memristor-assisted approach faces the inherent limitations of frame-based image processing, such as a reduced performance under low-light conditions and a narrower dynamic range, while offering the ability to directly

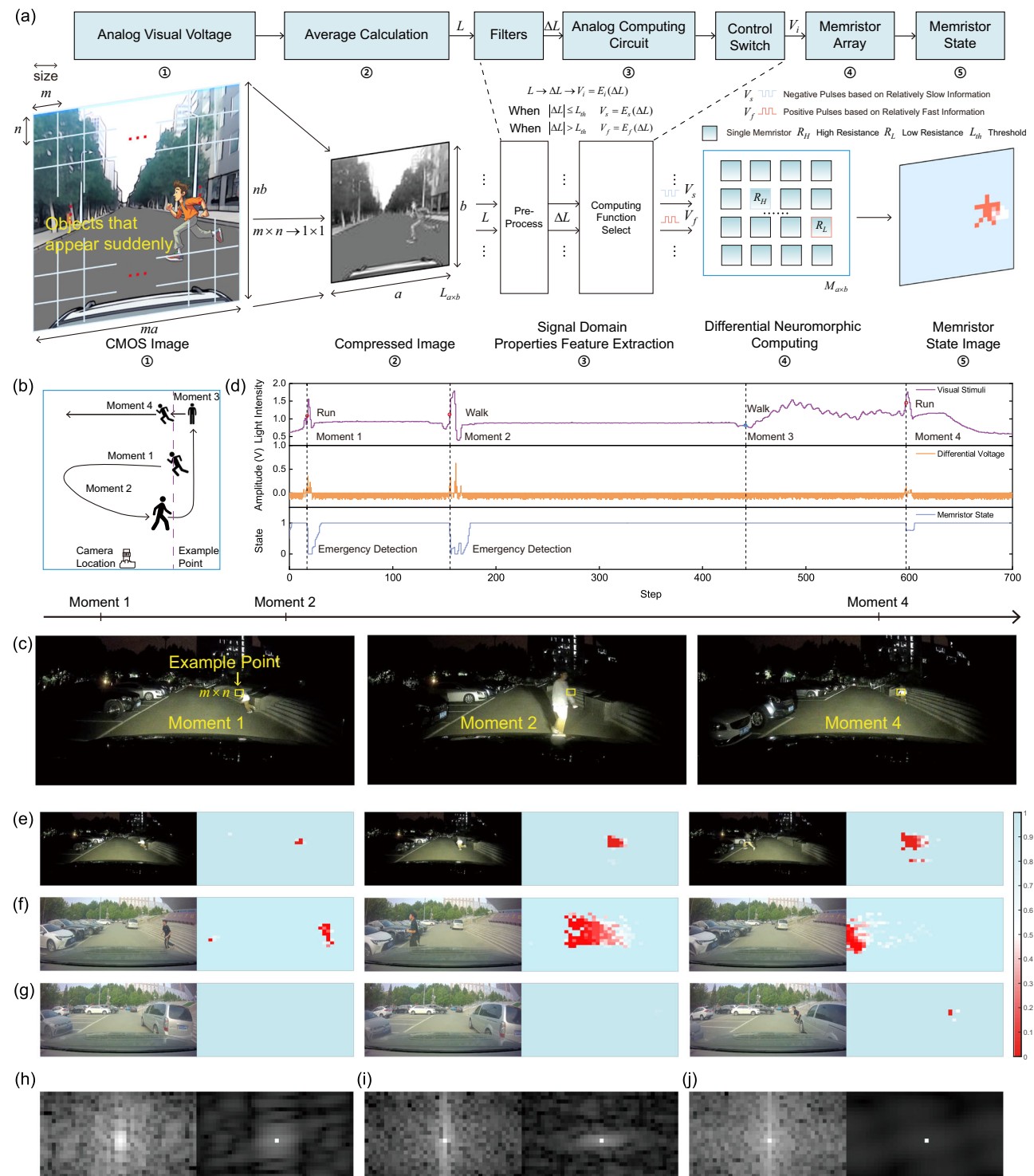

**Fig. 4 | Utilizing differential processing method to process and learn visual information. a** The diagram of visual information differential processing. It is important to note that the hardware components, indicated by symbols such as ①, ②, etc., correspond to the stages of this processing pipeline. **b** The experimental design. **c** The corresponding real-world scenes of the example point visual information processing. **d** The visual information processing within the yellow box in Fig. 4c, including the change in light intensity (from 0 to 2.56), the differentiation voltage, and the memristor state changes (1 represents high resistance state, 0 represents low resistance state). **e** Sudden movement of pedestrian at night. **f** Sudden movement of pedestrian during the day. **g** Sudden movement of pedestrian hiding behind a car, with background vehicles moving slowly. **h** The amplitude spectrum comparison between the compressed image and the differentiated image at the last moment in Fig. 4e. **i** The amplitude spectrum comparison between the compressed image and the differentiated image at the last moment in Fig. 4f. **j** The amplitude spectrum comparison between the compressed image and the differentiated image at the last moment in Fig. 4g.

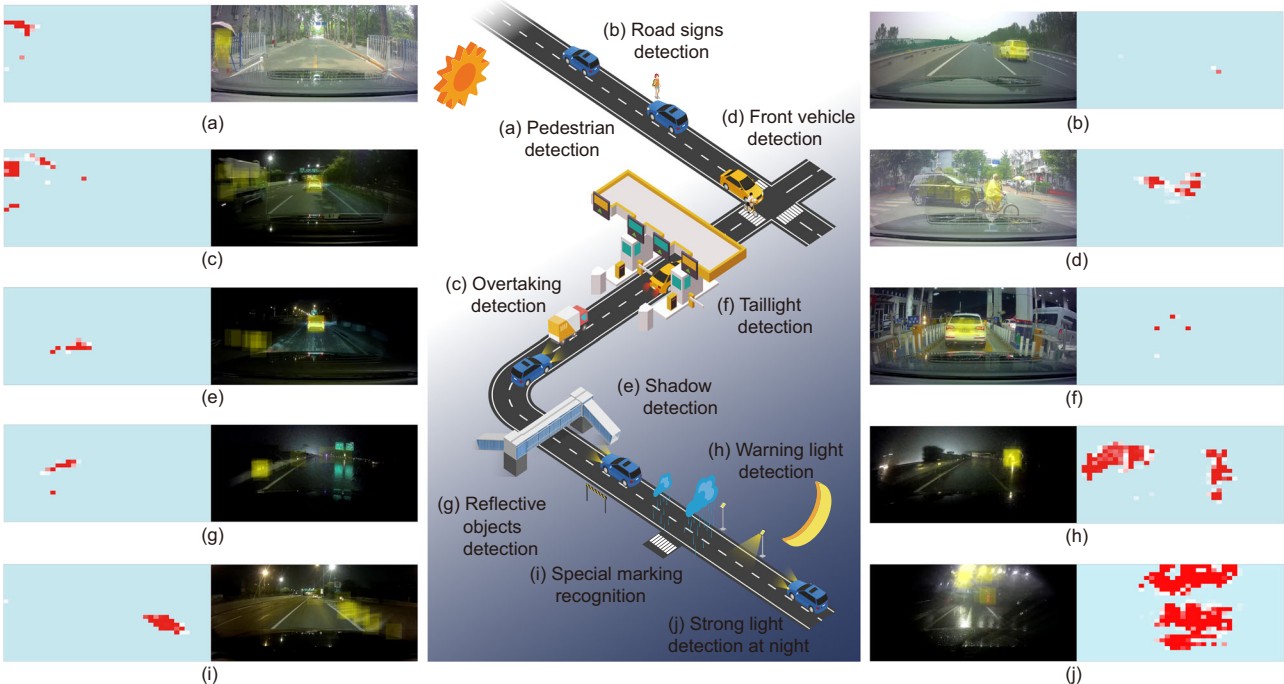

**Fig. 5 | Generalization capability of visual differentiation processing method in unstructured environments. a** Pedestrian detection. **b** Road signs detection. **c** Overtaking detection. **d** Front vehicle detection. **e** Shadow detection. **f** Taillight detection. **g** Reflective objects detection. **h** Warning light detection. **i** Special marking recognition. **j** Strong light detection at night.

generate afterimages that contain crucial temporal information, thanks to the local processing implemented via the memristor-based architecture (Supplementary Discussion 14).

## Discussion

In this study, we propose a memristor-assisted perception method that exploits both the synapse-like characteristics of memristors, and a bio-inspired workflow design enabling robotics to effectively adapt and operate in unstructured environments. This innovative approach utilizes key insights derived from biological analogies to enhance the adaptability and dexterity of neuromorphic computing. Compared to conventional technologies, this memristor-assisted method exhibits higher adaptability and efficiency. However, there are some limitations that need to be addressed for practical real-world application.

### Key insights from biological analogies

Biological perception benefits from the synergy among various receptors and sensory neurons. For example, different types of tactile mechanoreceptors and their corresponding neurons enable the processing of a wide range of pressure stimuli, each with unique characteristics. In the memristive implementation, the above complex synergy process is folded into two computational phases: feature extraction and the processing of corresponding voltage stimuli based on the identified features using memristors. Specifically, our method employs the nonlinear state space of the memristor alongside an adaptive memristive modulation scheme that adjusts its electrical characteristics based on specific features of sensory stimuli. This design enables a memristor to acquire a multifaceted understanding of the surroundings, exploiting the full computational potential of memristors. The memristor state serves as a nexus for integrating historical stimulus data and current sensory data; this state not only adjusts to evolving environmental conditions to provide neuromorphic style adaptation but also offers valuable references in decision-making. Through mimicking biological perception mechanisms, the proposed method of use of a memristive device for

neuromorphic processing is more general and versatile than currently known uses of memristors in literature. The detailed comparison can be found in Supplementary Table 3.

### Comparison with conventional technologies

Differential neuromorphic computing, as a memristor-assisted perception method, holds the potential to enhance subsequent decision-making and control processes. Compared with conventional technologies, both the PID control approach and the proposed differential neuromorphic computing share a fundamental principle of smartly adjusting outputs in response to feedback, they diverge significantly in the data manipulation process (Supplementary Discussion 12 and Fig. S26); our method leverages the nonlinear characteristics of the memristor and a dynamic selection scheme to execute more complex data manipulation than linear coefficient-based error correction in PID. Additionally, the intrinsic memory function of memristors in our system enables real-time adaptation to changing environments. This represents a significant advantage compared to the static parameter configuration of PID systems. To perform similar adaptive control functions in tactile experiments, the von Neumann architecture follows a multi-step process involving several data movements: 1. Input data about the piezoresistive film state is transferred to the system memory via an I/O interface. 2. This sensory data is then moved from the memory to the cache. 3. Subsequently, it is forwarded to the Arithmetic Logic Unit (ALU) and waits for processing.4. Historical tactile information is also transferred from the memory to the cache unless it is already present. 5. This historical data is forwarded to the ALU. 6. ALU calculates the current sensory and historical data and returns the updated historical data to the cache. In contrast, our memristor-based approach simplifies this process, reducing it to three primary steps: 1. ADC reads data from the piezoresistive film. 2. ADC reads the current state of the memristor, which represents the historical tactile stimuli. 3. DAC, controlled by FPGA logic, updates the memristor state based on the inputs. This process reduces the costs of operation and enhances data processing efficiency.

## Limitations for practical use

In real-world settings, robotic tactile systems are required to elaborate large amounts of tactile data and respond as quickly as possible, taking less than 100 ms, similar to human tactile systems[58,59]. The current state-of-the-art robotics tactile technologies are capable of elaborating sudden changes in force, such as slip detection, at millisecond levels (from 500 µs to 50 ms)[59–62], and the response time of our tactile system has also reached this detection level. For the visual processing, suppose a vehicle travels 40 km per hour in an urban area and wants control effective for every 1 m. In that case, the requirement translates a maximum allowable response time of 90 ms for the entire processing pipeline, which includes sensors, operating systems, middleware, and applications such as object detection, prediction, and vehicle control[63,64]. When incorporating our proposed memristor-assisted method with conventional camera systems, the additional time delay includes the delay from filter circuits (less than 1 ms) and the switching time for the memristor device, which ranges from nanoseconds (ns) to even picoseconds (ps)[21,65–67]. Compared to the required overall response time of the pipeline, these additions are negligible, demonstrating the potential of our method application in real-world driving scenarios[68]. Although our memristor-based perception method meets the response time requirement for described scenarios, our approach faces several challenges that need to be addressed for real-world applications. Apart from the common issues such as variability in device performance and the nonlinear dynamics of memristive responses, our approach needs to overcome the following challenges:

### Automatic modulation scheme and control algorithm

Currently, the modulation voltage applied to memristors is preset based on the external sensory feature, and the control algorithm is based on hard threshold comparison. This setting lacks the flexibility required for diverse real-world environments where sensory inputs and required responses can vary significantly. Therefore, it is crucial to develop a more automatic memristive modulation method along with a control algorithm that can dynamically adjust based on varying application scenarios.

### Scalable parallel circuit design

As our method potentially involves controlling a large number of memristors, designing scalable parallel circuits that maintain signal synchronization across extensive memristors poses a significant engineering challenge. Effective practical circuit design must ensure the synchronization and speed of signal processing simultaneously.

In conclusion, this method marks a significant advancement in harnessing inherent characteristics of memristors leveraging functions that in nature lead to perception abilities and support intelligent behaviors through rapid non-linear adaptation features. Due to the small size of memristive devices, the possibility of having high-density memristors over large areas and flexible substrates, and their similarity with the fundamental biological perception mechanisms, the presented method has the potential to enable robotics to possess sensory capabilities on a scale comparable to humans when combined with diverse sensors (Supplementary Discussion 9), allowing them to sense and adapt to the environment efficiently (as depicted in Fig. S20, S21).

## Methods

### Material selection of the memristor

The differential neuromorphic computing method is universally applicable across memristors, regardless of their underlying switching mechanisms, making it a versatile solution for neuromorphic computing applications. This universal ability is evidenced by the material selection, i.e., a commercially available memristor in tactile sensing experiments and a well-recognized simulation model for visual information processing.

### Electrical measurements

Electrical measurements were conducted with a RIGOL DG4062 function/arbitrary waveform generator and MSO1074 oscilloscope. For the memristor U-I test in Fig. 2c, a 10 kΩ resistor was connected in series with the memristor. A 10 Hz, 500 mV sine wave was applied across both components. The voltage across the memristor and resistor was recorded by separate oscilloscope channels to calculate the memristor's current. The oscilloscope was set to normal triggering mode. During the pulse tests shown in Fig. 2d, e, the setup remained unchanged, and voltage pulses were applied with the oscilloscope, capturing the resulting voltages in single mode.

### Control circuit of the memristor

The control circuit of the memristor is designed around an operational amplifier, which is pivotal in achieving precise modulation of the memristor state. Utilizing this design, the memristor state information can be observed based on the output voltage of this circuit. Detailed schematics and tests of this circuit are documented in Figs. S3, S5.

### Tactile processing system

In our tactile system, we process external tactile information by treating the memristor as a synapse, and the modulation of the memristor is dynamically controlled through a multi-branch function $V(\cdot)$ dependent on current sensory features $f(t)$ and memristor resistance $M(x)$, facilitated by the FPGA. The tactile response strength is determined by the product of the stimulus input and the memristor conductance (stimulus input divided by the memristor resistance).

For hazardous stimuli, when current pressure surpasses a predefined value, matching the criteria for dangerous features, positive voltage pulses are generated to increase the memristor conductance. This process aligns with the 'threshold' function of biological nociceptors, as demonstrated in the following formula:

$$\nu(t) = \nu_{noc} \ for \ \ \text{r}(t) \subset r_d, \ t_{sta} < t < t_{end}$$
$$\nu(t) = 0 \ otherwise \tag{1}$$

Where $\nu(t)$ represents the voltage stimuli applied to the memristor, and only if the pressure stimulus meets the hazard characteristics $r_d$ is a voltage pulse with amplitude $\nu_{noc}$ and duration from $t_{sta}$ to $t_{end}$ is generated. For continuous hazardous stimuli, the memristor conductance gradually increases under sustained positive voltage stimuli, achieving 'no adaptation' to dangerous stimuli. The process can be represented as follows:

$$\frac{dx}{dt} = f(x, \nu(t)) < 0 \tag{2}$$

where $x$ represents the memristor state. To realize the 'sensitization' function, we draw inspiration from biology, considering information across various time scales. When the stimulus matches hazardous features, and the memristor conductance surpasses a set threshold, it suggests long-term exposure to danger. Consequently, the amplitude of the positive voltage applied to the memristor is increased to amplify the response to hazardous stimuli further. Upon stimulus removal, recovery pulses are generated to reset the memristor to its initial resistance, achieving the 'recovery' function.

The mild stimuli are processed similarly; the characteristics of external tactile are analyzed, and if the mild criteria are met, negative voltage pulses are generated to reduce the memristor conductance, achieving the 'adaptation' function. This method allows the emulation of both rapidly and slowly adapting receptors by modifying the features (amplitude and width) of negative pulses to adjust the 'adaptation' speed. Upon removal of stimuli or mismatching the mild criteria, the memristor resets to its initial resistance.

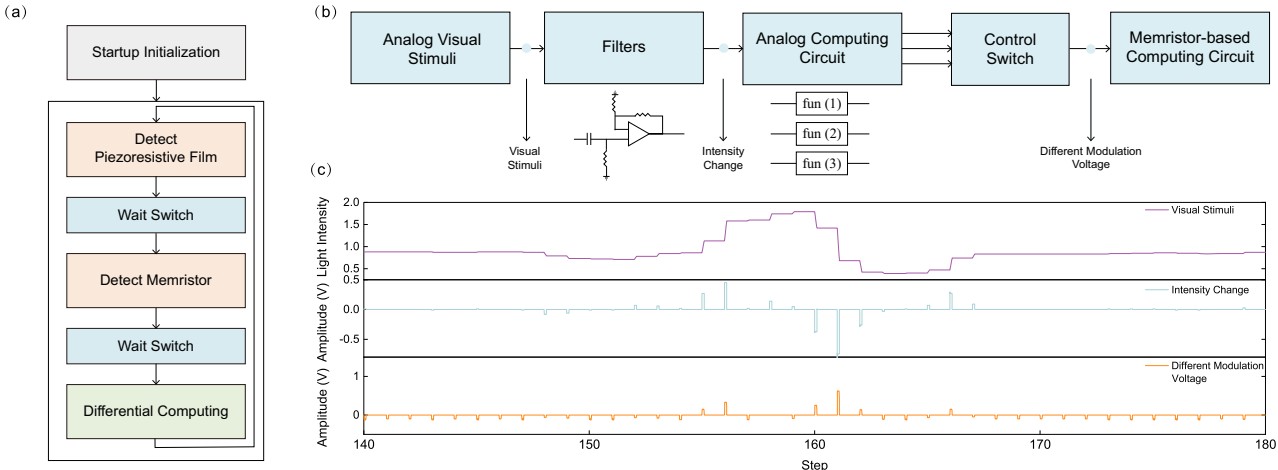

**Fig. 6 | Method used to stimulate memristors in tactile and visual experiments.**
**a** Control schemes in tactile differential processing. The system initiates by entering the initialization phase, where it performs an initial power-on reset on the device. Afterward, the system proceeds to the normal working cycle. Within each working cycle, the system first employs a DAC and an ADC to detect the resistance value of the piezoresistive film. It then waits for the system analog switch to activate when read is completed. Subsequently, it detects the resistance value of the memristor through the memristor read and write control circuit. Once the status of both readings are complete, the DAC circuits generate the corresponding modulation voltage. This voltage is then used to modulate the resistance value of the memristor, achieving the differential processing of pressure information. **b** Circuit design of the visual differential process system. In this simulation, the external visual stimuli captured by the CMOS in-vehicle camera are utilized as the input signal for the system in the form of analog voltage signals. The visual information between two frames undergoes linear changes at a fixed time interval. Subsequently, the changes in visual information are extracted using filters. The extracted information is then translated into different modulation voltages through the analog operation circuit. Finally, the modulation voltage is applied to the memristor using the read and write control circuit. **c** Observed visual stimuli, intensity change extracted by the filter circuits and different modulation voltages applied to the memristor.

To achieve the above tactile processing, we employ the FPGA platform to collect pressure data, generate modulation schemes, and upload data. In every control cycle, the system initially detects the current pressure information using the pressure sensor. Subsequently, the FPGA generates an appropriate modulation scheme based on the obtained pressure data and the current state of the memristor. Following this, a digital-to-analog converter applies the modulation voltage to the memristor. Finally, the system uploads the data of the memristor state and pressure information. The detailed descriptions and schematics are provided in Fig. 6a and S4. More detailed information about mimicking the biological tactile system can be found in Supplementary Discussion 7.

### The detection logic for slip events

Slip events, triggered by external disturbances, are characterized by abrupt decreasing changes in the previously stable contact force. Traditional methods for extracting such features involve recording historical contact forces and analyzing them with current interaction forces[59,69]. For example[69], slip detection can be accomplished by examining the frequency change based on the historical contact force $F_h$ and the current force $F_c$; the frequency change can be determined using the following formula:

$$\Delta f = f(F_c) - f(F_h) \qquad (3)$$

where $f$ is a function mapping observing force points to the frequency information. In the memristive implementation, we have developed a detection logic only based on the current memristor state and contact force change to identify such slip events. This method leverages the memristor state as a cumulative record of the features of historical contact force $F_h$ in differential neuromorphic computing to enhance detection efficiency. Specifically, when the memristor is in a high-resistance state (>225 kΩ) due to the modulation scheme during stable force contact and the piezoresistive film changes drastically, resembling a spike (increases above 350 kΩ from a last point below 250 kΩ), a slip event is inferred to have

occurred. This design allows for the direct interpretation of slip events from the single readout of the current memristor state and piezoresistive film state change without substantial data historical storage and analysis, significantly enhancing processing efficiency. Note that these threshold value selections result from careful hand-tuning based on the memristor's characteristics and the object's contact characteristics.

### Robot experiment

In robot experiments, the FPGA platform is utilized to generate the control commands. Upon activation of the control algorithm, the FPGA sends a command to the DH gripper, triggering the appropriate reflex response timely. These commands are communicated via the Modbus protocol, using the RS485 interface standard. Each command includes a slave number, function code, operation register, operand, and a CRC check code for verification. Once the DH gripper receives a command, it adjusts its gripping force and position to execute the reflex action.

### Visual processing system

To implement large-scale memristor-based visual differential computing, we employ the SPICE simulation platform. The voltage threshold adaptive memristor (VTEAM) model, selected for its versatility, serves as the fundamental computation unit. Detailed information about the electrical characteristics of this model is available in Fig. S11. In visual information processing, images captured by a CMOS sensor in driving settings are converted into analog voltage inputs for the visual differential processing system. Subsequently, these pieces of time-discrete visual information are transformed into continuous voltage data. These voltage profiles are then processed by the memristor-based visual system through SPICE simulation, mimicking the data processing manner after direct integration with the CMOS sensor. The visual process system consists of four main components: filters, analog computing circuit, control switch, and memristor-based computing circuit. Further details of this circuitry are provided in Fig. 6b, c.

In visual information processing, analog filters first extract changes in light intensity, categorizing them into high and low frequencies. High-frequency light information is essential for real-time decision making, while low-frequency corresponds to slowly moving or stationary objects. To process the visual information differently, the high-frequency and low-frequency information are transformed into positive and negative pulses, respectively. The relationship can be represented as:

$$\nu(t) = m \times f_{light} + b > 0 \quad f_{light} \subset f_{high}$$
$$\nu(t) = n \times f_{light} + c < 0 \quad f_{light} \subset f_{low} \tag{4}$$

Where $f_{light}$ is the frequency information of the light intensity, $f_{high}$ and $f_{low}$ represent high-frequency and low-frequency change features, respectively, and the remaining parameters are constant coefficients. When the memristor exhibits a low-resistance state, it has effectively perceived high-frequency stimuli from the external environment. Conversely, when the memristor is in a high-resistance state, it suggests that the changes in light intensity within that area have been slow. More details can be found in Supplementary Discussion 8.

### Technical explanation of the demonstrated visual processing methods

Yellow box: This represents an example case of visual information processing, consisting of a region spanning m×n pixels within the original image. After being compressed, this region transforms into a single pixel point. Subsequently, this compressed point is subject to processing via a memristor, employing a one-to-one approach.

Preprocessing steps: The process involves utilizing filter circuits to extract the change in light intensity $\Delta L$, within the compressed pixel point.

Methods of identification: The light intensity change $\Delta L$ within the compressed pixel point serves as the criterion for classification. Should $\Delta L$ surpass the predefined threshold $L_{th}$, the visual information in this point is categorized as fast. Otherwise, it is classified as slow. This process is implemented by a voltage comparison circuit.

Interpretation of states: For a compressed point categorized as 'fast,' the modulation voltage for the memristor is formulated as:

$$V_f = E_f(\Delta L), \Delta L > L_{th} \tag{5}$$

Otherwise, for a point considered 'slow,' the voltage is expressed as:

$$V_s = E_s(\Delta L), \Delta L \leq L_{th} \tag{6}$$

Speed boundaries: The demarcation of states is reliant on the threshold $L_{th}$, which establishes the speed boundary for categorizing the movement of an object as either relatively fast or slow.

### Constructing components of differential neuromorphic computing

Leveraging the bifurcation of memristor states alongside environmental sensory features, differential neuromorphic computing provides robotics with fine-grained adaptive sensing capabilities in a way that draws parallels to how perception works in biology. This approach can be conceptually described as a cooperation of sensory, signal encoding, and neuromorphic operation modules (Fig. 1b). The first module can be of the desired sensor type according to mimicked biological sensory. It operates by converting a physical stimulus into electrical information and can be described as:

$$P_i(t) = R_i(s_i(t)) \tag{7}$$

where $s(t)$ denotes the physical stimulus, $R$ is the response function of the sensor, and $p(t)$ is output the outputted electrical signal for each $i$-th channel/memristor.

The signal encoding module extracts features from $p(t)$ and then creates the associated memristive encoding schemes to process sensor information in different manners. The created encoding scheme is applied to the memristor in the neuromorphic operation module, and the changed resistive values indicate the properties of the suffered stimuli. Through the three steps above, different features are properly processed by the predesigned differential modulation methods for memristors, yielding a multifeature differentiation-based comprehensive understanding of environmental knowledge. The entire process corresponds to organisms' differential information perception capability in stimuli reception, transduction and processing[70], as expressed below:

$$f_i(t) = F_i(p_i(t))$$
$$\nu_i(t) = V_i(f_i(t), M_i(x_i(t-1)))$$
$$\frac{dx_i}{dt} = C_i(x_i, \nu_i) \tag{8}$$

where $F$ is the extracting function, $f(t)$ is the current extracted features calculated by $F$, $x(t\text{-}1)$ is the memristor state after the previous modulation (at the time step $t\text{-}1$), $M$ is the eigenvalue calculation function whose input variable is the memristor state $x(t\text{-}1)$, therefore $M(x(t\text{-}1))$ is the scalar related to the memristor state, which is used to determine the appropriate modulation schemes, $V$ is the piecewise memristive encoding scheme (condition function) responsible for generating current modulation signals $\nu(t)$, $dx/dt$ is the derivative of the memristor state variable, and $C$ is the state derivative function related to the memristor mechanism, current state and external modulation voltage for each $i$-th channel/memristor.

Notably, in our tactile experiments, the scalar used to determine to the memristive modulation schemes is the memristor resistance, thus $M(x(t\text{-}1))$ refers to the observed memristor resistance value after the last modulation. $V$ is a condition function based on current stimulus strength $f(t)$ and memristor resistance value $M(x(t\text{-}1))$ to generate the modulation voltages $\nu(t)$, as shown in Supplementary Table 1. In visual experiments, the memristor state is not used to stimulate the memristor, and $V$ is a condition function based on the current frequency of light stimuli $f_{light}$ as below:

$$\nu(t) = m \times f_{light} + b > 0 \quad f_{light} \subset f_{high}$$
$$\nu(t) = n \times f_{light} + c < 0 \quad f_{light} \subset f_{low} \tag{9}$$

Where $f_{high}$ and $f_{low}$ represent high-frequency and low-frequency change features, respectively, and the remaining parameters are constant coefficients.

As explained, the proposed method provides a human-like information processing pipeline, which extracts key features of undergoing stimuli in real-time, opening the possibility for intelligent machines to operate in unstructured environments efficiently.

## Data availability

All data supporting this study and its findings are available within the article, its Supplementary Information and associated files. The source data underlying Figs. 2c–e, f, h, i–l, 3b, c, 4d are available in Figshare under accession code https://doi.org/10.6084/m9.figshare.25375696.

## Code availability

All the necessary codes used in the tactile experiments and visual experiments and their descriptions are available in https://github.com/RTCartist/Differential-Neuromorphic-Computing.

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

## Acknowledgements
S.G. acknowledges funding from National Key Research and Development Program of China (grant No. 2023YFB3208003), National Natural Science Foundation of China (grant No. 62171014), and Beihang University (grants No.KG161250 and ZG16S2103). L.G.O. acknowledges funding from EPSRC (grants No. EP/W024284/1, EP/P027628/1, EP/K03099X/1), E.O. was supported by UK Research and Innovation (UKRI Centre for Doctoral Training in AI for Healthcare grant No. EP/S023283/1).

## Author contributions
S.G. and She.W. contributed equally to the work. S.G. and She.W. conceived the idea and proposed the research. She.W. and C.L. conducted electrical measurements. She.W., S.G., C.T., and C.L. designed and performed the tactile experiments. Shu.W. and J.W. collected the origin data for visual simulation. She.W., S.G., C.T. and E.O. performed the visual simulations. S.G. and L.G.O. directed all the research. S.G. L.G.O., H.Z., G.H., A.N and R.D. revised the manuscript. All authors wrote the manuscript, discussed the results and implications and commented on the manuscript at all stages.

## Competing interests
All authors declare no competing interests.
