## [Peer Review File · Nature Communications]

Memristor-Based Adaptive Neuromorphic Perception in Unstructured EnvironmentsREVIEWER COMMENTS

Reviewer #1 (Remarks to the Author):

The paper proposes a novel approach to solving problems in unstructured environments by exploiting the variability inherent to memristors. The concept is intriguing and holds the potential to address a range of challenges. However, several aspects of the paper require further clarification and elaboration.

Main Concerns

1. Clarity and Relevance of Methodology: While the supplementary material provides detailed explanations of the proposed methodology, the main paper could benefit from incorporating more fundamental information to enhance understanding. For instance, the type of memristor used is only explicitly mentioned in Figure 2b, which could be addressed in the main text for better accessibility.
2. Memristor Properties and Scalability: The term "good scalability" should be defined more precisely. Does it refer to memristors' ability to exhibit multiple states or to their number? Both aspects are significant and deserve elaboration.
3. Amplification and Sensor Choice: The use of nociceptors for amplification is unconventional in grasping applications, where mechanoreceptors are typically employed. This deviation from conventional practice warrants justification. Is it a consequence of the model's design or a necessity arising from the use of memristors?
4. Unreferenced Claims: The opening statement of the main text, "Traditionally, the adaptability of organisms has been studied..." lacks supporting references.
5. Memristor Adaptation: The concept of memristor adaptation needs to be clearly defined. Does it refer to behavioral adaptation analogous to mechanoreceptors or simply to material-level changes?
6. Robot Adaptation Definition: The term "adaptation to robotics" requires more precise articulation. Does it encompass the ability to switch between pre-learned tasks or the capability to acquire new skills? These are very different tasks and both very large to address.
7. Corresponding Neurons: The expression "corresponding neurons" is ambiguous. At which level of the afferent pathway do these neurons reside?
8. Dynamic Environment Limitations: The connection between the two sentences "Specifically, by extracting the main stimulus..." and "...highly dynamic unstructured environment (9-13)" is not fully apparent. Why does the extraction of a dominant stimulus render traditional approaches inefficient in dynamic environments? Classic von Neumann machines can handle numerous tasks in dynamic settings, and this limitation needs to be better explained.
9. Results vs. Methods: The section on "specific components of differentiation processing and environmental learning" in the Results section seems more methodological than a true result. It would be more suitable to move this section to the end of the paper and integrate additional information on memristors to enhance its connection to the overall discussion.
10. Biological Behavior and Recovery State: The term "recovery state" in the biological behavior section is unclear. Is it synonymous with the refractory period?
11. Tactile Task Timing: The video demonstration of the tactile task highlights a rapid response time, while the textual information conveys longer durations. This discrepancy needs clarification. Specifically, what occurs between 8.2s and 17.4s during the grasping process?
12. Figure Explanations: Figures 2h, 2j, and 4a lack adequate explanations. They should be accompanied by descriptions of the yellow box representation, identification methods, pre-processing steps, state interpretation (e.g., distance), and speed boundaries to enhance clarity.
13. Discussions Section: The Discussions section primarily serves as a summary of conclusions. It would benefit from expanding on the results and potentially incorporating insights from the supplementary material.
14. References: The reference list could be improved by:
 - Placing the need for robotics at the beginning, rather than neuromorphics.
 - Citing another relevant Nature Communications paper on neuromorphic and autonomous robots (<https://www.nature.com/articles/s41467-022-28487-2>).
 - Remembering to cite CMOS papers, as neuromorphics encompasses more than memristors.

Reviewer #2 (Remarks to the Author):

A. General comments

The authors present a somewhat "out of the ordinary" use of memristor technology as a sort-of PID controller, where they exploit bifurcations of the non-linear operation regimes of the memristor to facilitate fine-grained adaptive sensing capabilities in a way that draws parallels to how perception works in biology. They use two use-cases to demonstrate the enabled capabilities with detailed and comprehensive evaluation.

Overall I welcome their contribution as stimulating, relevant, and interesting, that deserves attention from the community.

The manuscript as it stands together with the supplementary materials are reach in content to assist the understanding of the contribution, but I think it needs a re-positioning and re-organisation (and occasionally some introductory claims need to be toned down). On reading the main manuscript i had initially difficulty to grasp the context and decode the approach or connect some of the main claims to the presented results. It all became much clearer upon reading the supplementary materials. However i am of the opinion that the supplementary materials are there to explain the details, not to decode the key ideas of the paper and make them understandable! Therefore my comments, critique, and suggestions, aim at making the main manuscript more informative and self-contained and the supplementary materials ... well supplementary of details.

Some general remarks to be considered throughout the paper regarding general positioning and easing contextualization are the following:

- The way you discuss and defend "learning" in this work is a bit unusual or misleading to me, I suggest "(online) adaptation" or "short-term plasticity", which are more short-term (duration of the task), are more appropriate and less confusing in the context of the paper.
- Similarly the term "differential computing" to describe dynamic function selection (or a multi-branch functions) is also confusing/misleading. It becomes even more confusing when you also use the term "differential encoding" (which i believe is indeed correctly used in this case).

My suggestion is to either replace them .. or define their meaning explicitly upfront in the text.

B. Specific comments and suggestions

B.1. Title:

The title is a bit too abstract/vague. Some keywords missing that would couple it to the subject matter of the paper is "memristor", "adaptive sensing", "(neuromorphic) perception", or similar

E.g. "Memristor-based adaptive neuromorphic sensing in unstructure environments" or something of that sort that suits the authors liking.

B.2 Abstract:

The abstract needs re-writing altogether: What is the challenge, what is the contribution towards addressing it, how did you test it, what you achieved.

> "Efficient operation of intelligent machines in the real world requires methods that allow them to understand and predict the uncertainties presented by the unstructured environments ..."

While there is nothing wrong with this sentence, more relevant to your work is "..to understand and adapt to conditions of unstructured environments.." or "..evaluate the uncertainties presented

by the unstructured environments.."

> "Current methods rely on pretrained networks .. suffer .. limited generalization capabilities".

... in unseen or unstructured environments. I believe overall large pretrained networks are rather good at generalisation unless trained with little data or the data distribution is very dynamic (which is the scope of the use-cases in the paper). My point is you should focus on perception in unstructured environments.

> "we present a memristor-based differential neuromorphic computing, perceptual signal processing and learning method for intelligent machines"

My issue here is the use of terms "differential computing" and "learning" as explained in the general remarks.

> "The main features of environmental information such as amplification (>720%) and adaptation (<50%) of mechanical stimuli encoded in memristors, are extracted to obtain human-like processing in unstructured environments"

Very cryptic sentence without prior context at this point. I suggest removing it.

> "takes advantage of the intrinsic multi-state property of memristors and exhibits good scalability and generalization, as confirmed by validation in two different application scenarios"

scalability and generalization in what sense? IMHO you showed fine-grained/good adaptation and generality (for different sensing modalities).

> ".. through fast learning (in ~1 ms) "

"adaptation" or the more biologically relevant "short-term plasticity", is more preferred term

> "... adapting to diverse sensing technologies"

You probably mean different sensing modalities (vision, tactile sensing)?

B.2 Introduction

You need to improve the general positioning, by putting focus on adaptation or adaptive-control in sensing (for feature extraction), not quoting learning because learning implies more longer-term plasticity/adaptation.

Somewhere you need to clearly mention that you exploit the bifurcations of the non-linear state-space of the memristor for providing neuromorphic style sensing/adaptation. (Analogous to and possibly better than PID control?)

The last sentence of S1 is more informative than any affirmation in the intro about the work presented and should therefore be brought in the intro: "We aspire to equip these intelligent machines with a comprehensive perceptual capability akin to that of living organisms, enabling them to better comprehend environmental information."

> "the acquisition of diverse types of physical information"

you mean "integration" ?

> "these receptors and neurons can rapidly form a complex set of intricate network-based perception functions, such as integration and firing, potentiation and depotentiation, associative memory, environmental mapping, and motion control tasks."

Integration firing, (de)potentiation/learning, are not "perception"-level functions rather intricate/basic nnet operations at a lower-level than perception.

> "In contrast, conventional intelligent machines operate within the von Neumann separated storage and computation architecture, which makes it inefficient to differentially process highly dynamic unstructured information (9–13)"

This sentence as an antithesis is irrelevant. Whether the compute substrate is Von-Neumann or not (inefficient or not) an emulated artificial nnet implements the same algorithm and the task performance should be (approximately) the same.

> "This can be resolved with memristors, as they integrate storage and computation capabilities, coupled with synapse-like characteristics, ... "

Such as ? (what synapse like characteristics)

> ".. perceptual signal processing and learning method for intelligent machines."

Scaling of the weights is not automatically implying "learning ability" (long-term plasticity), however it is adaptation. I understand why you defend it as online learning (after reading the supp material), but as explained in the general comments I think it is misleading/overstatement.

> "encompasses multidimensional features"

multidimensional and multimodal?

> "on keeping a memristor to a fixed receptor (Figure 1a), e.g. a nociceptor (22–25), thus restricting their full potential"

A bit more elaboration please for context. Particularly to explain how the use of the memristor in 23,23 differ from the use here? Is it the different M(.) used/selected by the digital part to modulate different regions of operation of the memristor? (based on the explanations of 22,23 in S2).

B.3 Results

S7/8 entry sentences need to be brought in the main text too to contextualize the use-cases either in the methods or here in the results sections.

> "The first can be of the desired sensor .."

The first module can ..

> "and $p(t)$ is the outputted electrical signal .."

output electrical signal

> "The extraction and modulation scheme generation are both conducted in a field programmable gate array (FPGA) platform, as shown in Figure 2a"

I think the explanations are a bit fuzzy and ambiguous, they are much better explained in the supp material. When you say "modulation scheme generation", you mean function *selection* for the modulation scheme? Is a modulation scheme dynamically generated?

> "After the attribute of a force is extracted .."

Rather magnitude of the force? Is this not the criteria for the selection of the appropriate modulation scheme?

> "The curve trends are similar to the biological response strengths in Figure 2g"

Fig 2g is not showing much, no curves anyway, or else at least explain it a bit?.

> "Specifically, during the encoding process, we additionally consider the state of the memristor. When its state falls below a certain threshold (100 k Ω in this case), .."

Where is the switching happening? And how are the thresholds determined? Are they learned, or related to the environment or were they found based on hand-tuning for the specific memristor?

> "tactile stimuli is offered (Figure 2g), exceeding the state-of-the-art works .."

You mean Fig 2f ?

> "Upon detecting a slip event onset, the robot prompts "

How is the slip detected ?

> Line 216-243

In this paragraph it is not clear to me how/why the information of the memristor state is more useful or informative than the information from the differential encoding of the sensor data? Especially since the stimuli from APS camera encodes analog domain in the pixel intensities.

Also in this use-case it has not been clear to me, if your setup connects directly on the camera sensor array with memristors or if you process image data from the APS camera.

B.4 Discussion

Discussion is short and mostly superficial. I would use the space to bring in summarised form some of the content from the supp material to enlighten the biology analogy discussion. Specifically S5/6 are paramount for the understanding of the paper and hence need to be brought in the main text.

I believe a very welcomed input would be a comparison/discussion of relevance/differences and advantages/disadvantages to PID control!

Also since memristors are useful for computational efficiency, a relative computational cost indication of using the memristor as such versus implementing a model of the memristor (in the digital domain) to achieve similar functionality, would be very welcome.

> " ..allowing them to genuinely learn and efficiently comprehend the environment."

Well comprehension is far fetched probably, it requires reasoning abilities. And the analogy to "learning" as opposed to adaptation, has not been explained at all until this point. (Leaving out mention of "learning" altogether is best as suggested in the general comments).

B.5 Supp material

S4 needs proper elaboration and corrections. Eg.

> fxx, fGG, fxG

are not explained

> "..., the function $h()$ is a linear operation or a multiplication and division operation; thus $M \leq 0$ and $M > 0$."

Not obvious at all this conclusion

> "the derivative with respect to y : $dy =$ "

the differential of y maybe? or the temporal derivative dy/dt ?

> $h(s,G)$

not explained

Reviewer #2 (Remarks on code availability):

I have looked at the github repository, however there is no documentation or information on what is available where, or how to use the code, or what the code is to be used for?

For example what can one do with this code? Would reproducibility require models of the memristors, or bitfiles for FPGAs? Is the code meant to generate graphs from pre-collected data? What is in the dataset?

No readme files and no instructions in the sup material.

Reviewer #3 (Remarks to the Author):

Clarity and Coherence:

The paper introduces the concept of differential neuromorphic computing with memristive and in-memory computing devices. The paper also claims a giant leap (line 39) that would link the presented results to human low-level perception mechanisms, which would then be able to "generate smart high-level decisions in the real world" (lines 41-42).

While I recognize the inspiration with human perception as drawn in Figure 1, I also recognize that these are very general claims. Such claims make the narrative's arguments unclear and unspecific until the reader further examines the results section. Additionally, a clear comparison with the state of the art is currently missing. In fact, the results could be more comprehensively and quantitatively compared (clearly) to other implementations of touch or visual information processing with the many other neuromorphic and in-memory computing implementations. The lack of more specific claims and the lack of comparisons with SOTA is a significant drawback of the current version of the paper. Also, it is never clear in the paper if the results come from simulations or actual hardware devices. Only when inspecting the 43 pages of supplementary material does the full system architecture become clear (fig. S4, S13). For clarity, I suggest bringing figures S4 and S13 or a simplified version of those into the main manuscript.

Originality:

It is worth noting that the concept of differential processing is already present in many state-of-the-art neuromorphic sensors, such as silicon retinas (or dynamic vision sensors), silicon cochleas, and other touch sensors. However, these endeavors have not been mentioned in the current paper. In addition, many works in the literature showing various materials (e.g., electrochemical metallization materials, phase-change materials, ferroelectric materials, ionic/electronic hybrid materials) have been employed to fabricate in-sensing computing touch sensory processing, and these are also not mentioned in the current paper. Currently, the work provides little originality

over the current state of the art, as it fails to compare with those.

Methodology:

While the methodology is adequately executed, it needs more focus and distinctly addresses the paper's key contributions. A comparative analysis, perhaps in the form of a table, highlighting improvements over current SOTA technologies (e.g., latency, amplification) is necessary. The paper's abstract claims advancements in three areas:

- 1- Differential neuromorphic computing (for which novelty needs to be clarified and compared to a rich literature of sensors and devices in neuromorphic engineering)
- 2- Hardware implementation of sensory processing with memristor arrays (which seems to be a significant contribution to the current work and for which the novelty is high)
- 3- Bridging low-level sensory processing with high-level decision-making (which is not being demonstrated in this work and not compared with SOTA)

More clearly stated, the methodology does not address these three main aspects, which are claimed in the abstract, and does not show how the in-memory computing memristive approach benefits all of these aspects. Also, the current limitations of deep learning and neuromorphic sensory systems and how this paper goes beyond those (other than amplification and lower latency, which are also not accurate, for example, compared to silicon retinas) need to be clarified.

Impact:

The impact of the work, in its current status, is minimal, as claims are too general, the state-of-the-art comparison needs to be included, and the novelty of differential neuromorphic computing needs to be clarified.

Reviewer #3 (Remarks on code availability):

The code is minimal, the readme does not include:

- 1- configuration instruction
- 2- license
- 3- dependencies
- 4- required software and version

Response Letter to Reviewers' Comments

We sincerely appreciate the valuable time and efforts the reviewers have spent reviewing our manuscript. The insightful comments and feedback are instrumental in improving the quality of our work. We believe we have addressed all of the reviewer's comments, resulting in a manuscript that is more rigorous in content and clearer in presentation. Our point-by-point responses to the reviewers' comments are as follows.

Reviewer #1

The paper proposes a novel approach to solving problems in unstructured environments by exploiting the variability inherent to memristors. The concept is intriguing and holds the potential to address a range of challenges. However, several aspects of the paper require further clarification and elaboration.

Response:

Thank you very much for your kind recognition of our work's novelty and potential impact. Our detailed responses to your valuable comments are provided below.

Comment #1

Clarity and Relevance of Methodology: While the supplementary material provides detailed explanations of the proposed methodology, the main paper could benefit from incorporating more fundamental information to enhance understanding. For instance, the type of memristor used is only explicitly mentioned in Figure 2b, which could be addressed in the main text for better accessibility.

Response:

Thank you for your kind suggestion on improving the clarity. We have incorporated fundamental information into the main paper to enhance understanding, including the type of memristors, the switching mechanisms, and the simulation environment of the visual differential process experiment.

To help readers better understand the content, the original manuscript has been revised as follows (Red represents modified or newly added text):

(Article) On page 2, lines 85-89: In the former, we **utilized a single Self-Directed Channel (SDC) memristor, a type of chalcogenide-based electrochemical metallization (ECM) device, to** emulate nociceptors to achieve amplification (>720%) for hazardous stimuli and adapting receptors to achieve adaptation (<50%) for mild stimuli, which play a key role in grasping unknown objects. In the latter, we differentiate visual information by a 40×25 memristor matrix **via SPICE simulations** and reach a commendable accuracy of 94% in extracting critical information within 10 autonomous driving scenes after comparison with human labelled results.

Comment #2

Memristor Properties and Scalability: The term "good scalability" should be defined more precisely. Does it refer to memristors' ability to exhibit multiple states or to their number? Both aspects are significant and deserve elaboration.

Response:

Thank you for offering us an opportunity to make this term clear. The term “scalability” here refers to the proposed method’s ability to **handle different numbers of memristors**. In our study, this method can be used by a single memristor to process pressure information, and it can also work with a 25×40 memristor array to compute visual information. Hence, we used the term “scalability” to emphasize this in the original version.

To avoid confusion for readers, we removed the term and explained the ability to handle different numbers of memristors in a sentence below.

(Article) On page 3, lines 88-92: In the latter, we differentiate visual information by a 40×25 memristor matrix via SPICE simulations and reach a commendable accuracy of 94% in extracting critical information within 10 autonomous driving scenes, after comparison with human labelled results. **These two experiments showcase that the proposed method has the capability to handle a different number of memristors.**

Comment #3

Amplification and Sensor Choice: The use of nociceptors for amplification is unconventional in grasping applications, where mechanoreceptors are typically employed. This deviation from conventional practice warrants justification. Is it a consequence of the model's design or a necessity arising from the use of memristors?

Response:

We highly agree that employing nociceptors for amplification is unconventional for grasping tasks, typically mechanoreceptors being the norm. To explain more clearly the reason for using nociceptors, we add the following text to the Supplemental Information as **Supplementary Discussion 10**.

(SI) On page 18: For humans, nociceptors play a vital role in interacting with environments by protecting against dangerous stimuli, suggesting their utility in robotics for increasing safety¹⁻³. Specifically for grasping tasks, nociceptors facilitate the safe establishment of contact through amplifying dangerous stimuli. For example, Ravinder et al. developed a neuromorphic e-skin for robots to interact with environments, with a novel design to emulate the nociceptor to achieve pain reflex⁴. Mathews et al. demonstrated a self-healing material-based neuromorphic nociceptor that allows a sensorized robotic arm to enhance the response to noxious stimuli and trigger motor response immediately to avoid potential physical damage⁵. To enhance the safety of robotics grasping in unstructured environments, our manuscript presents the use of memristor-based nociceptors to detect and amplify the noxious stimuli.

It is worth noting that nociceptor-based amplification is the consequence of the model's design rather than a necessity arising from the use of memristors. The synapse-like characteristics of memristors bear a foundational resemblance to biological receptors, allowing for diverse memristor-based receptor implementations, such as slow-adapting receptors, fast-adapting receptors, etc.; nociceptor is merely one kind of them. Thus, the use of the nociceptor in our manuscript is the consequence of the model's design rather than a necessity arising from the use of memristors.

Comment #4

Unreferenced Claims: The opening statement of the main text, "Traditionally, the adaptability of organisms has been studied..." lacks supporting references.

Response:

We have added the following references to support this statement; thanks.

The **added references** are listed as:

1. Stein, Barry E., Terrence R. Stanford, and Benjamin A. Rowland. "Development of multisensory integration from the perspective of the individual neuron." *Nature Reviews Neuroscience* 15.8 (2014): 520-535.
2. Peterka, Robert J. "Sensory integration for human balance control." *Handbook of clinical neurology* 159 (2018): 27-42.
3. Van Atteveldt, Nienke, et al. "Multisensory integration: flexible use of general operations." *Neuron* 81.6 (2014): 1240-1253.

Comment #5

Memristor Adaptation: The concept of memristor adaptation needs to be clearly defined. Does it refer to behavioral adaptation analogous to mechanoreceptors or simply to material-level changes?

Response:

We appreciate your feedback highlighting the need for clearly defining ‘memristor adaptation’. The concept ‘adaptation’ refers to the behavioral adaptation analogous to mechanoreceptors.

After going through the entire manuscript, we find out that this term is also used in the sentence ‘... and the memristor is modulated into an adaptation state (high-resistance state), achieving safe and stable grasping’. We believe this easily leads to confusion. Thus, we have revised the statement of “adaptation state” to “high-resistance state”, and the term “adaptation” is clearly defined in the resubmitted manuscript:

(Article) On page 5, line 178: the memristor is modulated **into the high-resistance state**, achieving safe and stable grasping

(Article) On page 3, lines 123-125: **the adaptation refers to the adaptation processing of mild pressure (the behavioral adaptation of the tactile system analogous to mechanoreceptors)**

Comment #6

Robot Adaptation Definition: The term "adaptation to robotics" requires more precise articulation. Does it encompass the ability to switch between pre-learned tasks or the capability to acquire new skills? This are very different tasks and both very large to address.

Response:

Thanks for your kind question. The term 'adaptation to robotics' encompasses the ability to **switch between pre-learned tasks**, rather than the capability to acquire new skills. We believe the statement 'The perception method should be scalable and generalizable, allowing intelligent machines to perceive and understand information in different modalities (including vision and touch) and adapt flexibly to unlearned or unexperienced scenarios or stimuli' in the SI file may lead to this confusion. Hence, we change 'adapt flexibly to unlearned or inexperienced scenarios or stimuli' to 'switch flexibly to pre-learned tasks'. The revised **Supplementary Discussion 1** is detailed below:

(SI) On page 5: ...The perception method should be of good adaptability and generality, allowing intelligent machines to perceive and understand information in different modalities (including vision and touch) and **switch flexibly to pre-learned tasks**.

Comment #7

Corresponding Neurons: The expression "corresponding neurons" is ambiguous. At which level of the afferent pathway do these neurons reside?

Response:

Thank you for highlighting the need to clarify the term 'corresponding neurons'. In our context, this term specifically refers to sensory neurons located in the sensory ganglia. To provide a clear expression for readers, the manuscript has been revised as:

(Article) On page 2, lines 46-47: For a given stimulus, multiple types of receptors and **subsequent sensory neurons located in sensory ganglia** participate in differential processing, and they adjust their structure and synaptic weight on the features of external stimuli

Comment #8

Dynamic Environment Limitations: The connection between the two sentences "Specifically, by extracting the main stimulus..." and "...highly dynamic unstructured environment (9-13)" is not fully apparent. Why does the extraction of a dominant stimulus render traditional approaches inefficient in dynamic environments? Classic von Neumann machines can handle numerous tasks in dynamic settings, and this limitation needs to be better explained.

Response:

Thank you for helping us improve the logical connections. The limitation is the substantial power consumption and notable time delays arising from von Neumann architecture when handling numerous tasks in dynamic settings. It is known that von Neumann architecture separates storage and computation, and drawbacks brought by this separation become significant in unstructured environments, where sensory data requires substantial storage for feature extraction. The frequent transfer of information between storage and computation units in von Neumann architecture leads to significant power usage and delays. These limitations become increasingly pronounced as the volume of perceived sensory data approaches the human-level quantities.

As this sentence may lead to a logic gap for readers, we removed it from the manuscript. In contrast, we explain more details in the SI file to help readers have a better understanding:

(Article): ~~In contrast, conventional intelligent machines operate within the von Neumann separated storage and computation architecture, which makes it inefficient to differentially process highly dynamic unstructured information (9–13).~~

Specifically, the original statement “Conventional intelligent machines operate within the von Neumann separated storage and computation architecture, leading to transmission bottlenecks and processing delays.” in **Supplementary Discussion 2** has been replaced with the statement below:

(SI) On page 7: ~~Conventional intelligent machines operate within the von Neumann separated storage and computation architecture. When handling numerous tasks in dynamic settings, the frequent transfer of information between storage and computation units in von Neumann architecture leads to significant power usage and delays. These limitations become increasingly pronounced as the volume of perceived sensory data approaches the human-level quantities.~~

Comment #9

Results vs. Methods: The section on "specific components of differentiation processing and environmental learning" in the Results section seems more methodological than a true result. It would be more suitable to move this section to the end of the paper and integrate additional information on memristors to enhance its connection to the overall discussion.

Response:

We agree that the section on specific components of differentiation processing and environmental learning is more methodological than a true result. Thus, the complete statement about specific components of differential processing and environmental learning has been moved to the end of the **Method** Section and revised as follows:

(Article) On page 10, lines 397-428:

Constructing Components of Differential Neuromorphic Computing

Leveraging the bifurcation of memristor states alongside environmental sensory features, differential neuromorphic computing enables intelligent machines with fine-grained adaptive sensing capabilities in a way that draws parallels to how perception works in biology. This approach can be conceptually described as a cooperation of sensory, signal encoding, and neuromorphic operation modules (Figure 1b). The first module can be of the desired sensor type according to mimicked biological sensory. It operates by converting a physical stimulus into electrical information and can be described as:

$$p_i(t) = R_i(s_i(t))$$

where $s(t)$ denotes the physical stimulus, R is the response function of the sensor, and $p(t)$ is output the outputted electrical signal for each i th channel/memristor.

The signal encoding module extracts features from $p(t)$ and then creates the associated memristive encoding schemes to process sensor information in different manners. The created encoding scheme is applied to the memristor in the neuromorphic operation module, and the changed resistive values indicate the properties of the suffered stimuli. Through the three steps above, different features are properly processed by the predesigned differential modulation methods for memristors, yielding a multifeature differentiation-based comprehensive understanding of environmental knowledge. The entire process corresponds to organisms' differential information perception capability in stimuli reception, transduction and processing⁶, as expressed below:

$$\begin{aligned} f_i(t) &= F_i(p_i(t)) \\ v_i(t) &= V_i(f_i(t), M_i(x_i(t-1))) \\ \frac{dx_i}{dt} &= C_i(x_i, v_i) \end{aligned}$$

where F is the extracting function, $f(t)$ is the current extracted features, $x(t-1)$ is the memristor state after the previous modulation, M is the eigenvalue used for the memristor weight update and associated with its state, V is the piecewise memristive encoding scheme responsible for generating current modulation signals $v(t)$, $\frac{dx}{dt}$ is the derivative of the memristor state variable, and C is the state derivative function related to the memristor mechanism, current state and external modulation voltage for each i th channel/memristor.

As explained, the proposed method provides a human-like information processing pipeline, which extracts key features of undergoing stimuli in real-time, opening the possibility for intelligent machines to operate in unstructured environments efficiently.

Comment #10

Biological Behavior and Recovery State: The term "recovery state" in the biological behavior section is unclear. Is it synonymous with the refractory period?

Response:

Thank you for requesting clarification on the term 'recovery state'. In our context, this term specifically describes the process in which the memristor returns to its initial middle resistance after removing external sensory stimuli. In this process, the memristor-based system maintains the capability to respond to external stimuli. In contrast, the refractory period in biology refers to the phase during which neurons are temporarily unable to be excited again after an initial excitation. Hence, they have different meanings. The use of the term 'adaptation' mainly derives from the fact that the term is frequently utilized in research involving memristor-based nociceptor systems, referring to that the nociceptor resets its sensitivity to external stimuli through modulating the memristor to the initial state after removing the stimuli^{7,8}. We acknowledge that the sentence 'This arrangement facilitates the adaptive, normalization, and nociception perception of external stimuli, aligning with 3 types of biological perception behaviors, i.e., nociception, adaptation, and recovery (Figure S7, S8 and Supplementary Discussion 7)' may lead to confusion. Thus, an additional explanation is added as:

(Article) On page 3, lines 122-126: ...**adaptation, recovery and nociception** (Figure S7, S8 and Supplementary Discussion 7). **The nociception refers to the amplification processing of strong pressure, the adaptation refers to the adaptation processing of mild pressure, and the recovery indicates resetting the memristor to its initial state, akin to the sensitivity reset of the biological receptor sensitivity after removing external stimuli.**

Comment #11

Tactile Task Timing: The video demonstration of the tactile task highlights a rapid response time, while the textual information conveys longer durations. This discrepancy needs clarification. Specifically, what occurs between 8.2s and 17.4s during the grasping process?

Response:

Thank you for requiring clarification regarding the observed timing discrepancy of our tactile tasks. Across video demonstration and textual information, both illustrate the robotic tactile system's rapid response to pain reflex and object slipping events, with consistent patterns observed in the piezoresistive film and memristor state during the rapid responses.

The variation in manipulation duration stems from our decision to streamline the video demonstration for conciseness. As such, the robot's manipulation time required to adjust its grasping position and gesture for subsequent safe handling in task 1 is not demonstrated in the Supplementary Video 1, resulting in a duration discrepancy. To avoid potential confusion, the original statement 'At 8.2 s, the memristor resistance falls below the threshold of 35 k Ω , and the pressure stimulus is amplified by 500%. This indicates that the robot has been in contact with the sharp point, triggering the pain reflex. Subsequently, the robot learns to change the grasping posture at 14.4 s.' has been replaced with the following statement about the manipulation process between 8.2s and 17.4s:

(Article) On page 4, lines 171-177: ...At 8.2 s, the memristor resistance falls below the threshold of 35 k Ω , and the pressure stimulus is amplified by 500%. This indicates that the robot has been in contact with the sharp point, triggering the pain reflex. **Based on this pain experience, the robot changes its grasping position and hand gesture. Specifically, the grasping position of the robot is adjusted to an appropriate position to expect the subsequent safe contact between 8.2 and 14.4s. Subsequently, the robot hand moves again at 14.4s and makes contact with the object at 16.1s.** By 17.4 s, the pressure perceived by the contact point stabilizes,

Accordingly, explanatory notations have been added to Figure 3 as shown below.

Figure R1 (The revised Figure 3). Grasping diverse objects in unstructured environments. (a) Overall logical flowchart of learning environmental information. (b) Learning the properties of sharp objects and ultimately achieving safe grasping. (c) Learning the properties of slippery objects and ultimately achieving stable grasping.

Comment #12

Figure Explanations: Figures 2h, 2j, and 4a lack adequate explanations. They should be accompanied by descriptions of the yellow box representation, identification methods, pre-processing steps, state interpretation (e.g., distance), and speed boundaries to enhance clarity.

Response:

We apologize for the omission of adequate explanations for these three charts.

1. For Figures 2h and 2j, the explanations are added as detailed below:

(Article) On page 4, lines 135-139: **The experimental results in Figure 2f demonstrate the responses of this tactile system to hazardous stimuli, with a more than 170% amplification of the hazard signal through applying positive encoding pulses to the memristor. In contrast, when perceiving mild tactile stimuli (Figure 2h), the memristor is modulated by the negative encoding pulses, achieving a more than 50% attenuation of the mild signal.**

2. Figure 2j was mistakenly written as Figure 2g, it is now corrected as detailed below:

(Article) On page 4, line 156: In our work, a >720% amplification of tactile stimuli is offered **(Figure 2j)**, exceeding the state-of-the-art works (22, 25, 33).

3. For Figure 4a, we acknowledge that the current representation is unclear. Hence, we add descriptions of the yellow box representation, the methods for identifying fast information, pre-processing steps, state interpretation (e.g., distance), and speed boundaries. Note that the interpretation of state information and the determination of speed boundaries within our system are achieved by analyzing the change in light intensity of a single pixel across two successive frames. This variation is classified as either fast or slow by comparing it with a hand-tuning threshold. In scenarios involving moving objects, the speed boundary for classifying the moving objects is strongly related to this predefined threshold. The manuscript has been revised as below.

The original statement ‘in which the yellow boxes are the m by n pixel area that is first compressed to a single pixel and then processed by the same memristor.’ has been replaced as the following statement:

(Article) On page 6, lines 236-239: **The yellow box in Figure 4c is the example m by n pixel area that is first compressed to a single pixel through aggregating the analog voltage outputs from the $m \times n$ pixel region, and the aggregated visual information of this compressed pixel is processed by a memristor in the 25×40 memristor array.**

Besides, the original statement ‘When the yellow box expands to the whole image, global attributes are gathered’ has been replaced with the following statement:

(Article) On page 6, lines 259-260: **When the above processing method is applied to the whole image area, global attributes are gathered through the memristor array, and reflected in terms of the memristor array resistance.**

The original statement ‘The gray CMOS image from the recorder is compressed, by averaging a matrix of $m \times n$ into one pixel, based on image spatial redundancy theory...which are then encoded into electrical pulses to modulate the memristor into low- and high-resistance states’ has been replaced with the following statement:

(Article) On page 5, lines 212-221: During the processing procedure shown in Figure 4a, the gray CMOS image from the recorder is compressed by averaging a matrix of $m \times n$ into one pixel (in this set parameters, m and n equals 48 and 36, respectively) based on image spatial redundancy theory. Next, filtering circuits for pre-process extracts the visual information changes ΔL within two adjacent frames. Then, the computing function select module, based on the analog computing circuit and the control switch, divides the light intensity changes into fast and slow categories based on a comparison to a pre-set threshold, resulting in the different encoding functions E_i . Specifically, if the change exceeds this threshold, the visual stimuli in this pixel are classified as fast; otherwise, it is considered slow. Subsequently, the fast and slow visual attributes are encoded into electrical pulses V_f and V_s to modulate the memristor into low- and high-resistance states, respectively.

The following statement has also been added to explain the state interpretation and speed boundaries:

(Article) On page 6, lines 224-228: In scenarios involving moving objects, the light intensity changes correlate to the relative moving speed of an object to the observing car. Therefore, the predefined threshold effectively establishes the speed boundary for categorizing the movement of an object as either relatively fast or slow, further modulating the memristor state differently.

Accordingly, the revised Figure 4 is detailed as below:

Figure R2 (The revised Figure 4). Utilizing the differential processing method to process and learn visual information. (a) The diagram of visual information differential processing. It is important to note that the hardware components, indicated by symbols such as ①, ②, etc., correspond to the stages of this processing pipeline. (b) The experimental design. (c) The corresponding real-world scenes of the example point visual information processing. (d) The visual information processing within the yellow box in Figure 4c, including the change in light intensity (scaled from 0 to 2.56), the differential voltage, and the memristor state changes (1 represents high resistance state; 0 represents low resistance state). (e) Sudden movement of pedestrian at night. (f) Sudden movement of pedestrian during the day. (g) Sudden movement of pedestrian hiding behind a car, with background vehicles moving slowly. (h) The amplitude spectrum comparison between the compressed image and the differentiated image at the last moment in Figure 4e. (i) The amplitude spectrum comparison between the compressed image

and the differentiated image at the last moment in Figure 4f. (j) The amplitude spectrum comparison between the compressed image and the differentiated image at the last moment in Figure 4g.

Besides, we have added a detailed description for Figure 4 as **Supplementary Discussion 11**, covering the yellow box representation, methods of identification, preprocessing steps, interpretation of states (e.g., distance), and speed boundaries.

(Supplementary Discussion 11):

Yellow box: This represents an example case of visual information processing, consisting of a region spanning $m \times n$ pixels within the original image. After being compressed, this region transforms into a single pixel point. Subsequently, this compressed point is subject to processing via a memristor, employing a one-to-one approach.

Preprocessing steps: The process involves utilizing filter circuits to extract the change in light intensity ΔL , within the compressed pixel point.

Methods of identification: The light intensity change ΔL within the compressed pixel point serves as the criterion for classification. Should ΔL surpass the predefined threshold L_{th} , the visual information in this point is categorized as fast. Otherwise, it is classified as slow. This process is implemented by a voltage comparison circuit.

Interpretation of states: For a compressed point categorized as ‘fast,’ the modulation voltage for the memristor is formulated as:

$$V_f = E_f(\Delta L), \Delta L > L_{th} \quad (1)$$

Otherwise, for a point considered ‘slow,’ the voltage is expressed as:

$$V_s = E_s(\Delta L), \Delta L \leq L_{th} \quad (2)$$

Speed boundaries: The demarcation of states is reliant on the threshold L_{th} , which establishes the speed boundary for categorizing the movement of an object as either relatively fast or slow.

Comment #13

Discussions Section: The Discussions section primarily serves as a summary of conclusions. It would benefit from expanding on the results and potentially incorporating insights from the supplementary material.

Response:

Thanks for your valuable feedback regarding the need for improving the Discussion section. The revised Discussion section is detailed below:

(Article) On page 7, lines 294-341:

In this study, we propose a memristive perception method harnessing both the synapse-like characteristics of memristors and a bio-inspired workflow design for intelligent machines to understand unstructured environments with desired good adaptation and generality. This innovative approach utilizes key insights derived from biological analogies to enhance neuromorphic computing and surpasses existing intelligent machine technologies in versatility and adaptability.

Key Insights from Biological Analogies

The nuanced sensing in biological perception benefits from the synergy of various receptors and sensory neurons—for instance, various types of tactile mechanoreceptors and their corresponding sensory neurons enable biology to sense and process a broad spectrum of pressure stimuli, each with distinct characteristics. In the memristive implementation, the above complex synergy process is folded into two computational phases: feature extraction and encoding sensory stimuli and differential processing based on memristors. Specifically, our method employs the nonlinear state space of the memristor alongside an adaptive memristive modulation scheme that adjusts its electrical characteristics based on specific features of sensory stimuli. This design enables a memristor to acquire a multifaceted understanding of the surroundings, exploiting the full computational potential of memristors. The memristor state serves as a nexus for integrating historical stimulus data and current sensory data; this state not only adjusts to evolving environmental conditions to provide neuromorphic style adaptation but also offers valuable references in decision-making. Through mimicking the fundamental perception mechanisms, such a memristive method surpasses current neuromorphic technologies in both adaptability and generalizability. It achieves seamless integration with dynamic environmental conditions and across various sensory modalities, and the detailed comparison can be found in Supplementary Table 3.

Comparison with Conventional Technologies

Differential neuromorphic computing, as a perception method, holds the potential to enhance subsequent decision-making and control processes. Compared with conventional technologies, both the PID control approach and the proposed differential neuromorphic computing share a fundamental principle of smartly adjusting outputs in response to feedback, they diverge significantly in the data manipulation process (Supplementary Discussion 14 and Figure S29);

our method leverages the nonlinear characteristics of the memristor and a dynamic selection scheme to execute more complex data manipulation than linear coefficient-based error correction in PID. Additionally, the intrinsic memory function of memristors in our system enables real-time adaptation to changing environments. This represents a significant advantage compared to the static parameter configuration of PID systems. In addition, the synapse-like characteristics of memristors significantly enhance computational efficiency in achieving human-like perception capabilities, which is evidenced by the fact that our hardware tactile system shows a 44% improvement over the equivalent simulation system (Figure S30). Such enhancement underscores the benefits of applying our method in real-time perception scenarios, enabling intelligent machines to respond and adapt to unstructured environments in time.

In conclusion, this method represents a significant advancement toward achieving fully autonomous and intelligent robot systems capable of effective perception in real-world scenarios. Due to the small size of memristive devices, the possibility of having high-density memristors over large areas and flexible substrates, and their similarity with the fundamental biological perception mechanisms, the presented method could enable intelligent machines to possess sensory capabilities on a scale comparable to humans when combined with diverse sensors (Supplementary Discussion 9), allowing them to sense and adapt to the environment efficiently (as depicted in Figure S22 and S23).

Comment #14

References: The reference list could be improved by:

- **Placing the need for robotics at the beginning, rather than neuromorphics.**
- **Citing another relevant Nature Communications paper on neuromorphic and autonomous robots (<https://www.nature.com/articles/s41467-022-28487-2>).**
- **Remembering to cite CMOS papers, as neuromorphics encompasses more than memristors.**

Response:

Thank you for pointing out the need for adjusting the reference list. The added references are detailed as below:

Relevant Nature Communications paper on neuromorphic and autonomous robots:

1. Bartolozzi, Chiara, Giacomo Indiveri, and Elisa Donati. "Embodied neuromorphic intelligence." *Nature communications* 13.1 (2022): 1024.

The need for robotics:

2. George, Jacob A., et al. "Biomimetic sensory feedback through peripheral nerve stimulation improves dexterous use of a bionic hand." *Science Robotics* 4.32 (2019): eaax2352.
3. Shih, Benjamin, et al. "Electronic skins and machine learning for intelligent soft robots." *Science Robotics* 5.41 (2020): eaaz9239.

CMOS papers:

1. Merolla, Paul A., et al. "A million spiking-neuron integrated circuit with a scalable communication network and interface." *Science* 345.6197 (2014): 668-673.
2. Davies, Mike, et al. "Loihi: A neuromorphic manycore processor with on-chip learning." *Ieee Micro* 38.1 (2018): 82-99.
3. Benjamin, Ben Varkey, et al. "Neurogrid: A mixed-analog-digital multichip system for large-scale neural simulations." *Proceedings of the IEEE* 102.5 (2014): 699-716.
4. Pehle, Christian, et al. "The BrainScaleS-2 accelerated neuromorphic system with hybrid plasticity." *Frontiers in Neuroscience* 16 (2022): 795876.

Reviewer #2

The authors present a somewhat "out of the ordinary" use of memristor technology as a sort-of PID controller, where they exploit bifurcations of the non-linear operation regimes of the memristor to facilitate fine-grained adaptive sensing capabilities in a way that draws parallels to how perception works in biology. They use two use-cases to demonstrate the enabled capabilities with detailed and comprehensive evaluation.

Overall I welcome their contribution as stimulating, relevant, and interesting, that deserves attention from the community.

The manuscript as it stands together with the supplementary materials are reach in content to assist the understanding of the contribution, but I think it needs a re-positioning and re-organisation (and occasionally some introductory claims need to be toned down). On reading the main manuscript i had initially difficulty to grasp the context and decode the approach or connect some of the main claims to the presented results. It all became much clearer upon reading the supplementary materials. However i am of the opinion that the supplementary materials are there to explain the details, not to decode the key ideas of the paper and make them understandable! Therefore my comments, critique, and suggestions, aim at making the main manuscript more informative and self-contained and the supplementary materials ... well supplementary of details.

Response:

We are very grateful to the reviewer for his/her recognition of our work and offering us critical insights to improve the quality of this paper, especially for the suggestion on emphasizing the novelty of exploiting nonlinear state-space of memristors for providing neuromorphic style adaptation. Our detailed responses to your valuable comments are provided below.

Comment #1

Some general remarks to be considered throughout the paper regarding general positioning and easing contextualization are the following:

- The way you discuss and defend "learning" in this work is a bit unusual or misleading to me, I suggest "(online) adaptation" or "short-term plasticity", which are more short-term (duration of the task), are more appropriate and less confusing in the context of the paper.

- Similarly the term "differential computing" to describe dynamic function selection (or a multi-branch functions) is also confusing/misleading. It becomes even more confusing when you also use the term "differential encoding" (which i believe is indeed correctly used in this case).

My suggestion is to either replace them .. or define their meaning explicitly upfront in the text.

Response:

Thank you for helping us remove misleading expressions. In the revised manuscript, the term ‘learning’ is deleted or replaced by the term ‘online adaptation.’ In terms of the use of ‘differential computing,’ we added a short definition part to clearly explain the term and the relationship between “differential computing” and “differential encoding” to avoid any potential confusion. The manuscript has been revised as below (Red represents modified or newly added text):

(Article) On page 1, lines 38-39: Understanding ~~and learning~~ sensory data efficiently to achieve human-like perception of the real world is pivotal for next-generation intelligent machines.

(Article) On page 2, lines 44-46: However, recent advancements in life sciences have revealed that the most crucial mechanism employed by humans for understanding unstructured environments is sensory information ~~differentiation and learning~~ differential processing mechanisms.

(Article) On page 2, lines 57-59: To this end, herein we present a memristor-based differential neuromorphic computing, perceptual signal processing and ~~learning~~ online adaptation method for intelligent machines.

Besides, we have replaced other instances of ‘learning’ with ‘online adaptation’ throughout the manuscript and SI for consistency.

Regarding ‘differential computing’, this term is defined explicitly in the revised manuscript:

(Article) On page 2, lines 59-62: ~~Specifically, differential neuromorphic computing is a data manipulation method involving multi-branch functions that emulate biological sensory processing, where sensory stimuli are selectively perceived by receptors and undergo different calculations supported by diverse neurons.~~

(Article) On page 2, lines 76-81: Therefore, drawing inspiration from the sensory information processing model and taking advantage of memristors’ intrinsic multistate property, the proposed ~~differential neuromorphic computing involves extracting features from unstructured data and applying memristive modulation schemes~~ applying an adaptive memristive modulation scheme called differential encoding to enhance the adaptability of intelligent machines for operation in unstructured environments.

Comment #2

B.1. Title:

The title is a bit too abstract/vague. Some keywords missing that would couple it to the subject matter of the paper is "memristor", "adaptive sensing", "(neuromorphic) perception", or similar

E.g. "Memristor-based adaptive neuromorphic sensing in unstructured environments" or something of that sort that suits the authors liking.

Response:

We appreciate your feedback regarding the need for a more descriptive and specific title. We have revised the title to better reflect the focus and scope of our work as '**Memristor-Based Adaptive Neuromorphic Perception in Unstructured Environments**'.

Comment #3

B.2 Abstract:

The abstract needs re-writing altogether: What is the challenge, what is the contribution towards addressing it, how did you test it, what you achieved.

> "Efficient operation of intelligent machines in the real world requires methods that allow them to understand and predict the uncertainties presented by the unstructured environments ..."

While there is nothing wrong with this sentence, more relevant to your work is "...to understand and adapt to conditions of unstructured environments.." or "...evaluate the uncertainties presented by the unstructured environments.."

> "Current methods rely on pretrained networks .. suffer .. limited generalization capabilities".

... in unseen or unstructured environments. I believe overall large pretrained networks are rather good at generalisation unless trained with little data or the data distribution is very dynamic (which is the scope of the use-cases in the paper). My point is you should focus on perception in unstructured environments.

> "we present a memristor-based differential neuromorphic computing, perceptual signal processing and learning method for intelligent machines"

My issue here is the use of terms "differential computing" and "learning" as explained in the general remarks.

> "The main features of environmental information such as amplification (>720%) and adaptation (<50%) of mechanical stimuli encoded in memristors, are extracted to obtain human-like processing in unstructured environments"

Very cryptic sentence without prior context at this point. I suggest removing it.

> "takes advantage of the intrinsic multi-state property of memristors and exhibits good scalability and generalization, as confirmed by validation in two different application scenarios"

scalability and generalization in what sense? IMHO you showed fine-grained/good adaptation and generality (for different sensing modalities).

> "... through fast learning (in ~1 ms) "

"adaptation" or the more biologically relevant "short-term plasticity", is more preferred term

> "... adapting to diverse sensing technologies"

You probably mean different sensing modalities (vision, tactile sensing)?

Response:

We appreciate your feedback regarding the need for a more structured and clearer abstract, the revised abstract is detailed as below:

Efficient operation of intelligent machines in the real world requires perception methods that allow them to understand and adapt to unstructured environments with good accuracy, adaptation, and generality, similar to humans. Towards this need, we present a memristor-based differential neuromorphic computing, a perceptual signal processing and online adaptation

method that leverages the nonlinear state-space of memristors to provide neuromorphic style adaptation to external sensory stimuli. The adaptation and generality of this method are confirmed by validation in two different application scenarios: object grasping and autonomous driving. In the former, a robot hand experimentally realizes safe and stable grasping through fast adaptation (in ~ 1 ms) of the unknown object features (e.g., sharp corner and smooth surface) with a single memristor. In the latter, the decision-making information of 10 unstructured environments in autonomous driving (e.g., overtaking cars, pedestrians) is accurately (94%) extracted with a 40×25 memristor array. By mimicking the intrinsic nature of human low-level perception mechanisms, the electronic memristive neuromorphic circuit-based method presented here shows the potential for adapting to diverse sensing modalities and helping intelligent machines generate smart high-level decisions in the real world.

Comment #4.1

Introduction

You need to improve the general positioning, by putting focus on adaptation or adaptive-control in sensing (for feature extraction), not quoting learning because learning implies more longer-term plasticity/adaptation.

Response:

As shown in Response for Comment#1, we have put focus on online adaptation instead of learning.

Comment #4.2

Somewhere you need to clearly mention that you exploit the bifurcations of the non-linear state-space of the memristor for providing neuromorphic style sensing/adaptation. (Analogous to and possibly better than PID control?)

Response:

We are immensely grateful for your insightful and accurate summary of the core principles of our approaches. The revised manuscript is detailed below:

(Article) On page 2, lines 76-85: Therefore, drawing inspiration from the sensory information processing model and taking advantage of memristors' intrinsic multistate property, the proposed differential neuromorphic computing method involves extracting features from unstructured data and applying an adaptive memristive modulation scheme called differential encoding to enhance the adaptability of intelligent machines for operation in unstructured environments (Supplementary Discussion 4 to 6). This approach **exploits the bifurcations of the nonlinear state-space of the memristor for providing neuromorphic style sensory adaptation to environmental stimuli. As a result, this method** experimentally facilitates the processing and learning of unstructured data within two complex environments, i.e., grasping of objects and autonomous driving.

Comment #4.3

The last sentence of S1 is more informative than any affirmation in the intro about the work presented and should therefore be brought in the intro: "We aspire to equip these intelligent machines with a comprehensive perceptual capability akin to that of living organisms, enabling them to better comprehend environmental information."

Response:

Thanks for your valuable comments for a better presentation of our insights. The following statement is added to the manuscript:

(Article) On page 3, lines 88-95: In the latter, we differentiate visual information by a 40×25 memristor matrix via SPICE simulations and reach a commendable accuracy of 94% in extracting critical information within 10 autonomous driving scenes, after comparison with human labelled results. These two experiments showcase that the proposed method has the

capability to handle a different number of memristors. **Moreover, this method shows the potential to equip these intelligent machines with a comprehensive perceptual capability akin to that of living organisms, enabling them to better comprehend environmental information.**

Comment #4.4

**> "the acquisition of diverse types of physical information"
you mean "integration" ?**

Response:

Thanks for pointing out the inappropriate word use, this statement has been revised as:

(Article) On page 2, lines 42-43: Traditionally, the adeptness of organisms within unstructured environments has been attributed to the **integration** of diverse types of physical information.

Comment #4.5

**> "these receptors and neurons can rapidly form a complex set of intricate network-based perception functions, such as integration and firing, potentiation and depotentiation, associative memory, environmental mapping, and motion control tasks."
Integration firing, (de)potentiation/learning, are not "perception"-level functions rather intricate/basic nnet operations at a lower-level than perception.**

Response:

We highly agree with your understanding of perception, and the revised statement is:

(Article) On page 2, lines 50-52: these receptors and neurons can rapidly form a complex set of intricate network-based perception functions, **such as environmental mapping, motion control tasks, associative memory, etc.**

Comment #4.6

"In contrast, conventional intelligent machines operate within the von Neumann separated storage and computation architecture, which makes it inefficient to differentially process highly dynamic unstructured information (9–13)"

This sentence as an antithesis is irrelevant. Whether the compute substrate is Von-Neumann or not (inefficient or not) an emulated artificial nnet implements the same algorithm and the task performance should be (approximately) the same.

Response:

Thanks for your valuable comments. To address this issue, this sentence has been removed.

Comment #4.7

"This can be resolved with memristors, as they integrate storage and computation capabilities, coupled with synapse-like characteristics, ... "

Such as ? (what synapse like characteristics)

Response:

To clarify the synapse-like characteristics, the original statement ‘This can be resolved with memristors, as they integrate storage and computation capabilities, coupled with synapse-like characteristics, and thus bear a striking resemblance to the biological synapse at a foundational level. This similarity opens the possibility for the realization of human-like perception functionalities in intelligent machines’ has been replaced with the following statement:

(Article) On page 2, lines 52-57: **Memristors, a kind of neuromorphic device, not only integrate storage and computation capabilities but also have the property of changing their information transmission efficiency. These capabilities, often referred to as synapse-like characteristics (14–19), underscore the remarkable similarity between memristors and synapses, i.e., the basic computational unit in biology. Such similarities make memristors the ideal fundamental device for realizing human-like perception functionalities in intelligent machines.**

Comment #4.8

> **".. perceptual signal processing and learning method for intelligent machines."
Scaling of the weights is not automatically implying "learning ability" (long-term plasticity), however it is adaptation. I understand why you defend it as online learning (after reading the supp material), but as explained in the general comments I think it is misleading/overstatement.**

Response:

Thanks for your feedback regarding a clear and appropriate statement, and the term ‘learning’ has been replaced with ‘online adaptation’.

(Article) On page 2, lines 57-59: To this end, herein we present a memristor-based differential neuromorphic computing, perceptual signal processing and **online adaptation** method for intelligent machines.

Comment #4.9

> **"encompasses multidimensional features"
multidimensional and multimodal?**

Response:

Thanks for your insightful suggestions, we have revised the content as below:

(Article) On page 2, line 63: Unstructured information encompasses **multidimensional and multimodal** features,

Comment #4.10

"on keeping a memristor to a fixed receptor (Figure 1a), e.g. a nociceptor (22–25), thus restricting their full potential"

A bit more elaboration please for context. Particularly to explain how the use of the memristor in 23,23 differ from the use here? Is it the different M(.) used/selected by the

digital part to modulate different regions of operation of the memristor? (based on the explanations of 22,23 in S2).

Response:

Thanks for pointing out the need for more elaboration in this context. Recent advancements have been added for comparison in the aspect of the memristor use and the system function.

Figure R3. Comparison between other works. (a) Schematic diagram of the typical nociceptor nervous system. (b) The circuit configuration of a memristor-based artificial nociceptor. Note that the DUT (device under test) represents the memristor. (c) Input voltages (V_{Ch1}) produced by the pulse generator, and output voltages (V_{Ch2}) applied to the p-FET (upper panel), and currents (I_{SPA}) measured by the semiconductor parameter analyzer (lower panel). a-c reproduced with permission⁹. (d) Schematic diagram of an artificial thermal nociceptor consisting of a thermoelectric module and the diffusive memristor. (e) The generated voltage from the thermoelectric module and the ON-switching and OFF-switching of the threshold switch monitored by Ch1 and Ch2 of the oscilloscope, respectively. d-e reproduced with permission⁸. (f) Schematic of the artificial mechanoreceptor system. In this system, a piezoelectric device acts as the tactile sensor for receiving external sensory information and is connected to the artificial afferent nerve. (g) The relationship between the input voltage and the frequency of response voltage of the artificial afferent nerve. (h) The frequency response of the system under external pressure. f-h reproduced with permission¹⁰.

Figure R4. Schematic diagrams illustrating neuromorphic computing approaches in sensory processing. (a) The schematic of current methods. (b) The schematic of the differential neuromorphic computing method, distinguishing our approach from existing techniques.

As depicted in Figure R3, previous methodologies, such as the memristor-based nociceptors depicted in Figures R3 (a) and (d), are designed to process dangerous stimuli exclusively. This kind of design restricts the capability of the memristor to process stimuli of other nature, e.g., mild ones. It is also noted in Figure R3 (h) that the output frequency response shifts only when the input voltage surpasses a certain threshold. However, given the high similarity between the memristor and biological synapse, the memristor has the capability to replicate diverse synaptic plasticity, thus effectively emulating the intrinsic characteristics of different receptors and sensory neurons. Hence, we leverage this capability in our work by employing the memristor as a **dynamic computing synapse**. As shown in Figure R4, **the modulation scheme is selected smartly, enabling the memristor processing and adapting to varying external sensory stimuli**. The manuscript is revised as:

(Article) On page 2, lines 66-76: However, existing methods focus on keeping a memristor to a fixed receptor (Figure 1a), e.g., **the memristor-based pressure nociceptor only processes pressure inputs above a pre-set threshold. This kind of design omits useful information in pressure amplitudes below this threshold, failing to utilize the full spectrum of pressure data for understanding dynamic inputs in a manner akin to human skin. A potential solution is to operate multiple memristors to process pressure data across a broader spectrum. However, given the high similarity between the memristor and biological synapse, the memristor has the capability to replicate diverse synaptic plasticity, thus effectively emulating the intrinsic characteristics of different receptors and sensory neurons.** Furthermore, distinct features of sensory data exhibit time domain independence, implying that a memristor can adaptively switch to an appropriate state to process each feature at a time slot.

In our method, $M(.)$ is the relationship between the memristor state and the feature used for selecting modulation schemes. In the tactile experiments, $M(.)$ **remains constant**, and $M(x)$ represents the memristor resistance. To clarify this, we have added the following discussion in the **Supplementary Discussion 7**.

(SI) On page 14: In the perception of external tactile information, we utilize the current pressure magnitude as the stimulus input, considering the memristor as a synapse and $M(\cdot)$ defining the correlation between the memristor state and the feature employed to determine the appropriate modulation schemes. The function $V(\cdot)$ for memristor modulation acts as a dynamic, multi-branch function based on current sensory features $f(t)$ and memristor resistance $M(x)$, and this modulation is selected by the FPGA digital part.

Further, Figures R3 and R4 are added to the Supplementary Information as **Figures S25** and **S26**, respectively. The above discussion is also added as **Supplementary Discussion 12**.

Comment #5.1

S7/8 entry sentences need to be brought in the main text too to contextualize the use-cases either in the methods or here in the results sections.

Response:

Thanks for pointing out the need for the expansion of the manuscript and the added manuscript is detailed below:

(Article) On page 3, lines 104-108: Realizing safe and stable manipulation of unknown objects by robotic hands is tricky but highly needed (27–31). Unlike lab and factory settings, an unknown object may exhibit sharp or slippery attributes. The former potentially damages the contacting end, and the latter places a heavy burden on the sensing and computing modules when keeping the object balanced between stabilization and deformation. To address this issue, the proposed memristor-based differential neuromorphic computing method is embedded into the sensing and controlling strategy of a robotic hand to **achieve intricate and nuanced tactile perception**, as depicted in Figure 2a. **Specifically, this method is utilized to emulate multiple essential tactile receptors and sensory neurons in tactile stimuli perception, including nociceptors, fast-adapting, and slow-adapting receptors, along with their respective neural pathways.**

(Article) On page 5, lines 198-210: In previous tactile information processing, attributes based on force strength were used to understand environmental knowledge. In contrast, visual frequency-based attributes are more important for autonomous driving, as they imply the relative position change of surrounding objects to the car (45–47) and are vital for real-time decision making (48–50). For instance, the sudden appearance of pedestrians or vehicles can lead to life-threatening collisions. Thus, shortening the attribute extraction time of moving objects is important. In addition, the moving direction of the object can further help to make good decisions (51, 52). To this end, the proposed differential neuromorphic computing method is employed to **acquire and process visual frequency-based attributes. Drawing an analog to the frequency processing difference between cone and rod cells in biological vision perception-where cone cells excel in capturing rapidly changing visual stimuli-this method retrieves fast information akin to cone cells, and produces neural excitation to maintain fast information for a while (Supplementary Discussion 8).**

Comment #5.2

> "The first can be of the desired sensor .."

The first module can ..

> "and $p(t)$ is the outputted electrical signal .."

output electrical signal

Response:

Thanks for your careful review. The revised statement is detailed as below:

(Article) On page 10, lines 402-407: The first **module** can be of the desired sensor type according to mimicked biological sensory...and $p(t)$ is the **output** electrical signal...

Comment #5.3

> "The extraction and modulation scheme generation are both conducted in a field programmable gate array (FPGA) platform, as shown in Figure 2a"

I think the explanations are a bit fuzzy and ambiguous, they are much better explained in the supp material. When you say "modulation scheme generation", you mean function *selection* for the modulation scheme? Is a modulation scheme dynamically generated?

Response:

Thanks for pointing out the ambiguous expression. The modulation scheme is dynamically selected based on the current features of sensory stimuli and memristor state rather than generation. Thus, the manuscript is revised as:

(Article) On page 3, lines 116-117: The **sensory feature extraction and modulation scheme selection** are both conducted in a field programmable gate array (FPGA) platform, as shown in Figure 2a.

Comment #5.4

> "After the attribute of a force is extracted .."

Rather magnitude of the force? Is this not the criteria for the selection of the appropriate modulation scheme?

Response:

Apologize for the incorrect expression. We acknowledge that the magnitude of the force is criteria for the selection of the appropriate modulation scheme. The revised manuscript is detailed as below:

(Article) On page 4, line 132: After the **magnitude** of a force is **measured**, an associated modulation scheme is selected and then applied to the memristor.

Comment #5.5

> "The curve trends are similar to the biological response strengths in Figure 2g"
Fig 2g is not showing much, no curves anyway, or else at least explain it a bit?.

Response:

Thanks for pointing out the need for further explanation. Figure 2 and the original statement 'The curve trends are similar to the biological response strengths in Figure 2g.' are revised as:

Figure R5 (The revised Figure 2). Utilizing differential processing method to process tactile information and mimic multiple receptors. (a) Schematic diagram of the comprehensive pipeline designed to understand tactile sensory stimuli. This approach initially captures immediate tactile interactions through the resistance changes in a piezoresistive film, employing Analog-to-Digital Converters (ADC) and Digital-to-Analog Converters (DAC) and the required information exchange interface. Concurrently, it archives historical sensory data from the memristor state, leveraging the memory properties of a memristor. Subsequently, the approach extracts sensory features from both the current state of the piezoresistive film and the

memristor to select a memristive adaptive modulation scheme. The entire sensory data acquisition, feature extraction, and modulation scheme selection process are managed through the FPGA control platform. Note that the memristor is detected and modulated under the cooperation of the encoder and memristor-based process modules within the hardware setup. Additionally, a communication module is established for transferring information to external PC devices alongside a specifically designed interface for manipulator control. (b) The device structure of the self-directed memristor. (c) The U-I test of memristors. In this test, a 500mV peak-to-peak sine wave is applied to the memristor and the 10k Ω fixed-value resistor. (d) Electrical characterization of memristors using positive pulses. (e) Electrical characterization of memristors using negative pulses. (f) Amplification processing of hazardous stimuli. (g) Corresponding biological processing functions. (h) Adaptation processing of mild stimuli. (i) Sensitization processing of hazardous stimuli achieved through further differentiation. (j) Quantitative evaluation of the sensitization function. (k) Realization of adaptation at different speeds. (l) Comparison of adaptation speeds in Figure R5 (k), in which the derivative of the memristor resistance change is calculated.

(Article) On page 4, lines 139-144: The curve trends of response strength in Figure 2f and 2h are similar to the biological response strengths in Figure 2g. For biological nociceptors, the sensitivity to external sensory stimuli increases when exposed to dangerous stimuli, resulting in a continuously increasing response strength; in contrast, the adapting receptors adapt to external tactile stimuli by gradually reducing their sensitivity when the feature of stimuli is mild, resulting in a decreasing response strength.

Comment #5.6

> "Specifically, during the encoding process, we additionally consider the state of the memristor. When its state falls below a certain threshold (100 k Ω in this case), .."

Where is the switching happening? And how are the thresholds determined? Are they learned, or related to the environment or were they found based on hand-tuning for the specific memristor?

Response:

We appreciate your request for additional details. When the memristor resistance falls below a predefined threshold (100 k Ω in this case), the switch in the modulation scheme occurs, implemented by FPGA controlling (Figure R6). This threshold is programmed into the FPGA-based logic circuit, serving as a trigger for adjusting the modulation scheme. The selection of 100 k Ω in our experiments is the result of careful hand-tuning, taking into account the electrical characteristics of the memristor used as well as the modulation scheme. The revised manuscript is detailed as below:

(Article) On page 4, lines 150-155: When its state falls below a predefined threshold (100 k Ω in this case), the switch in the modulation scheme occurs. This threshold value 100 k Ω is pre-programmed into the FPGA-based logic circuit, serving as a trigger for modulation scheme adjustments. Consequently, there is an increase in both the amplitude and the pulse duty cycle

of the positive pulses (as shown in Figure 2i), achieving the functionality of sensitized processing for external stimulus information. Further details about the threshold value selection are provided in Supplementary Table 1.

(Supplementary Table 1) On page 52: The threshold selection in our modulation schemes is the result of careful hand-tuning based on the electrical characteristics of the memristor used as well as the modulation scheme. Taking the threshold 100 k Ω as an example, it is designed to recognize the condition when the system encounters dangerous stimuli for a while; the nociceptor enters into the sensitization state. In other words, the memristor has been modulated by the positive voltage pulses for a while. Thus, it is set below the memristor initial middle resistance value (~ 170 k Ω), and 100 k Ω effectively serves as an indicator of sustained exposure to dangerous stimuli.

Figure R6. The switch in modulation schemes achieved by FPGA controlling. When the sensory stimuli feature $f(t)$ is in a constant hazardous condition, the system selects the encoding strategy that adjusts the memristor to a low-resistance state. This strategy bifurcates into two distinct modulation schemes based on the memristor resistance $x(t)$: Scheme 1.1, which operates at a 13.3% duty cycle and a 0.30 V voltage amplitude, and Scheme 1.2, featuring a 20% duty cycle and a 0.45 V voltage amplitude. Scheme 1.1 is selected when the pair $(x(t), f(t))$ falls within Set S_1 , i.e., the memristor resistance exceeds the predefined threshold of 100 k Ω , and the feature is hazardous. If the memristor resistance at the next computing cycle is below this 100 k Ω threshold, indicating the pair $(x(t+1), f(t+1))$ belongs to Set S_2 , the scheme switches into scheme 1.2, achieved by FPGA controlling.

Additionally, Figure R6 is added to the **Supplementary Information** as **Figure S27**.

Comment #5.7

> "tactile stimuli is offered (Figure 2g), exceeding the state-of-the-art works .."
 You mean Fig 2f ?

Response:

Thank you for pointing out the incorrect figure number. The Figure 2g in this statement has been updated into Figure 2j.

Comment #5.8

> "Upon detecting a slip event onset, the robot prompts "
How is the slip detected ?

Response:

Slip events, triggered by external disturbances, are characterized by abrupt decreasing changes in the previously stable contact force. Traditional methods for extracting such features involve recording historical contact forces and analyzing them with current interaction forces^{11,12}. For example¹¹, slip detection can be accomplished by examining the frequency change Δf based on the historical contact force F_h and the current force F_c ; the frequency change can be determined using the following formula:

$$\Delta f = f(F_c) - f(F_h) \quad (3)$$

where f is a function mapping observing force points to the frequency information. In the memristive implementation, we have developed a **detection logic** only based on the **current memristor state** and **contact force change** to identify such slip events. This method leverages the memristor state as a cumulative record of the features of historical contact force F_h in differential neuromorphic computing to enhance detection efficiency. Specifically, when the memristor is in a high-resistance state ($>225 \text{ k}\Omega$) due to the modulation scheme during stable force contact and the piezoresistive film changes drastically, resembling a spike (increases above $350 \text{ k}\Omega$ from a last point below $250 \text{ k}\Omega$), a slip event is inferred to have occurred. This design allows for the direct interpretation of slip events from the single readout of the current memristor state and piezoresistive film state change without substantial data historical storage and analysis, significantly enhancing processing efficiency. Note that these threshold value selections result from careful hand-tuning based on the memristor's characteristics and the object's contact characteristics.

To clarify, the above discussion is added to Supplementary Information as Supplementary Discussion 13:

(SI) On page 20: Slip events, triggered by external disturbances, are characterized by abrupt decreasing changes in the previously stable contact force. Traditional methods for extracting such features involve recording historical contact forces and analyzing them with current interaction forces^{11,12}. For example¹¹, slip detection can be accomplished by examining the frequency change Δf based on the historical contact force F_h and the current force F_c ; the frequency change can be determined using the following formula:

$$\Delta f = f(F_c) - f(F_h)$$

where f is a function mapping observing force points to the frequency information. In the memristive implementation, we have developed a **detection logic** only based on the **current memristor state** and **contact force change** to identify such slip events. This method leverages the memristor state as a cumulative record of the features of historical contact force F_h in differential neuromorphic computing to enhance detection efficiency. Specifically, when the memristor is in a high-resistance state ($>225 \text{ k}\Omega$) due to the modulation scheme during stable force contact and the piezoresistive film changes drastically, resembling a spike (increases above $350 \text{ k}\Omega$ from a last point below $250 \text{ k}\Omega$), a slip event is inferred to have occurred. This design allows for the direct interpretation of slip events from the single readout of the current memristor state and piezoresistive film state change without substantial data historical storage and analysis, significantly enhancing processing efficiency. Note that these threshold value selections result from careful hand-tuning based on the memristor's characteristics and the object's contact characteristics.

Comment #5.9

In this paragraph it is not clear to me how/why the information of the memristor state is more useful or informative than the information from the differential encoding of the sensor data? Especially since the stimuli from APS camera encodes analog domain in the pixel intensities.

Also in this use-case it has not been clear to me, if your setup connects directly on the camera sensor array with memristors or if you process image data from the APS camera.

Figure R7. Comparison between the image captured by a standard car camera, the output from an event-based camera, and the outcomes achieved through the differential

neuromorphic computing approach. (a) Car camera image. (b) Differential processing in event-based camera. In event-based cameras, differential processing typically refers to the specialized circuit designed for detecting changes in light intensity, with the outcomes represented as D . (c) Our proposed differential neuromorphic computing. In our proposed differential neuromorphic computing, differential encoding mainly refers to the appropriate encoding scheme based on current sensory features. For our visual processing demonstration, we identify changes in the light intensity as the primary sensory feature, and the differential encoding pulses V are applied to the memristor array, leading to the memristor states in M .

Response:

Thanks for pointing out the need for further explanation. A comprehensive comparison between the differential processing utilized in neuromorphic sensors, such as event-based cameras, and the differential encoding approach in our method, alongside the processed results, is illustrated in Figure R7.

1. In event-based cameras, a specialized circuit is designed to perform differential processing, i.e., detect the light intensity change, denoted as ΔV . When F_t represents the static image at time t , the light intensity change ΔV can be discerned by calculating D . In the visualizing results, it is observed that the differences between two consecutive frames are highlighted. Similarly, the differential encoding V (similar to D) merely identifies changes between frames, and the information is noisy after compression. In contrast, the memristor array M not only distinctly **emphasizes rapid changes** but also **maintains historical change information**, resulting in a clear position of the running pedestrian and the afterimages that imply his moving direction. These capabilities provide cleaner and more actionable data for further high-level processing, such as deducing the direction of movement through afterimages, predicting future locations, and other decision-making processes crucial for navigating dynamic environments.

Accordingly, the manuscript is revised as below:

(Article) On page 6, lines 263-268: Figures 4e to 4g are 3 examples taken from daytime and evening, together with their corresponding differential neuromorphic computing results; the fast-running pedestrian is accurately captured, and afterimages are generated, implying the orientation. **Observing the whole image, the memristor state information, even in a single frame, not only distinctly emphasizes rapid changes but also maintains historical change information. These capabilities provide cleaner and more actionable data for further high-level processing, such as deducing the direction of movement through afterimages, predicting future locations, and other decision-making processes crucial for navigating dynamic environments.**

In addition, Figure R7 is added to the Supplementary Information as **Figure S28** alongside the above discussion.

2. Regarding the implementation, digital images captured by the advanced photo system (APS) camera are converted into analog voltage profiles before being fed into the memristor array for processing in the simulation. To clarify this, the following statement is added:

(Article) On page 9, lines 381-387: In visual information processing, images captured by a CMOS sensor in real driving scenarios are converted into analog voltage inputs for the visual differential processing system. In the specific implementation, videos captured by the CMOS camera are initially divided into frames and converted into greyscale. Subsequently, these pieces of time-discrete visual information are transformed into continuous voltage data. These voltage profiles are then processed by the memristor-based visual system through SPICE simulation, mimicking the data processing manner after direct integration with the CMOS sensor.

Comment #6

B.4 Discussion

Discussion is short and mostly superficial. I would use the space to bring in summarised form some of the content from the supp material to enlighten the biology analogy discussion. Specifically S5/6 are paramount for the understanding of the paper and hence need to be brought in the main text.

I believe a very welcomed input would be a comparison/discussion of relevance/differences and advantages/disadvantages to PID control!

Also since memristors are useful for computational efficiency, a relative computational cost indication of using the memristor as such versus implementing a model of the memristor (in the digital domain) to achieve similar functionality, would be very welcome.

> " ..allowing them to genuinely learn and efficiently comprehend the environment."

Well comprehension is far fetched probably, it requires reasoning abilities. And the analogy to "learning" as opposed to adaptation, has not been explained at all until this point. (Leaving out mention of "learning" altogether is best as suggested in the general comments).

Figure R8. Comparison between the differential neuromorphic computing and PID control.

Response:

We are grateful for your constructive feedback on enhancing the discussion section. In response, we have integrated key insights from Supplementary Discussions 5 and 6 to enlighten the biology analogy discussion. In addition, we have introduced a comparative analysis with PID control to highlight the distinctions, advantages, and potential applications of our method. Additionally, to address concerns regarding word choice, we have replaced 'comprehend' and 'learning' with more precise expressions like 'perceive' and 'adaptation.'

Key Insights from Biological Analogies

The nuanced sensing in biological perception benefits from the synergy of various receptors and sensory neurons—for instance, various types of tactile mechanoreceptors and their corresponding sensory neurons enable biology to sense and process a broad spectrum of pressure stimuli, each with distinct characteristics. In the memristive implementation, the above complex synergy process is folded into two computational phases: feature extraction and encoding sensory stimuli and differential processing based on memristors. Specifically, our method employs the nonlinear state space of the memristor alongside an adaptive memristive modulation scheme that adjusts its electrical characteristics based on specific features of sensory stimuli. This design enables a memristor to acquire a multifaceted understanding of the surroundings, exploiting the full computational potential of memristors. The memristor state serves as a nexus for integrating historical stimulus data and current sensory data; this state not only adjusts to evolving environmental conditions to provide neuromorphic style adaptation but also offers valuable references in decision-making. Through mimicking the fundamental perception mechanisms, such a memristive method surpasses current neuromorphic technologies in both adaptability and generalizability. It achieves seamless integration with dynamic environmental conditions and across various sensory modalities, and the detailed comparison can be found in Supplementary Table 3.

Comparison between our methods and PID control

As illustrated in Figure R8, while the differential neuromorphic computing method and PID control share a fundamental principle of smartly adjusting outputs in response to feedback, they diverge significantly in the data manipulation process. Specifically, PID systems focus on automatically applying accurate responsive correction $v_2(t)$ to a control function based on the error value $e(t)$, and the relationship between $v_2(t)$ and $e(t)$ can be expressed as:

$$v_2(t) = K_p e(t) + K_i \int_0^t e(\tau) d\tau + K_d \frac{de(t)}{dt} \quad (4)$$

where K_p , K_i , and K_d are all non-negative, denote the coefficients for the proportional, integral, and derivative terms, respectively. In contrast, our method focuses on automatically selecting an appropriate modulation scheme $v_1(t)$ for memristors to process sensory stimuli, and this process can be expressed as:

$$v_1(t) = V(f(t), M(x(t-1))) \quad (5)$$

where $f(t)$ is the extracted features from sensory input, $x(t-1)$ is the memristor state after the previous modulation, M is the eigenvalue used for the memristor weight update and associated with its state, and V is the function used for selecting the scheme. Note that, in our system, $M(x(t-1))$ represents the memristor resistance after previous modulation. Thus, comparing the two equations, it is observed that the presented method leverages the nonlinear characteristics of the memristor and a dynamic selection scheme to execute more complex data manipulation than linear coefficient-based error correction in PID. Additionally, the intrinsic memory function of memristors in our system enables real-time adaptation to changing environments. This represents a significant advantage compared to the static parameter configuration of PID systems.

Working performance comparison

The computational efficiency of using memristor versus digital simulations is discussed when replicating the processing functions of the hardware tactile system. As shown in Figure R9, one sensory sample point averages 1.8 ms in simulations, whereas the hardware tactile system completes a computing cycle in just 1.0 ms. This stark contrast in processing times underscores the significant efficiency advantage offered by employing memristors directly in hardware, proving the potential for robotics to achieve human-like scale perception.

Figure R9. Comparative analysis of the efficiency between the hardware tactile system and the equivalent simulation system incorporating memristor models. (a) The duration of a single computing cycle, encompassing the initial sensing of external stimuli to the final generation of control instructions. Note that the wait switch refers to the process in which the analog switch in circuits is managed to enable the reuse of ADC (Analog-to-Digital Converters) and DAC (Digital-to-Analog Converters) for the detection of the piezoresistive film or memristors states. Additionally, differential computing includes the selection of modulation schemes for memristors and the subsequent communication with external devices. (b) Schematic representation of the simulation system. This diagram presents the simulation system architecture designed to replicate the functionalities of the hardware tactile system. (c) Comparison of computing efficiency between the hardware tactile system and its simulation counterpart.

Based on the above responses, the enhanced discussion is detailed as below:

(Article) On page 7, lines 294-341:

In this study, we propose a memristive perception method harnessing both the synapse-like characteristics of memristors and a bio-inspired workflow design for intelligent machines to understand unstructured environments with desired good adaptation and generality. This innovative approach utilizes key insights derived from biological analogies to enhance neuromorphic computing and surpasses existing intelligent machine technologies in versatility and adaptability.

Key Insights from Biological Analogies

The nuanced sensing in biological perception benefits from the synergy of various receptors and sensory neurons—for instance, various types of tactile mechanoreceptors and their corresponding sensory neurons enable biology to sense and process a broad spectrum of pressure stimuli, each with distinct characteristics. In the memristive implementation, the above complex synergy process is folded into two computational phases: feature extraction and encoding sensory stimuli and differential processing based on memristors. Specifically, our method employs the nonlinear state space of the memristor alongside an adaptive memristive modulation scheme that adjusts its electrical characteristics based on specific features of sensory stimuli. This design enables a memristor to acquire a multifaceted understanding of the surroundings, exploiting the full computational potential of memristors. The memristor state serves as a nexus for integrating historical stimulus data and current sensory data; this state not only adjusts to evolving environmental conditions to provide neuromorphic style adaptation but also offers valuable references in decision-making. Through mimicking the fundamental perception mechanisms, such a memristive method surpasses current neuromorphic technologies in both adaptability and generalizability. It achieves seamless integration with dynamic environmental conditions and across various sensory modalities, and the detailed comparison can be found in Supplementary Table 3.

Comparison with Conventional Technologies

Differential neuromorphic computing, as a perception method, holds the potential to enhance subsequent decision-making and control processes. Compared with conventional technologies, both the PID control approach and the proposed differential neuromorphic computing share a fundamental principle of smartly adjusting outputs in response to feedback, they diverge significantly in the data manipulation process (Supplementary Discussion 14 and Figure S29); our method leverages the nonlinear characteristics of the memristor and a dynamic selection scheme to execute more complex data manipulation than linear coefficient-based error correction in PID. Additionally, the intrinsic memory function of memristors in our system enables real-time adaptation to changing environments. This represents a significant advantage compared to the static parameter configuration of PID systems. In addition, the synapse-like characteristics of memristors significantly enhance computational efficiency in achieving human-like perception capabilities, which is evidenced by the fact that our hardware tactile system shows a 44% improvement over the equivalent simulation system (Figure S30). Such enhancement underscores the benefits of applying our method in real-time perception scenarios, enabling intelligent machines to respond and adapt to unstructured environments in time.

In conclusion, this method represents a significant advancement toward achieving fully autonomous and intelligent robot systems capable of effective perception in real-world scenarios. Due to the small size of memristive devices, the possibility of having high-density memristors over large areas and flexible substrates, and their similarity with the fundamental biological perception mechanisms, the presented method could enable intelligent machines to possess sensory capabilities on a scale comparable to humans when combined with diverse sensors (Supplementary Discussion 9), allowing them to sense and adapt to the environment efficiently (as depicted in Figure S22 and S23).

Additionally, the detailed comparison of our methods with PID is added to the Supplementary Information as **Supplementary Discussion 14**, and Figures R8 and R9 are also added to the Supplementary Information as **Figures S29 and S30**, respectively.

Comment #7

B.5 Supp material

S4 needs proper elaboration and corrections. Eg.

> f_{xx} , f_{GG} , f_{xG}

are not explained

> "..., the function $h()$ is a linear operation or a multiplication and division operation; thus $M \leq 0$ and $M \geq 0$."

Not obvious at all this conclusion

> "the derivative with respect to y : $dy =$ "

the direrential of y maybe? or the temporal derivative dy/dt ?

> $h(s,G)$

not explained

Response:

We are grateful for your careful check. A revised proof is provided below:

Considering a single memristor system, where the external sensory input is denoted as s , the memristor conductance as G , and the system output as y , the relationship between input and output of this system can be formalized as:

$$y = h(s, G) \quad (6)$$

where h represents the mapping function from inputs s and G to output y . Notably, s , G and y are all time-dependent variables. Assuming the initial state of this system at t_0 is characterized by s_0 , G_0 and y_0 , respectively, and given that h is sufficiently smooth (i.e. possesses all necessary partial derivatives), the system output change Δy over time Δt can be determined by the sensory change Δs and memristive conductance change ΔG through a Taylor series expansion as follows:

$$\Delta y = \frac{\partial h}{\partial s} \Delta s + \frac{\partial h}{\partial G} \Delta G + \frac{1}{2!} \left(\frac{\partial^2 h}{\partial s^2} \Delta s^2 + 2 \frac{\partial^2 h}{\partial s \partial G} \Delta s \Delta G + \frac{\partial^2 h}{\partial G^2} \Delta G^2 \right) + \dots \quad (7)$$

here, $\frac{\partial h}{\partial s}$ and $\frac{\partial h}{\partial G}$ represent the first-order partial derivative of y with respect to s and G at

(s_0, G_0) , respectively, while $\frac{\partial^2 h}{\partial s^2}$, $\frac{\partial^2 h}{\partial s \partial G}$ and $\frac{\partial^2 h}{\partial G^2}$ denote the second-order partial derivative of y with respect to s and G at (s_0, G_0) . Subsequent terms include higher-order partial derivatives, each multiplied by the power of the difference from the corresponding variable and divided by the factorial of that order, forming the typical form of the Taylor series. Typically, the degree of the function h is usually less than or equal to 2. For instance, in neuromorphic computing, s is often translated to an applied modulation voltage using a linear transformation, with the memristor response current as the output. Thus, higher-order partial derivatives in the Eq. 7 are negligible, and the change in system output simplifies to:

$$\Delta y = \frac{\partial h}{\partial s} \Delta s + \frac{\partial h}{\partial G} \Delta G + \frac{1}{2!} \left(\frac{\partial^2 h}{\partial s^2} \Delta s^2 + 2 \frac{\partial^2 h}{\partial s \partial G} \Delta s \Delta G + \frac{\partial^2 h}{\partial G^2} \Delta G^2 \right) \quad (8)$$

Observing Eq. 8, the values of partial derivatives at (s_0, G_0) are determined and can be considered as constants. Moreover, when Δs is fixed, the system output change Δy , i.e., the processing function of this neuromorphic system, hinges on ΔG . Moreover, according to the memristor electrical characteristics, the memristor conductance change ΔG can be expressed as:

$$\Delta G = Z(\int I dt) = Z(\int m(s) dt) \quad (9)$$

where I is the current through the memristor; Z represents the relationship between the memristor conductance change and current, with function m representing the modulation scheme applied to the memristor based on the current sensory s . Z is usually determined according to the specific material mechanisms and can be deemed a fixed function. From Eq. 9, it is observed that the modulation scheme $m(\cdot)$ highly affects ΔG , therefore deciding the system performing function.

In our differential neuromorphic computing approach, we exploit the adaptability of the encoding function m , which varies with the nature of sensory stimuli, enabling neuromorphic systems to emulate the complex processing functions observed in biology.

Comment #8

(Remarks on code availability):

I have looked at the github repository, however there is no documentation or information on what is available where, or how to use the code, or what the code is to be used for?

For example what can one do with this code? Would reproducibility require models of the memristors, or bitfiles for FPGAs? Is the code meant to generate graphs from pre-collected data? What is in the dataset?

No readme files and no instructions in the sup material.

Response:

We are grateful for your careful check, and the GitHub repository has been completed with readme files and instructions. The repository link is github.com/RTCartist/Differential-Neuromorphic-Computing; in the current version, the repository provides the necessary codes in the tactile and visual experiments with additional configuration instructions and resources to reproduce our work.

Reviewer #3

The paper introduces the concept of differential neuromorphic computing with memristive and in-memory computing devices. The paper also claims a giant leap (line 39) that would link the presented results to human low-level perception mechanisms, which would then be able to "generate smart high-level decisions in the real world" (lines 41-42).

While I recognize the inspiration with human perception as drawn in Figure 1, I also recognize that these are very general claims. Such claims make the narrative's arguments unclear and unspecific until the reader further examines the results section. **Additionally, a clear comparison with the state of the art is currently missing. In fact, the results could be more comprehensively and quantitatively compared (clearly) to other implementations of touch or visual information processing with the many other neuromorphic and in-memory computing implementations.** The lack of more specific claims and the lack of comparisons with SOTA is a significant drawback of the current version of the paper. Also, it is never clear in the paper if the results come from simulations or actual hardware devices. Only when inspecting the 43 pages of supplementary material does the full system architecture become clear (fig. S4, S13). For clarity, I suggest bringing figures S4 and S13 or a simplified version of those into the main manuscript.

Response:

Thanks for offering us your valuable insights, especially in pointing out the shortages of the current version on both lacking a comparison with the state-of-the-art (SOTA) works and bringing content from SI to the main manuscript. To clearly demonstrate the advances yielded by the developed method, a detailed comparison with the SOTA works across various aspects is provided in the following paragraphs. Before the comparison, we would like to first edit and explain our claim with more explanations in the manuscript, in order to avoid the situation met by the reviewer that "Such claims make the narrative's arguments unclear and unspecific until the reader further examines the results section". (Red represents modified or newly added text):

Explain the claim of "differential computing"

(Article) On page 2, lines 59-62: To this end, herein we present a memristor-based differential neuromorphic computing, perceptual signal processing and online adaptation method for intelligent machines. **Specifically, differential neuromorphic computing is a data manipulation method involving multi-branch functions that emulate biological sensory processing, where sensory stimuli are selectively perceived by receptors and undergo different calculations supported by diverse neurons.**

Comparison with SOTA works

(Article) On page 2, lines 66-74: However, existing methods focus on keeping a memristor to a fixed receptor (Figure 1a), **e.g., the memristor-based pressure nociceptor only processes pressure inputs above a pre-set threshold. This kind of design omits useful information in**

pressure amplitudes below this threshold, failing to utilize the full spectrum of pressure data for understanding dynamic inputs in a manner akin to human skin. A potential solution is to operate multiple memristors to process pressure data across a broader spectrum. However, given the high similarity between the memristor and biological synapse, the memristor has the capability to replicate diverse synaptic plasticity, thus effectively emulating the intrinsic characteristics of different receptors and sensory neurons.

Please note that a comparison to SOTA works will be provided under the reply to Comment 2; thanks.

In addition, the figures in the manuscript are revised as detailed below for better expression:

Figure R10 (The revised Figure 2). Utilizing differential processing method to process tactile information and mimic multiple receptors. (a) Schematic diagram of the comprehensive pipeline designed to understand tactile sensory stimuli. This approach initially captures immediate tactile interactions through the resistance changes in a piezoresistive film, employing Analog-to-Digital Converters (ADC) and Digital-to-Analog Converters (DAC) and the required information exchange interface. Concurrently, it archives historical sensory data from the memristor state, leveraging the memory properties of a memristor. Subsequently, the approach extracts sensory features from both the current state of the piezoresistive film and the

memristor to select a memristive adaptive modulation scheme. The entire sensory data acquisition, feature extraction, and modulation scheme selection process are managed through the FPGA control platform. Note that the memristor is detected and modulated under the cooperation of the encoder and memristor-based process modules within the hardware setup. Additionally, a communication module is established for transferring information to external PC devices alongside a specifically designed interface for manipulator control. (b) The device structure of the self-directed memristor. (c) The U-I test of memristors. In this test, a 500mV peak-to-peak sine wave is applied to the memristor and the 10k Ω fixed-value resistor. (d) Electrical characterization of memristors using positive pulses. (e) Electrical characterization of memristors using negative pulses. (f) Amplification processing of hazardous stimuli. (g) Corresponding biological processing functions. (h) Adaptation processing of mild stimuli. (i) Sensitization processing of hazardous stimuli achieved through further differentiation. (j) Quantitative evaluation of the sensitization function. (k) Realization of adaptation at different speeds. (l) Comparison of adaptation speeds in Figure R10 (k), in which the derivative of the memristor resistance change is calculated.

In order to be consistent with the modified Figure 2, Figure S4 is also modified as below:

Figure R11 (The revised Figure S4). The main structure in tactile differential processing. The system is composed of several main components. The FPGA serves as the primary controller, with Intel's E4CE10F17C8 chosen as the FPGA main control chip. The external crystal oscillator operates at a frequency of 50 MHz. The front-end sensing module comprises a DAC, ADC, and piezoresistive film. **The ADC, with the memristor process circuit, forms the memristor neuromorphic processing module.** The DAC in the encoder module cooperates with the memristor-based process module to facilitate both the memristor resistance detection and the differential encoding generating. For communication, the CY7C68013A chip facilitates USB connectivity with the external host computer alongside CyUSB suites. This setup enables the transmission of current perception information and the state information of the memristor's resistance value. **In addition, the system features a communication interface designed to control the manipulator via the Modbus protocol, enhancing the capability of the robotic system to interact with external devices and environments.**

Figure R12 (The revised Figure 4). Utilizing the differential processing method to process and learn visual information. (a) The diagram of visual information differential processing. It is important to note that the hardware components, indicated by symbols such as ①, ②, etc., correspond to the stages of this processing pipeline. (b) The experimental design. (c) The corresponding real-world scenes of the example point visual information processing. (d) The visual information processing within the yellow box in Figure 4c, including the change in light intensity (scaled from 0 to 2.56), the differential voltage, and the memristor state changes (1 represents high resistance state; 0 represents low resistance state). (e) Sudden movement of pedestrian at night. (f) Sudden movement of pedestrian during the day. (g) Sudden movement of pedestrian hiding behind a car, with background vehicles moving slowly. (h) The amplitude spectrum comparison between the compressed image and the differentiated image at the last moment in Figure 4e. (i) The amplitude spectrum comparison between the compressed image

and the differentiated image at the last moment in Figure 4f. (j) The amplitude spectrum comparison between the compressed image and the differentiated image at the last moment in Figure 4g.

Comment #1

It is worth noting that the concept of differential processing is already present in many state-of-the-art neuromorphic sensors, such as silicon retinas (or dynamic vision sensors), silicon cochleas, and other touch sensors. However, these endeavors have not been mentioned in the current paper. In addition, many works in the literature showing various materials (e.g., electrochemical metallization materials, phase-change materials, ferroelectric materials, ionic/electronic hybrid materials) have been employed to fabricate in-sensing computing touch sensory processing, and these are also not mentioned in the current paper. Currently, the work provides little originality over the current state of the art, as it fails to compare with those.

Response:

Thanks for your valuable insights; we believe your question is crucial in making readers understand the novelty of our work. In our understanding, we think three points should be well explained and supplemented in the manuscript.

A clear explanation of the differential processing method proposed in this article and the differences with relevant SOTA neuromorphic sensors

In this manuscript, differential processing is a computing method involving multi-branch functions that emulate biological sensory processing. Specifically, in our method, the memristor will be **automatically modulated into various states** to handle different kinds of pressure or visual information adeptly. This methodology effectively utilizes **the nonlinear state-space of the memristor** to provide neuromorphic style adaptation to external sensory stimuli, as shown in Figure R13. This stands in contrast to state-of-the-art (SOTA) neuromorphic sensors, where ‘differential processing’ refers to the **detection of the changes** in sensory stimuli, as shown in Figure R14, which is a **sub-function** of our proposed method.

Taking the event-based sensor as an example, the differential processing concept refers to the detection of light intensity change rather than static light intensity across the image. In comparison, first, our method can process both static and dynamic signals. For instance, in tasks involving object grasping tasks, static harmful sensory signals are used to modulate the memristor into a nociceptor state, demonstrating the versatility of our approach. Second, when dealing with dynamic signals, we consider both the immediate sensory change and historical data stored using memristor states, which eventually results in the abstraction of the moving object and the hint of its moving direction through afterimages. The differences are further highlighted through comparative result images in Figure R14.

To clarify this point, the following explanation is added to the manuscript. Besides, the above detailed explanation is added to the Supplementary Information as **Supplementary Discussion 15**, and Figures R13 and R14 are added to the Supplementary Information as **Figures S26** and **S28**, respectively.

(Article) On page 2, lines 59-62: To this end, herein we present a memristor-based differential neuromorphic computing, a perceptual signal processing method for intelligent machines.

Specifically, differential neuromorphic computing is a data manipulation method involving multi-branch functions that emulate biological sensory processing, where sensory stimuli are selectively perceived by receptors and undergo different calculations supported by diverse neurons.

Figure R13. The schematic diagram of our proposed method to process sensory stimuli.

Figure R14. Comparison between the image captured by a standard car camera, the output from an event-based camera, and the outcomes achieved through the differential neuromorphic computing approach. (a) Car camera image. (b) Differential processing in event-based camera. In event-based cameras, differential processing typically refers to the specialized circuit designed for detecting changes in light intensity, with the outcomes represented as D . (c) Our proposed differential neuromorphic computing. In our proposed differential neuromorphic computing, differential encoding mainly refers to the appropriate encoding scheme based on current sensory features. For our visual processing demonstration, we identify changes in the light intensity as the primary sensory feature, and the differential encoding pulses V are applied to the memristor array, leading to the memristor states in M .

The material selection of the memristor

The innovation presented in our manuscript lies in the development of **a data processing methodology for general neuromorphic computing**. This approach leverages the bifurcation of memristor states alongside environmental sensory features to enable fine-grained adaptive sensing capabilities in a way that draws parallels to how perception works in biology. Notably, our method is **universally applicable across memristors**, regardless of their underlying switching mechanisms, making it a versatile solution for neuromorphic computing applications. To demonstrate this, we purchased a commercially available memristor in tactile sensing experiments and opted for a well-recognized simulation model for visual perception. Thus, emerging materials are not used in the memristor fabrication. To claim it, the following discussion is added to the manuscript method part:

(Article) On page 9, lines 392-396: **Differential neuromorphic computing method is universally applicable across memristors, regardless of their underlying switching mechanisms, making it a versatile solution for neuromorphic computing applications. This universal ability is evidenced by the material selection, i.e., a commercially available memristor in tactile sensing experiments and a well-recognized simulation model for visual perception.**

Comparison with SOTA works

A comprehensive comparison is provided in the response to **Comment #2**.

Comment #2

While the methodology is adequately executed, it needs more focus and distinctly addresses the paper's key contributions. A comparative analysis, perhaps in the form of a table, highlighting improvements over current SOTA technologies (e.g., latency, amplification) is necessary. The paper's abstract claims advancements in three areas:

1- Differential neuromorphic computing (for which novelty needs to be clarified and compared to a rich literature of sensors and devices in neuromorphic engineering)

2- Hardware implementation of sensory processing with memristor arrays (which seems to be a significant contribution to the current work and for which the novelty is high)

3- Bridging low-level sensory processing with high-level decision-making (which is not being demonstrated in this work and not compared with SOTA)

More clearly stated, the methodology does not address these three main aspects, which are claimed in the abstract, and does not show how the in-memory computing memristive approach benefits all of these aspects. Also, the current limitations of deep learning and neuromorphic sensory systems and how this paper goes beyond those (other than amplification and lower latency, which are also not accurate, for example, compared to silicon retinas) need to be clarified.

Response #2:

We appreciate your feedback for a more focused elucidation of our key contributions. To address this, we have prepared a comparative analysis table (Table R1) that outlines our advancements in relation to current state-of-the-art (SOTA) neuromorphic technologies. Specifically, this table compares our work in the following aspects:

Differential Neuromorphic Computing

Compared to existing neuromorphic technologies, our method stands out for its compatibility with multiple sensory modalities, the utilized neuromorphic device maturity, and its interoperability with conventional sensor and neuromorphic computing technologies.

Hardware Implementation with Memristor Arrays

In our demonstration, the memristive tactile system is realized through a hardware implementation, showcasing its efficiency as illustrated in Figure R15; the visual information processing is conducted through simulations. Thus, a direct comparison of hardware implementation in visual information processing is not feasible. Instead, we focus on examining the data sources, decision-making features, and application scenarios utilized in our visual processing experiments. This comparison highlights the practicality of our method, illustrating its alignment with real-world data and scenarios.

Bridging Sensory Processing with Decision-Making

In this aspect, we focus on low-level processing akin to mimicking biological tactile receptors and the subsequent generation of intermediate visual features crucial for decision-making. The effectiveness of these features in reducing time delays and enhancing decision accuracy is a key focus. For high-level decision-making, we provide insights into the movement control used in tactile experiments as illustrative examples.

Table R1. Comparative analysis of our methods with current neuromorphic technologies.

	This work	Lee et al. ¹³	Lee et al. ¹⁴	Zhang et al. ¹⁵	Zhou et al. ¹⁶	Liu et al. ⁴	Jayachandran et al. ¹⁷	John et al. ⁵	Yuan et al. ¹⁸	Huang et al. ¹⁹	Li et al. ²⁰
Sensory modalities	4 types	1 type	2 types	1 type	1 type	1 type	1 type	1 type	4 types	1 type	1 type
Neuromorphic device maturity	Commercial available	Lab fabrication	Not applicable	Lab fabrication	Lab fabrication	Lab fabrication	Lab fabrication	Lab fabrication	Lab fabrication	Lab fabrication	Not applicable
Interoperability	¹ ✓	✗	✓	✗	✗	✓	✗	✓	✓	✗	✗
	² ✓	✗	✓	✗	✗	✗	✗	✗	✗	✗	✗
Low-level processing functions	Nociception		Slow adaptation								Slow adaptation
	Slow adaptation	Proprioception		Not applicable	Not applicable	Nociception	Not applicable	Nociception	Spike encoding	Not applicable	
	Fast adaptation		Fast adaptation								Fast adaptation
High-level decision-making	Pain reflex	Voluntary motion	Slip detection	Not applicable	Not applicable	Pain reflex	Not applicable	Pain reflex	Not applicable	Not applicable	Slip detection
	Slip detection										
Data sources	Field-based data	Not applicable	Not applicable	Laboratory based data	Laboratory based data	Not applicable	Laboratory based data	Not applicable	Not applicable	Simulation	Not applicable
Features for decision-making	Relative motion										
	Brightness change	Not applicable	Not applicable	Motion target	Brightness change	Not applicable	Approaching target	Not applicable	Not applicable	Historical trajectory	Not applicable
	Historical trajectory										
Scenarios	Moving pedestrian										
	Road signs			Moving pedestrian			RC car approaching another car				
	Warning light	Not applicable	Not applicable		Waving hands	Not applicable		Not applicable	Not applicable	Moving light spots	Not applicable
	Overtaking			Moving remote control (RC) car			RC car approaching wall				
Taillight and five other scenarios											

Note: Interoperability¹ addresses the system’s capability to integrate and function with commercial standard sensors in place of the originally demonstrated one. Interoperability² examines the system’s flexibility to be adapted or replaced by different neuromorphic technologies.

The above discussion and Table R1 have been added to Supplementary Information as **Supplementary Table 3**, and Figure R15 is added to Supplementary Information as **Figure S30**.

Figure R15. Comparative analysis of the efficiency between the hardware tactile system and the equivalent simulation system incorporating memristor models. (a) The duration of a single computing cycle, encompassing the initial sensing of external stimuli to the final generation of control instructions. Note that the wait switch refers to the process in which the analog switch in circuits is managed to enable the reuse of ADC (Analog-to-Digital Converters) and DAC (Digital-to-Analog Converters) for the detection of the piezoresistive film or memristors states. Additionally, differential computing includes the selection of modulation schemes for memristors and the subsequent communication with external devices. (b) Schematic representation of the simulation system. This diagram presents the simulation system architecture designed to replicate the functionalities of the hardware tactile system. (c) Comparison of computing efficiency between the hardware tactile system and its simulation counterpart.

Comment #3

Impact:

The impact of the work, in its current status, is minimal, as claims are too general, the state-of-the-art comparison needs to be included, and the novelty of differential neuromorphic computing needs to be clarified.

Response:

Thanks for pointing out the need to explain the impact of our work more clearly. I recognize the original claims are too general, as the reviewer commented. Therefore, we narrow down our claims to make the novelty focus on perception. In addition, we have rewritten our abstract and made adequate changes in the overall manuscript.

Comment #4

The code is minimal, the readme does not include:

1- configuration instruction

2- license

3- dependencies

4- required software and version

Response:

We are grateful for your careful check, and the GitHub repository has been completed with the necessary codes, readme files, instructions, license, dependencies, and required software and version. The repository link is github.com/RTCartist/Differential-Neuromorphic-Computing; in the current version, the repository provides the necessary codes in the tactile and visual experiments with additional configuration instructions and resources to reproduce our work.

References:

1. Basbaum, A. I., Bautista, D. M., Scherrer, G. & Julius, D. Cellular and molecular mechanisms of pain. *Cell* **139**, 267–284 (2009).
2. Dahiya, R. S., Metta, G., Valle, M. & Sandini, G. Tactile sensing—from humans to humanoids. *IEEE Transactions on Robotics* **26**, 1–20 (2010).
3. Osborn, L. E. *et al.* Prosthesis with neuromorphic multilayered e-dermis perceives touch and pain. *Science Robotics* **3**, eaat3818 (2018).
4. Liu, F. *et al.* Printed synaptic transistor-based electronic skin for robots to feel and learn. *Science Robotics* **7**, eabl7286 (2022).
5. John, R. A. *et al.* Self healable neuromorphic memtransistor elements for decentralized sensory signal processing in robotics. *Nat Commun* **11**, 4030 (2020).
6. Wark, B., Lundstrom, B. N. & Fairhall, A. Sensory adaptation. *Current Opinion in Neurobiology* **17**, 423–429 (2007).
7. Ge, J., Zhang, S., Liu, Z., Xie, Z. & Pan, S. Flexible artificial nociceptor using a biopolymer-based forming-free memristor. *Nanoscale* **11**, 6591–6601 (2019).
8. Yoon, J. H. *et al.* An artificial nociceptor based on a diffusive memristor. *Nat Commun* **9**, 417 (2018).
9. Kim, Y. *et al.* Nociceptive memristor. *Adv. Mater.* **30**, 1704320 (2018).
10. Zhang, X. *et al.* An artificial spiking afferent nerve based on Mott memristors for neurorobotics. *Nat Commun* **11**, 51 (2020).
11. Stachowsky, M., Hummel, T., Moussa, M. & Abdullah, H. A. A slip detection and correction strategy for precision robot grasping. *IEEE/ASME Transactions on Mechatronics* **21**, 2214–2226 (2016).
12. Romeo, R. A. & Zollo, L. Methods and sensors for slip detection in robotics: a survey. *IEEE Access* **8**, 73027–73050 (2020).
13. Lee, Y. *et al.* A low-power stretchable neuromorphic nerve with proprioceptive feedback. *Nat. Biomed. Eng* **7**, 511–519 (2023).
14. Lee, W. W. *et al.* A neuro-inspired artificial peripheral nervous system for scalable electronic skins. *Science Robotics* **4**, eaax2198 (2019).
15. Zhang, Z. *et al.* All-in-one two-dimensional retinomorphic hardware device for motion detection and recognition. *Nat. Nanotechnol.* **17**, 27–32 (2022).
16. Zhou, Y. *et al.* Computational event-driven vision sensors for in-sensor spiking neural networks. *Nat Electron* **6**, 870–878 (2023).
17. Jayachandran, D. *et al.* A low-power biomimetic collision detector based on an in-memory molybdenum disulfide photodetector. *Nat Electron* **3**, 646–655 (2020).
18. Yuan, R. *et al.* A calibratable sensory neuron based on epitaxial VO₂ for spike-based neuromorphic multisensory system. *Nat Commun* **13**, 3973 (2022).
19. Huang, P.-Y. *et al.* Neuro-inspired optical sensor array for high-accuracy static image recognition and dynamic trace extraction. *Nat Commun* **14**, 6736 (2023).
20. Li, S. *et al.* Bioinspired robot skin with mechanically gated electron channels for sliding tactile perception. *Science Advances* **8**, eade0720 (2022).

REVIEWER COMMENTS

Reviewer #1 (Remarks to the Author):

The paper was significantly revised to follow the reviewers' instructions. I am satisfied with the modifications. The only concern is the new title. I would add tactile and visual to make it less generic.

Reviewer #2 (Remarks to the Author):

There is significant improvement of the information and quality of the manuscript since the previous version. There remain a number of (smaller) issues to address in my opinion, which I would like to raise.

Please proof read the paper with the help of a proficient or native English language speaker. It will dramatically improve the coherence of the text.

There is a tendency of the authors to stray into the hand-wavy land of overgeneralizable statements. Compared to the previous version this has been significantly limited (more in the discussion section this time). I want to encourage the authors to eliminate this type of statements. Generalisation (based on deduction) is ok, but overgeneralisation is not ok for scientific work.

One example in this version is the direct comparison of the plastic resistance of memristors with intelligence. It is no accident that in Neuromorphic engineering researchers have been using memristors to enable a form of (synaptic) plasticity and synaptic delays based on this plastic resistive property. That feature alone however is far from intelligence in itself much like an individual synapse is not the basis of intelligence in living organisms. Therefore I propose that the authors revisit the text to tone-down such claims and only focus on the facts being that their novel indeed use of memristive device characteristics, leverages functions that in nature lead to perception abilities and support intelligent behavior through fast *non-linear* adaptation (and this seems to be the main advantage over LTI systems and linear control).

Another example is that perceptual capabilities (e.g. improving dexterity in the case of the robot hand use-case) is not perception, but rather one of the requirements for enabling and improving perception. So overall much as the work presented deserves a merit, it is by far not the solution to intelligence or even perception in its-self but rather an stepping stone towards such a feat. So again the suggestion here is to go through the manuscript and tone down the narrative to what the actual contribution is.

What qualifies as intelligent machines for the authors and what is the current generation of intelligent machines? .. Ideally I would drop the use of the term "next gen intelligent machines", again the work is about adaptive (non-linear) control for perceptive functions based on the use of memristors. Not about intelligent machines, or even AI.

Figures S26, S7.a, S13, should appear and explained in the main text

Some missing inputs that I would like to see in the manuscript are

a) Background noise elimination in autonomous driving use-case. The authors claim of real driving conditions used in the experiments. While obviously the photos in the manuscript are from a driving setup, how fast was the car driving? It seems to me that in most of the recording materials presented, the background is mostly still, when compared to the actors under detection, otherwise I would expect a lot of background clutter noise after the delta filter operation. If that is the case the claim of "real-world driving conditions" should be left out, or a bit more explanation about the filtering is needed to understand how a moving background scene has been eliminated. While it does not affect the importance of the contribution, it does affect the understanding of the maturity of the proposed method and work, and actual applicability in real perception case.

b) Discussion about the response times of the solution in connection to the real-world requirements of these two scenarios

c) Discussion of what would be the required steps (research or engineering challenges to be addressed) to get to a mature solution based in memristors for enabling real-use in perception tasks.

d) Discussion/explanation particularly for the vision use-case of what makes the memristor use (according to the proposed method) appealing for perception as opposed to actuation? Particularly given the potential of dynamic vision cameras for perception/detection tasks. More promising, less promising, pros/cons, etc.

Some specific examples of improving further the language and narrative follow, but I emphasize that they are cherry picks, the authors need to make sure that the flow is consistent in the end instead of only fixing individual sentences pointed out below.

Abstract

> "To address this need, we present a memristor-based differential neuromorphic computing, a perceptual signal processing and online adaptation method that leverages the nonlinear state-space ..."

"To address this need, we present a memristor-based differential neuromorphic computing, perceptual signal processing and online adaptation method that is leveraged by the nonlinear state-space ..."

> "validated in two different application scenarios: object grasping and autonomous driving"

"validated in two different application scenarios: robotic object grasping and object detection and reaction in autonomous driving"

> "through fast adaptation (in ~ 1 ms) of the unknown object features"

"through fast adaptation (in ~ 1 ms) based on the tactile object features"

> "the potential for adapting to diverse sensing modalities"

"the potential for real-time adaptation with inputs from diverse sensing modalities"

> "smart high-level decisions in the real world."

"adaptive high-level reaction in the real world."

Main

> "home and businesses that offer considerable 'variations'"

"home and businesses that entail considerable 'variability'"

> "recent advancements in life sciences have revealed that the most crucial mechanism humans employ to understand unstructured environments is sensory information differential processing mechanisms (10–14)."

"recent studies/findings in life sciences have revealed that the most common mechanism humans use to understand unstructured environments is differential processing of sensory information (10–14)."

> "Specifically, differential neuromorphic computing is a data manipulation method involving multi-branch functions that emulate biological sensory processing, where sensory stimuli are selectively perceived by receptors and undergo different processing supported by diverse neurons"

"Specifically, in this article we choose to term "differential neuromorphic computing" any data manipulation method involving multi-branch functions that emulate biological sensory processing, where sensory stimuli are selectively perceived by receptors and undergo different processing (pathways) supported by diverse groups of neurons. The use of the word differential in this context is not to be confused with differential (delta) encoding of sensory data as employed say in dynamic vision sensors."

This disambiguation is important since in the vision use-case the authors actually use both methods!

> "differential neuromorphic computing method entails extracting features from unstructured data and applying an adaptive memristive modulation scheme called differential encoding to enhance the adaptability of intelligent machines"

Why would you call this differential *encoding*? There is no encoding taking place. Can't you leave this term out? (it is already reserved for something different which you paradoxically also use and therefore need to be able to describe unambiguously).

> "As a result, this method experimentally facilitates the processing of unstructured data within two complex environments i.e., grasping of objects and autonomous driving"

"We apply this method to the processing of unstructured data within two complex environments i.e., robot control when grasping of objects and object detection and reaction in autonomous driving"

> "emulate nociceptors to achieve amplification (>720%) for hazardous stimuli and adapting receptors to achieve adaptation (<50%) for mild stimuli, which play a key role in grasping unknown objects."

"emulate nociceptors by achieving amplification (>720%) of hazardous stimuli, and adapting receptors by achieving regulation (<50%) of mild stimuli, which play a key role in grasping unknown objects."

> "In the latter, we differentiate visual information by a 40×25 memristor matrix via SPICE simulations and reaches a commendable accuracy of 94% in extracting critical information within 10 autonomous driving scenes, after comparison with human labelled results"

"In the latter scenario, we process differentially encoded visual motion information with a 40×25 memristor matrix and achieve a commendable accuracy of 94% in extracting critical information within each of 10 autonomous driving scenes, after comparison with human labelled results"

This is a good case of clash of terminology(!) and the reason why overloading of existing terminology should be avoided.

> "proposed method has the capability to handle a different number of memristors."

"proposed method is general enough to work with different types or number of memristors."

Results

Missing introductory text before discussing the use-cases. Pls summarise the common underlying use of the of memristor in the two scenarios, since after all that is the meat of the paper. Maybe something along the lines that in both scenarios the multistate characteristics of the memristor enable the selection of different "processing pathways" though distinct modulation functions to process stimuli from the sensor.

Tactile use-case results

> "exhibit sharp or slippery attributes"

"exhibit sharp or slippery surfaces"

> "After the magnitude of a force is measured, an associated modulation scheme is selected and then applied to the memristor. The resistance of the memristor is then changed and used as an indicator of the amplification factor."

In Fig 2.a it seems that the modulation scheme selection is taking into account not only the force amplitude but also the current state of the memristor modulation. Which one is true ? Adapt this sentence or the figure accordingly (I guess the latter since it is confirmed in the text later on?)

Discussion

> "In this study, we propose a memristive perception method harnessing"

"In this study, we propose a memristor-assisted perception method exploiting"

> "surpasses existing intelligent machine technologies in versatility and adaptability."

What is meant by "intelligent machines" ? Confusing dexterity with intelligence and adaptive control with learned systems is a bit of a hand-wavy overstatement that should be left out.

> "Through mimicking the fundamental perception mechanisms, such a memristive method surpasses current neuromorphic technologies in both adaptability and generalizability"

"Through mimicking biological perception mechanisms, the proposed method of use of a memristive device for neuromorphic processing is more general and versatile than currently known uses of memristors in literature."

> "In addition, the synapse-like characteristics of memristors significantly enhance computational efficiency in achieving human-like perception capabilities, which is evidenced by the fact that our hardware tactile system shows a 44% improvement over the equivalent simulation system (Figure S30)"

This is a very bloated statement with inaccurate statements. First of all S30 is trying to measure latency and not computational efficiency. Even so it is not clear to me what it shows/compares and what it proves. A hardware implementation is inherently more performant than a simulation. Also human-like perception capabilities imply a lot more than just reaction time (if that is what the authors have in mind). Finally, I do not see in this statement a comparison emerging between the proposed method and conventional technology.

Methods

The majority of the sections here, rather than describing the processing steps (methods) describe the experimental setup (materials) only. I would prefer if the authors swap some of this material with the material and figures in the SI document. For example the selection and stimulation process of the memristor would be more interesting and useful for the understanding than the model of devices in the measurements (which should be either kept very short or moved to the SI doc).

Similarly material from S.7.2, S.8.2, S.11, S.13 that explain the processing steps should be imported here.

> M is the eigenvalue used for the memristor weight update and associated with its state, V is the piecewise memristive encoding scheme responsible for generating current modulation signals $v(t)$

It remains ambiguous to me both how V and M are used or what they represent. Is V a simple condition function? If M is a scalar eigenval, what does $M(x(t))$ represent as a function? Can you exemplify it in the manuscript?

Reviewer #2 (Remarks on code availability):

I did not execute the code in the repository, however i examined the datasets photos.

The datasets and code in the repo concern only the vision use-case, not the robot hand control use-case. In the manuscript however, the authors claim that the code in the repo can be used to reproduce the results for both use-cases.

For the vision use-case there seems also to be too little scene action in the dataset by contrast to what the authors defend in the manuscript. I have a concern about how the research in the manuscript may have been produced based only on these datasets (although some parts would have).

Reviewer #3 (Remarks to the Author):

I thank the authors for addressing all my comments in detail and with scientific rigor. In particular, I thank the authors for narrowing down the claims and focusing on the novel perception aspect. The newly provided abstract is clear, on-point, and well-aligned with the experimental hardware results (for tactile systems) and for the visual experiment based on simulation results.

The rewritten text in the new abstract, part of the introduction, and many other sections now position the article in the current state-of-the-art landscape.

Table R1 and Figures R14 and R13 highlight the novelty and impact.

Finally, the code in the GitHub repository enables the reproduction of the current results, allowing researchers to expand further based on the current work.

Reviewer #3 (Remarks on code availability):

The code contains a readme, explanations, and comments. I was able to download the dataset. Finally, the code allows researchers to reproduce the results in this paper.

Response Letter to Reviewers' Comments

We appreciate the valuable time and insightful comments by all reviewers. We have addressed all of the reviewer's comments, as outlined in this point-to-point reply. We are grateful to all reviewers for their comments and suggestions which allowed us to improve the quality of our manuscript, resulting in a paper with more rigorous content and clearer in presentation.

Reviewer #1:

The paper was significantly revised to follow the reviewers' instructions. I am satisfied with the modifications. The only concern is the new title. I would add tactile and visual to make it less generic.

Response:

The authors are grateful to this reviewer for the valuable time spent in reviewing our manuscript. We are glad to see that previous revisions encountered the satisfaction of this reviewer and appreciate their advice on the title. In this paper we demonstrated the applicability of the proposed method to multiple sensor modalities, including tactile, visual, temperature, and humidity. We believe that the title aligns well to the work reported in the manuscript, which applied to multiple use-case applications falling under the definition of “unstructured environment”, hence we prefer to maintain the title unchanged.

Reviewer #2

There is significant improvement of the information and quality of the manuscript since the previous version. There remain a number of (smaller) issues to address in my opinion, which I would like to raise.

Response:

We greatly appreciate this feedback on our manuscript and are pleased to see that, according to this reviewer, there has been significant improvement in the information and quality of the manuscript since the previous version. We have carefully considered each of the points raised and have made the corresponding revisions. Below is a summary of the key changes and clarifications we have implemented in response to this comment:

1. Adjustment of Terminology Related to Intelligence and Perception: Based on this comment about *'So overall much as the work presented deserves a merit, it is by far not the solution to intelligence or even perception in its-self but rather an stepping stone towards such a feat'*, we have shifted our focus explicitly to **functions that contribute to perception abilities**, ensuring a precise and accurate description of the capabilities of our technology.

2. Terminology Adjustment from 'Intelligent Machines' to 'Robotics': In response to the suggestion *'Ideally I would drop the use of the term "next gen intelligent machines", again the work is about adaptive (non-linear) control for perceptive functions based on the use of memristors'*, we have replaced the instances of **'intelligent machines'** with **'robotics'** and referred to adaptive control systems of grasping and autonomous driving for validation.

3. Refinement of Terminology Related to Differential Encoding: to answer the question raised *'Why would you call this differential *encoding*? There is no encoding taking place. Can't you leave this term out? (it is already reserved for something different which you paradoxically also use and therefore need to be able to describe unambiguously).'*, we have **removed** 'differential encoding' and addressed the related **terminology clashes**.

4. Terminology Adjustment to 'Memristor-Assisted Perception Method': In line with the reviewer's recommendation *'In this study, we propose a memristor-assisted perception method exploiting'*, we have **replaced 'memristive perception' with 'memristor-assisted perception method'**. This adjustment more accurately describes the role of memristors in enhancing the perception capabilities of our system without overstating their function.

5. Limitations of Our Methods in Real-world Scenarios: to address the reviewer's comment *'I want to encourage the authors to eliminate this type of statements. Generalisation (based on deduction) is ok, but overgeneralisation is not ok for scientific work.'*, we have added discussions on the **current limitations** of our tactile and visual methods and **narrowed down** related claims.

In the following, we provide a more detailed point-wise reply to the detailed comments by this reviewer:

Comment #1.1

Please proof read the paper with the help of a proficient or native English language speaker. It will dramatically improve the coherence of the text.

Response:

Thanks for the advice. We have now proofread the paper with the help of a native English language speaker and acknowledged their contribution in the relevant section.

Comment #1.2

There is a tendency of the authors to stray into the hand-wavy land of overgeneralizable statements. Compared to the previous version this has been significantly limited (more in the discussion section this time). I want to encourage the authors to eliminate this type of statements. Generalisation (based on deduction) is ok, but overgeneralisation is not ok for scientific work.

Response:

We are grateful to this reviewer for advising on clarifying the text to remove any possible risk of overgeneralization. We have revised the manuscript and rephrased the corresponding sections to remove overgeneralization.

One example in this version is the direct comparison of the plastic resistance of memristors with intelligence. It is no accident that in Neuromorphic engineering researchers have been using memristors to enable a form of (synaptic) plasticity and synaptic delays based on this plastic resistive property. That feature alone however is far from intelligence in itself much like an individual synapse is not the basis of intelligence in living organisms. Therefore I propose that the authors revisit the text to tone-down such claims and only focus on the facts being that their novel indeed use of memristive device characteristics, leverages functions that in nature lead to perception abilities and support intelligent behavior through fast *non-linear* adaptation (and this seems to be the main advantage over LTI systems and linear control).

Response:

We appreciate this comment and suggested revision. We have addressed this feedback in the revised manuscript by clarifying the key characteristics of memristor-based non-linear adaptation as a fundamental mechanism leveraging functions that in nature lead to perception abilities and support intelligent behaviors, as follows:

(Main text) On page 9, lines 372-374: ~~In conclusion, this method represents a significant advancement toward achieving fully autonomous and intelligent robot systems capable of effective perception in real-world scenarios.~~ In conclusion, this method marks a significant advancement in harnessing inherent characteristics of memristors leveraging functions that in

nature lead to perception abilities and support intelligent behaviors through rapid non-linear adaptation features.

Another example is that perceptual capabilities (e.g. improving dexterity in the case of the robot hand use-case) is not perception, but rather one of the requirements for enabling and improving perception. So overall much as the work presented deserves a merit, it is by far not the solution to intelligence or even perception in its-self but rather an stepping stone towards such a feat. So again the suggestion here is to go through the manuscript and tone down the narrative to what the actual contribution is.

Response:

We highly appreciate this comment and suggestion to improve the narrative focusing on the actual contribution in the field. We have addressed this feedback in the revised manuscript focusing on the non-linear adaptation feature supporting the intelligent machine behavior. Examples of revisions made to address this feedback are as follows:

(Main text) On page 3, lines 105-108: To address this issue, the proposed memristor-based differential neuromorphic computing method **has been implemented** into the sensing and control system of a robotic hand to **highlight its non-linear adaptation feature as required to achieve** intricate and nuanced tactile perception, **illustrated** in Figure 2a.

(Main text) On page 7, lines 282-283: In contrast, the criterion of **our memristor-assisted visual perception method** is based on objective motion situation,...

(Main text) On page 7, lines 291-295: ~~Compared to traditional object detection methods in autonomous driving, the developed differential neurocomputing framework does not rely on prior trained dataset, illustrating its great power in processing unstructured information and hence prompt the implementation of real time decision making for autonomous driving.~~ Compared with dynamic vision camera, our memristor-assisted approach faces the inherent limitations of frame-based image processing, such as a reduced performance under low-light conditions and a narrower dynamic range, while offering the ability to directly generate afterimages that contain crucial temporal information, thanks to the local processing implemented via the memristor-based architecture (Supplementary Discussion 14).

Comment #1.3

What qualifies as intelligent machines for the authors and what is the current generation of intelligent machines? .. Ideally I would drop the use of the term "next gen intelligent machines", again the work is about adaptive (non-linear) control for perceptive functions based on the use of memristors. Not about intelligent machines, or even AI.

Response:

We acknowledge this insightful comment which helps us to clarify the main focus and contribution of our work. We have addressed this comment by replacing the term “intelligent machines” and opted for the use of terms like “control systems in robotics or autonomous driving”, or more specifically “robotics”, and stressed the use of “adaptive (non-linear) control of perceptive functions based on the use of memristors.”

Comment #1.4

Figures S26, S7.a, S13, should appear and explained in the main text.

Response:

Thanks for the valuable suggestion. We have incorporated and explained the figures in the main text of the revised manuscript, as follows:

Figure 1 (Revised). The proposed differential neuromorphic computing method. (a) Comparison of our proposed differential processing method with biological sensory processing methods and current neuromorphic processing methods. **(b)** The specific implementation of the memristor-based differential processing method. **(c)** The schematic of current methods. **(d)** The

schematic of the differential neuromorphic computing method, distinguishing our approach from existing techniques.

Figure 2 (Revised). Utilizing differential processing method to process tactile information and mimic multiple receptors. (a) Schematic diagram of the comprehensive pipeline designed to process tactile sensory stimuli. This approach initially captures immediate tactile interactions through the resistance changes in a piezoresistive film, employing Analog-to-Digital Converters (ADC) and Digital-to-Analog Converters (DAC) and the required information

exchange interface. Concurrently, it archives historical sensory data from the memristor state, leveraging the memory properties of a memristor. Subsequently, sensory features are extracted from both the current state of the piezoresistive film and the memristor to select a memristive adaptive modulation scheme, driving the nociception, recovery, or adaptation process, based on the memristor conductance change. The entire sensory data acquisition, feature extraction, and modulation scheme selection process are managed through the FPGA control platform. Note that the memristor status is detected and modulated through the encoder and memristor-based process modules within the hardware setup. Additionally, a communication module is established for transferring information to external PC devices alongside a specifically designed interface for manipulator control. (b) The device structure of the self-directed memristor. (c) The U-I test of memristors. In this test, a 500mV peak-to-peak sine wave is applied to the memristor and the 10kΩ fixed-value resistor. (d) Electrical characterization of memristors using positive pulses. (e) Electrical characterization of memristors using negative pulses. (f) Amplification processing of hazardous stimuli. (g) Corresponding biological processing functions. (h) Adaptation processing of mild stimuli. (i) Sensitization processing achieved through further differentiation. (j) Quantitative evaluation of the sensitization function. (k) Realization of adaptation at different speeds. (l) The adaptive rate.

Figure 6 (Added). Method used to stimulate memristors in tactile and visual experiments. (a) Control schemes in tactile differential processing. The system initiates by entering the initialization phase, where it performs an initial power-on reset on the device. Afterward, the system proceeds to the normal working cycle. Within each working cycle, the system first employs a DAC and an ADC to detect the resistance value of the piezoresistive film. It then waits for the system analog switch to activate when read is completed. Subsequently, it detects the resistance value of the memristor through the memristor read and write control circuit. Once the status of both readings are complete, the DAC circuits generate the corresponding modulation voltage. This voltage is then used to modulate the resistance value of the memristor, achieving the differential processing of pressure information. (b) Circuit design of the visual differential process system. In this simulation, the external visual stimuli captured by the CMOS in-vehicle camera are utilized as the input signal for the system in the form of analog voltage signals. The visual information between two frames undergoes linear changes at a fixed time interval. Subsequently, the changes in visual information are extracted using filters. The extracted information is then translated into different modulation voltages through the analog

operation circuit. Finally, the modulation voltage is applied to the memristor using the read and write control circuit. (c) Observed visual stimuli, intensity change extracted by the filter circuits and different modulation voltages applied to the memristor.

The added explanations are detailed below.

(Main text) On page 2, lines 69-71: However, existing methods focus on keeping a memristor to a fixed receptor (Figure 1a, 1c and Supplementary Discussion 11), e.g., the memristor-based pressure nociceptor (27–30) only processes pressure inputs above a pre-set threshold.

(Main text) On page 2, lines 79-83: Drawing inspiration from the sensory information processing model, our proposed differential neuromorphic computing method utilizes memristors' intrinsic multistate properties. It extracts features from unstructured data and modulates the memristor state (Figure 1d), enhancing the adaptability of robotic systems for operation in unstructured environments (Supplementary Discussion 4 to 6).

(Main text) On page 3, lines 125-126: The corresponding memristor conductance change (adaptation, recovery and nociception) can be found in Figure 2a.

(Main text) On page 10, lines 436-438: Following this, a digital-to-analog converter applies the modulation voltage to the memristor. Finally, the system uploads the data of the memristor state and pressure information. The detailed descriptions and schematics are provided in Figures 6a and S4.

(Main text) On page 11-12, lines 481-483: The visual process system consists of four main components: filters, analog computing circuit, control switch, and memristor-based computing circuit. Further details of this circuitry are provided in Figure 6b and 6c.

Comment #2.1

Some missing inputs that I would like to see in the manuscript are

a) Background noise elimination in autonomous driving use-case. The authors claim of real driving conditions used in the experiments. While obviously the photos in the manuscript are from a driving setup, how fast was the car driving? It seems to me that in most of the recording materials presented, the background is mostly still, when compared to the actors under detection, otherwise I would expect a lot of background clutter noise after the delta filter operation. If that is the case the claim of "real-world driving conditions" should be left out, or a bit more explanation about the filtering is needed to understand how a moving background scene has been eliminated. While it does not affect the importance of the contribution, it does affect the understanding of the maturity of the proposed method and work, and actual applicability in real perception case.

Response:

We appreciate the reviewer's feedback regarding the reported experiment on the autonomous driving use-case. We amended the corresponding sections in the revised manuscript and supplementary information as follows:

(Main text) On page 11, lines 474-476: In visual information processing, images captured by a CMOS sensor ~~in driving settings real-driving scenarios~~ are converted into analog voltage inputs for the visual differential processing system.

(SI) On page 48, lines 935-937: Instead, we focus on examining the data sources, decision-making features, and application scenarios utilized in our visual processing experiments. This comparison highlights the practicality of our method, illustrating its alignment with ~~driving real-world data and scenarios~~.

Comment #2.2

b) Discussion about the response times of the solution in connection to the real-world requirements of these two scenarios

Response:

We appreciate the comment by this reviewer, as it gives us the opportunity to discuss the response times of our memristor-assisted perception method in real-world scenarios. The text below has been added onto this revised manuscript, to address this comment.

(Main text) On page 8 lines 344-348: ~~In real-world settings, robotic tactile systems are required to elaborate large amounts of tactile data and respond as quickly as possible, taking less than 100 ms, similar to human tactile systems (58, 59). The current state-of-the-art robotics tactile technologies are capable of elaborating sudden changes in force, such as slip detection, at millisecond levels (from 500 μ s to 50 ms) (59-62), and the response time of our tactile system has also reached this detection level.~~

(Main text) On pages 8-9, lines 348-357: For the visual processing, suppose a vehicle travels 40 km per hour in an urban area and wants control effective for every 1 m. In that case, the requirement translates a maximum allowable response time of 90 ms for the entire processing pipeline, which includes sensors, operating systems, middleware, and applications such as object detection, prediction, and vehicle control (63, 64). When incorporating our proposed memristor-assisted method with conventional camera systems, the additional time delay includes the delay from filter circuits (less than 1 ms) and the switching time for the memristor device, which ranges from nanoseconds (ns) to even picoseconds (ps) (65–68). Compared to the required overall response time of the pipeline, these additions are negligible, demonstrating the potential of our method application in real-world driving scenarios (69).

Comment #2.3

c) Discussion of what would be the required steps (research or engineering challenges to be addressed) to get to a mature solution based in memristors for enabling real-use in perception tasks.

Response:

We are grateful to this reviewer as it gives us the opportunity to discuss the engineering and research challenges to be addressed for enabling real-use application of our proposed solution. To address this point, we have added the following text in the revised version of the manuscript:

(Main text) On page 9, lines 358-371: Although our memristor-based perception method meets the response time requirement for described scenarios, our approach faces several challenges that need to be addressed for real-world applications. Apart from the common issues such as variability in device performance and the nonlinear dynamics of memristive responses, our approach needs to overcome the following challenges:

Automatic modulation scheme and control algorithm: Currently, the modulation voltage applied to memristors is preset based on the external sensory feature, and the control algorithm is based on hard threshold comparison. This setting lacks the flexibility required for diverse real-world environments where sensory inputs and required responses can vary significantly. Therefore, it is crucial to develop a more automatic memristive modulation method along with a control algorithm that can dynamically adjust based on varying application scenarios.

Scalable parallel circuit design: As our method potentially involves controlling a large number of memristors, designing scalable parallel circuits that maintain signal synchronization across extensive memristors poses a significant engineering challenge. Effective practical circuit design must ensure the synchronization and speed of signal processing simultaneously.

Comment #2.4

d) Discussion/explanation particularly for the vision use-case of what makes the memristor use (according to the proposed method) appealing for perception as opposed to actuation? Particularly given the potential of dynamic vision cameras for perception/detection tasks. More promising, less promising, pros/cons, etc.

Response:

We appreciate your feedback regarding the need for further explanation. The above discussion has been added.

(SI) On Page 19-20, lines 606-634:

Comparison between the memristor-assisted visual perception approach and dynamic vision system

Artificial visual systems are designed to have a wide dynamic range, high temporal resolution, and efficient scene understanding ability. Visual sensors, as the initial sensory part, play a crucial role in these systems (53–55). Conventional frame-based cameras, like digital cameras, have the capability to capture visual information about target objects faithfully but generate lots of redundant data (54). This redundancy necessitates local preprocessing to make visual data more efficient, a requirement that aligns well with the capabilities of neuromorphic devices (56–58).

In our memristor-assisted visual system, the memristor array processes the light intensity changes directly at the detection point. This local processing highlights crucial information for decision-making and preserves afterimages that maintain historical visual context, filtering the background beside the moving object, making the visual data more efficient. This capability holds the potential to help with decision-making in real-world scenarios involving rapid movements. Furthermore, the system's reaction can be customized by adjusting the memristive modulation rules according to specific sensory input characteristics, meeting the requirements for different application scenarios. For instance, tuning the amplitude of negative pulses related to low-frequency information in the demonstrated visual systems can change the duration for which afterimages are maintained, providing flexibility in response according to the scenario's demands. Afterimages with longer duration are advantageous for tracking and predicting a single moving object, particularly when it is possibly obstructed by an obstacle. Conversely, when tracking and predicting multiple moving objects, longer afterimage duration could negatively impact accuracy due to overlapping, making shorter afterimage duration necessary.

Compared with dynamic vision cameras (59–61), our memristor-assisted approach faces the inherent limitations of frame-based cameras, including reduced performance under low-light conditions and a narrower dynamic range. In contrast, dynamic vision cameras offer advantages in capturing changes in the scene with higher temporal resolution, broader dynamic range, and lower power requirements. However, dynamic vision cameras typically lack the ability to directly generate afterimages that contain crucial temporal information. To obtain this kind of temporal information, dynamic vision cameras require additional computational resources and storage capacity.

(Main text) On page 7, lines 291-294: Compared with dynamic vision camera, our memristor-assisted approach faces the inherent limitations of frame-based image processing, such as a reduced performance under low-light conditions and a narrower dynamic range, while offering the ability to directly generate afterimages that contain crucial temporal information, thanks to the local processing implemented via the memristor-based architecture (Supplementary Discussion 14).

Comment #3

Some specific examples of improving further the language and narrative follow, but I emphasize that they are cherry picks, the authors need to make sure that the flow is consistent in the end instead of only fixing individual sentences pointed out below.

Abstract

> "To address this need, we present a memristor-based differential neuromorphic computing, a perceptual signal processing and online adaptation method that leverages the nonlinear state-space ..."

"To address this need, we present a memristor-based differential neuromorphic computing, perceptual signal processing and online adaptation method that is leveraged by the nonlinear state-space ..."

> "through fast adaptation (in ~1 ms) of the unknown object features"

"through fast adaptation (in ~1 ms) based on the tactile object features"

> "the potential for adapting to diverse sensing modalities"

"the potential for real-time adaptation with inputs from diverse sensing modalities"

> "smart high-level decisions in the real world."

"adaptive high-level reaction in the real world."

Response:

Thank you for your guidance in helping us improve the language and narrative of our manuscript. We have taken your valuable advice and revised the abstract accordingly. We have made these and other corrections to ensure the overall consistency of the Abstract, as follows:

(Main text) On page 1, lines 23-38: Efficient operation of **control systems in robotics or autonomous driving targeting real-world navigation scenarios** requires perception methods that allow them to understand and adapt to unstructured environments with good accuracy, adaptation, and generality, similar to humans. To address this need, we present a memristor-based differential neuromorphic computing, **perceptual signal processing**, and online adaptation method that **is leveraged by** the nonlinear state-space of memristors to provide neuromorphic style adaptation to external sensory stimuli. The **adaptation ability and generality** of this method **are confirmed by validation** in two different application scenarios: object grasping and autonomous driving. In the former, a robot hand experimentally realizes safe and stable grasping through fast adaptation (in ~1 ms) **based on the tactile** object features (e.g., sharp corner and smooth surface) with a single memristor. In the latter, the **essential information for** decision-making information of 10 unstructured environments in autonomous driving (e.g., overtaking cars, pedestrians) is **extracted with an accuracy of 94%** with a 40×25 memristor array. By mimicking the intrinsic nature of human low-level perception mechanisms, the electronic memristive neuromorphic circuit-based method presented here

shows the potential for **real-time adaptation with inputs from** diverse sensing modalities and helping **these control systems** generate **adaptive high-level reactions to unstructured environments** in the real world.

Comment #4.1

Main

> "home and businesses that offer considerable 'variations'"

"home and businesses that entail considerable 'variability'"

> "recent advancements in life sciences have revealed that the most crucial mechanism humans employ to understand unstructured environments is sensory information differential processing mechanisms (10–14)."

"recent studies/findings in life sciences have revealed that the most common mechanism humans use to understand unstructured environments is differential processing of sensory information (10–14)."

Response:

We appreciate these suggestions to help improve the language of our manuscript. We have implemented these changes as follows:

(Main text) On Page 2, lines 41-47: With such capabilities, **robotics** could truly transit from controlled environments such as factories and laboratories into unstructured environments of home and businesses that **entail considerable 'variability'** (Supplementary Discussion 1 and 2). Traditionally, the adeptness of organisms within unstructured environments has been attributed to the integration of diverse types of physical information (7–9). However, **recent studies in life sciences have revealed that the most common mechanism humans use to understand unstructured environments is differential processing of sensory information (10–14).**

Comment #4.2

> "Specifically, differential neuromorphic computing is a data manipulation method involving multi-branch functions that emulate biological sensory processing, where sensory stimuli are selectively perceived by receptors and undergo different processing supported by diverse neurons"

"Specifically, in this article we choose to term "differential neuromorphic computing" any data manipulation method involving multi-branch functions that emulate biological sensory processing, where sensory stimuli are selectively perceived by receptors and undergo different processing (pathways) supported by diverse groups of neurons. The use of the word differential in this context is not to be confused with differential (delta) encoding of sensory data as employed say in dynamic vision sensors."

This disambiguation is important since in the vision use-case the authors actually use both methods!

Response:

Thanks very much for this valuable comment and revision, which helps disambiguate the term 'differential.' We have revised the manuscript as advised:

(Main text) On Page 2, lines 60-65: Specifically, **in this article we choose to term** ‘differential neuromorphic computing’ **any** data manipulation method involving multi-branch functions that emulate biological sensory processing, where sensory stimuli are selectively perceived by receptors and undergo different processing (**pathways**) supported by diverse **groups of** neurons. **The use of the word differential in this context is not to be confused with differential (delta) encoding of sensory data as employed say in dynamic vision sensors.**

Comment #4.3

> "differential neuromorphic computing method entails extracting features from unstructured data and applying an adaptive memristive modulation scheme called differential encoding to enhance the adaptability of intelligent machines"

Why would you call this differential *encoding*? There is no encoding taking place. Can't you leave this term out? (it is already reserved for something different which you paradoxically also use and therefore need to be able to describe unambiguously).

Response:

We thank the reviewer for spotting this potential ambiguity. We have left the term ‘differential encoding’ out and revised the statement as follows:

(Main text) On Page 2, lines 79-83: **Drawing inspiration from the sensory information processing model, our proposed differential neuromorphic computing method utilizes memristors' intrinsic multistate properties. It extracts features from unstructured data and modulates the memristor state, enhancing the adaptability of robotic systems for operation in unstructured environments** (Supplementary Discussion 4 to 6).

Comment #4.4

**> "As a result, this method experimentally facilitates the processing of unstructured data within two complex environments i.e., grasping of objects and autonomous driving
"We apply this method to the processing of unstructured data within two complex environments i.e., robot control when grasping of objects and object detection and reaction in autonomous driving"**

**> "emulate nociceptors to achieve amplification (>720%) for hazardous stimuli and adapting receptors to achieve adaptation (<50%) for mild stimuli, which play a key role in grasping unknown objects."
"emulate nociceptors by achieving amplification (>720%) of hazardous stimuli, and adapting receptors by achieving regulation (<50%) of mild stimuli, which play a key role in grasping unknown objects."**

Response:

Your feedback on enhancing the language has been incredibly helpful. We have made revisions to the manuscript accordingly:

(Main text) On Pages 2-3, lines 85-87: **We apply this method in two complex environments. Firstly, we address robot control for object grasping, and secondly, we focus on object detection and reaction in autonomous driving.**

(Main text) On Page 3, lines 87-90: In the former, we utilized a single Self-Directed Channel (SDC) memristor, a type of chalcogenide-based electrochemical metallization (ECM) device, to **emulate nociceptors by achieving amplification (>720%) of hazardous stimuli, and adapting receptors by achieving regulation (<50%) of mild stimuli, which play a key role in grasping unknown objects.**

Comment #4.5

> "In the latter, we differentiate visual information by a 40×25 memristor matrix via SPICE simulations and reaches a commendable accuracy of 94% in extracting critical information within 10 autonomous driving scenes, after comparison with human labelled results"

"In the latter scenario, we process differentially encoded visual motion information with a 40×25 memristor matrix and achieve a commendable accuracy of 94% in extracting critical information within each of 10 autonomous driving scenes, after comparison with human labelled results"

This is a good case of clash of terminology(!) and the reason why overloading of existing terminology should be avoided.

Response:

We appreciate the insightful advice on terminology. To address this issue, we have revised and refined the terminology in our manuscript, as follows:

(Main text) On Page 3, lines 90-93: In the latter **scenario**, we **process differentially encoded visual motion** information **with** a 40×25 memristor matrix **and achieve** a commendable accuracy of 94% in extracting critical information within **each of** 10 autonomous driving scenes, after comparison with human labelled results.

(Main text) On Page 3, lines 115-118: The piezoresistive layers and the memristor offer short-term and long-term force information, respectively, based on which the attributes of an object are first extracted by a status acquisition block and then utilized to generate **differential the corresponding** modulation schemes for the memristor.

(Main text) On Page 4, lines 148-149: When considering the unstructured information processing in the real world, it often becomes necessary to perform **further differentiation on multiple processing iterations of** the aforementioned functions.

(SI) On Page 11, lines 336-338: This stage resembles the achievement of computational functions through the memristor's intrinsic ability to alter its resistance in response to ~~differential stimulus encoding~~ the electrical signals generated by the features.

Comment #4.6

> **"proposed method has the capability to handle a different number of memristors."**

"proposed method is general enough to work with different types or number of memristors."

Response:

We deeply appreciate this reviewer advice to help improve clarity of the manuscript. We have implemented the following change to address this comment:

(Main text) On Page 3, lines 93-95: These two experiments **demonstrate** that **the proposed method is general enough to work with different types or number of memristors.**

Comment #5.1

Discussion

> **"In this study, we propose a memristive perception method harnessing"**

"In this study, we propose a memristor-assisted perception method exploiting"

Response:

We acknowledge the help by this reviewer to help rephrase the sentence, which has been amended as follows in the revised manuscript:

(Main text) On Page 7, lines 297-299: In this study, we propose **a memristor-assisted perception method that exploits** both the synapse-like characteristics of memristors, and a bio-inspired workflow design **enabling robotics to effectively adapt and operate in unstructured environments.**

Comment #5.2

> **"surpasses existing intelligent machine technologies in versatility and adaptability."**

What is meant by "intelligent machines" ? Confusing dexterity with intelligence and adaptive control with learned systems is a bit of a hand-wavy overstatement that should be left out.

Response:

We agree with this comment, thanks for point this out. The corresponding sentence has been amended in the revised manuscript, for better clarity and avoiding any overstatements:

(Main text) On Page 7, lines 299-301: This innovative approach utilizes key insights derived from biological analogies to **enhance the adaptability and dexterity of neuromorphic computing.**

Comment #5.3

> **"Through mimicking the fundamental perception mechanisms, such a memristive method surpasses current neuromorphic technologies in both adaptability and generalizability"**

"Through mimicking biological perception mechanisms, the proposed method of use of a memristive device for neuromorphic processing is more general and versatile than currently know uses of memristors in literature."

Response:

We acknowledge this careful revision of the text in our manuscript, which is reflected in the revised version of it:

(Main text) On Page8, lines 317-319: Through mimicking **biological** perception mechanisms, **the proposed method of use of a memristive device for neuromorphic processing is more general and versatile than currently known uses of memristors in literature.**

Comment #5.4

> **"In addition, the synapse-like characteristics of memristors significantly enhance computational efficiency in achieving human-like perception capabilities, which is evidenced by the fact that our hardware tactile system shows a 44% improvement over the equivalent simulation system (Figure S30)"**

This is a very bloated statement with inaccurate statements. First of all S30 is trying to measure latency and not computational efficiency. Even so it is not clear to me what it shows/compares and what it proves. A hardware implementation is inherently more performant than a simulation. Also human-like perception capabilities imply a lot more than just reaction time (if that is what the authors have in mind). Finally, I do not see in this statement a comparison emerging between the proposed method and conventional technology.

Response:

We apologize for the possible misunderstanding around computational efficiency of memristor-based methods.

To address this concern, it might be useful to add that this section was originally introduced in response to a comment in the previous round of revisions, namely *'since memristors are useful for computational efficiency, a relative computational cost indication of using the memristor as such versus implementing a model of the memristor in the digital domain to achieve similar functionality, would be very welcome'*.

Following this advice, we have built a digital memristor model to reproduce the same functionality of the corresponding tactile system hardware implementation and included the detailed simulation code in the GitHub repository associated with the manuscript.

In this revision, we compared the computational efficiency between memristors and the conventional von Neuman architecture. We looked at the cost of operation when both systems achieve similar adaptive control operations in a hardware system. To clarify the significance of this analysis, and avoid possible misunderstanding, we have removed the statement that *'the synapse-like characteristics of memristors significantly enhance computational efficiency in achieving human-like perception capabilities, which is evidenced by the fact that our hardware tactile system shows a 44% improvement over the equivalent simulation system (Figure S30)'* and replaced it with a new text as follows:

(Main text) On Page 8, lines 330-342:

To perform similar adaptive control functions in tactile experiments, the von Neumann architecture follows a multi-step process involving several data movements: 1. Input data about the piezoresistive film state is transferred to the system memory via an I/O interface. 2. This sensory data is then moved from the memory to the cache. 3. Subsequently, it is forwarded to

the Arithmetic Logic Unit (ALU) and waits for processing. 4. Historical tactile information is also transferred from the memory to the cache unless it is already present. 5. This historical data is forwarded to the ALU. 6. ALU calculates the current sensory and historical data and returns the updated historical data to the cache. In contrast, our memristor-based approach simplifies this process, reducing it to three primary steps: 1. ADC reads data from the piezoresistive film. 2. ADC reads the current state of the memristor, which represents the historical tactile stimuli. 3. DAC, controlled by FPGA logic, updates the memristor state based on the inputs. This process reduces the costs of operation and enhances data processing efficiency.

Comment #6.1

Methods

The majority of the sections here, rather than describing the processing steps (methods) describe the experimental setup (materials) only. I would prefer if the authors swap some of this material with the material and figures in the SI document. For example the selection and stimulation process of the memristor would be more interesting and useful for the understanding than the model of devices in the measurements (which should be either kept very short or moved to the SI doc).

Similarly material from S.7.2, S.8.2, S.11, S.13 that explain the processing steps should be imported here.

Response:

We appreciate the constructive feedback on how to better organize the Methods section in our manuscript. We have addressed this valuable comment in the revised manuscript and included more detailed descriptions of the processing steps that are crucial for understanding the proposed approach. Specifically, we have transferred relevant material and figures, including Supplementary Sections S7.2, S8.2, S11, and S13, into the main text and reorganized the Methods section as detailed below.

Methods

Material Selection of the Memristor

The differential neuromorphic computing method is universally applicable across memristors, regardless of their underlying switching mechanisms, making it a versatile solution for neuromorphic computing applications. This universal ability is evidenced by the material selection, i.e., a commercially available memristor in tactile sensing experiments and a well-recognized simulation model for visual information processing.

Electrical Measurements

Electrical measurements were conducted with a RIGOL DG4062 function/arbitrary waveform generator and MSO1074 oscilloscope. For the memristor U-I test in Figure 2c, a 10 k Ω resistor was connected in series with the memristor. A 10 Hz, 500 mV sine wave was applied across both components. The voltage across the memristor and resistor was recorded by separate oscilloscope channels to calculate the memristor's current. The oscilloscope was set to normal triggering mode. During the pulse tests shown in Figures 2d and 2e, the setup remained unchanged, and voltage pulses were applied with the oscilloscope, capturing the resulting voltages in single mode.

Control Circuit of the Memristor

The control circuit of the memristor is designed around an operational amplifier, which is pivotal in achieving precise modulation of the memristor state. Utilizing this design, the memristor state information can be observed based on the output voltage of this circuit. Detailed schematics and tests of this circuit are documented in Figure S3 and S6.

Tactile Processing System

In our tactile system, we process external tactile information by treating the memristor as a synapse, and the modulation of the memristor is dynamically controlled through a multi-branch function $V(\cdot)$ dependent on current sensory features $f(t)$ and memristor resistance $M(x)$, facilitated by the FPGA. The tactile response strength is determined by the product of the stimulus input and the memristor conductance (stimulus input divided by the memristor resistance).

For hazardous stimuli, when current pressure surpasses a predefined value, matching the criteria for dangerous features, positive voltage pulses are generated to increase the memristor conductance. This process aligns with the ‘threshold’ function of biological nociceptors, as demonstrated in the following formula:

$$v(t) = v_{noc} \text{ for } r(t) > r_d, t_{sta} < t < t_{end}$$
$$v(t) = 0 \text{ otherwise}$$

Where $v(t)$ represents the voltage stimuli applied to the memristor, and only if the pressure stimulus meets the hazard characteristics r_d is a voltage pulse with amplitude v_{noc} and duration from t_{sta} to t_{end} is generated. For continuous hazardous stimuli, the memristor conductance gradually increases under sustained positive voltage stimuli, achieving ‘no adaptation’ to dangerous stimuli. The process can be represented as follows:

$$\frac{dx}{dt} = f(x, v(t)) > 0$$

where x represents the memristor state. To realize the ‘sensitization’ function, we draw inspiration from biology, considering information across various time scales. When the stimulus matches hazardous features, and the memristor conductance surpasses a set threshold, it suggests long-term exposure to danger. Consequently, the amplitude of the positive voltage applied to the memristor is increased to amplify the response to hazardous stimuli further. Upon stimulus removal, recovery pulses are generated to reset the memristor to its initial resistance, achieving the ‘recovery’ function.

The mild stimuli are processed similarly; the characteristics of external tactile are analyzed, and if the mild criteria are met, negative voltage pulses are generated to reduce the memristor conductance, achieving the ‘adaptation’ function. This method allows the emulation of both rapidly and slowly adapting receptors by modifying the features (amplitude and width) of negative pulses to adjust the ‘adaptation’ speed. Upon removal of stimuli or mismatching the mild criteria, the memristor resets to its initial resistance.

To achieve the above tactile processing, we employ the FPGA platform to collect pressure data, generate modulation schemes, and upload data. In every control cycle, the system initially detects the current pressure information using the pressure sensor. Subsequently, the FPGA generates an appropriate modulation scheme based on the obtained pressure data and the current state of the memristor. Following this, a digital-to-analog converter applies the modulation voltage to the memristor. Finally, the system uploads the data of the memristor state and pressure information. The detailed descriptions and schematics are provided in

Figures S4 and S5. More detailed information about mimicking the biological tactile system can be found in Supplementary Discussion 7.

The detection logic for slip events

Slip events, triggered by external disturbances, are characterized by abrupt decreasing changes in the previously stable contact force. Traditional methods for extracting such features involve recording historical contact forces and analyzing them with current interaction forces (70, 71). For example (70), slip detection can be accomplished by examining the frequency change Δf based on the historical contact force F_h and the current force F_c ; the frequency change can be determined using the following formula:

$$\Delta f = f(F_c) - f(F_h)$$

where f is a function mapping observing force points to the frequency information. In the memristive implementation, we have developed a detection logic only based on the current memristor state and contact force change to identify such slip events. This method leverages the memristor state as a cumulative record of the features of historical contact force F_h in differential neuromorphic computing to enhance detection efficiency. Specifically, when the memristor is in a high-resistance state ($>225 \text{ k}\Omega$) due to the modulation scheme during stable force contact and the piezoresistive film changes drastically, resembling a spike (increases above $350 \text{ k}\Omega$ from a last point below $250 \text{ k}\Omega$), a slip event is inferred to have occurred. This design allows for the direct interpretation of slip events from the single readout of the current memristor state and piezoresistive film state change without substantial data historical storage and analysis, significantly enhancing processing efficiency. Note that these threshold value selections result from careful hand-tuning based on the memristor's characteristics and the object's contact characteristics.

Robot Experiment

In robot experiments, the FPGA platform is utilized to generate the control commands. Upon activation of the control algorithm (detailed in Figure S11), the FPGA sends a command to the DH gripper, triggering the appropriate reflex response timely. These commands are communicated via the Modbus protocol, using the RS485 interface standard. Each command includes a slave number, function code, operation register, operand, and a CRC check code for verification. Once the DH gripper receives a command, it adjusts its gripping force and position to execute the reflex action.

Visual Processing System

To implement large-scale memristor-based visual differential computing, we employ the SPICE simulation platform. The voltage threshold adaptive memristor (VTEAM) model, selected for its versatility, serves as the fundamental computation unit. Detailed information about the electrical characteristics of this model is available in Figure S12. In visual information processing, images captured by a CMOS sensor in driving settings are converted into analog voltage inputs for the visual differential processing system. In the specific implementation, videos captured by the CMOS camera are initially divided into frames and converted into greyscale. Subsequently, these pieces of time-discrete visual information are

transformed into continuous voltage data. These voltage profiles are then processed by the memristor-based visual system through SPICE simulation, mimicking the data processing manner after direct integration with the CMOS sensor. The visual process system consists of four main components: filters, analog computing circuit, control switch, and memristor-based computing circuit. Further details of this circuitry are also provided in Figure S13.

In visual information processing, analog filters first extract changes in light intensity, categorizing them into high and low frequencies. High-frequency light information is essential for real-time decision making, while low-frequency corresponds to slowly moving or stationary objects. To process the visual information differently, the high-frequency and low-frequency information are transformed into positive and negative pulses, respectively. The relationship can be represented as:

$$v(t) = m \times f_{light} + b > 0 \quad f_{light} \subset f_{high}$$

$$v(t) = n \times f_{light} + c < 0 \quad f_{light} \subset f_{low}$$

Where f_{light} is the frequency information of the light intensity, f_{high} and f_{low} represent high-frequency and low-frequency change features, respectively, and the remaining parameters are constant coefficients. When the memristor exhibits a low-resistance state, it has effectively perceived high-frequency stimuli from the external environment. Conversely, when the memristor is in a high-resistance state, it suggests that the changes in light intensity within that area have been slow. More details can be found in Supplementary Discussion 8.

Technical explanation of the demonstrated visual processing methods

Yellow box: This represents an example case of visual information processing, consisting of a region spanning $m \times n$ pixels within the original image. After being compressed, this region transforms into a single pixel point. Subsequently, this compressed point is subject to processing via a memristor, employing a one-to-one approach.

Preprocessing steps: The process involves utilizing filter circuits to extract the change in light intensity ΔL , within the compressed pixel point.

Methods of identification: The light intensity change ΔL within the compressed pixel point serves as the criterion for classification. Should ΔL surpass the predefined threshold L_{th} , the visual information in this point is categorized as fast. Otherwise, it is classified as slow. This process is implemented by a voltage comparison circuit.

Interpretation of states: For a compressed point categorized as ‘fast,’ the modulation voltage for the memristor is formulated as:

$$V_f = E_f(\Delta L), \Delta L > L_{th}$$

Otherwise, for a point considered ‘slow,’ the voltage is expressed as:

$$V_s = E_s(\Delta L), \Delta L \leq L_{th}$$

Speed boundaries: The demarcation of states is reliant on the threshold L_{th} , which establishes the speed boundary for categorizing the movement of an object as either relatively fast or slow.

Comment #6.2

> M is the eigenvalue used for the memristor weight update and associated with its state, V is the piecewise memristive encoding scheme responsible for generating current modulation signals $v(t)$

It remains ambiguous to me both how V and M are used or what they represent. Is V a simple condition function? If M is a scalar eigenval, what does $M(x(t))$ represent as a function? Can you exemplify it in the manuscript?

Response:

Thanks for the opportunity to clarify the roles and functionalities of M and V in our memristor-based approach. The revised version is detailed below.

We thank the reviewer for this opportunity to clarify the roles and functionalities of M and V in our memristor-based approach. We have addressed this comment in the revised version of the manuscript as follows:

In the main text, pages 12-13, lines 518-561:

“Constructing Components of Differential Neuromorphic Computing

Leveraging the bifurcation of memristor states alongside environmental sensory features, differential neuromorphic computing **provides** robotics with fine-grained adaptive sensing capabilities in a way that draws parallels to how perception works in biology. This approach can be conceptually described as a cooperation of sensory, signal encoding, and neuromorphic operation modules (Figure 1b). The first module can be of the desired sensor type according to mimicked biological sensory. It operates by converting a physical stimulus into electrical information and can be described as:

$$p_i(t) = R_i(s_i(t))$$

where $s(t)$ denotes the physical stimulus, R is the response function of the sensor, and $p(t)$ is output the outputted electrical signal for each i -th channel/memristor.

The signal encoding module extracts features from $p(t)$ and then creates the associated memristive encoding schemes to process sensor information in different manners. The created encoding scheme is applied to the memristor in the neuromorphic operation module, and the changed resistive values indicate the properties of the suffered stimuli. Through the three steps above, different features are properly processed by the predesigned differential modulation methods for memristors, yielding a multifeature differentiation-based comprehensive understanding of environmental knowledge. The entire process corresponds to organisms' differential information perception capability in stimuli reception, transduction and processing (58), as expressed below:

$$f_i(t) = F_i(p_i(t))$$
$$v_i(t) = V_i(f_i(t), M_i(x_i(t-1)))$$

$$\frac{dx_i}{dt} = C_i(x_i, v_i)$$

where F is the extracting function, $f(t)$ is the current extracted features calculated by F , $x(t-1)$ is the memristor state after the previous modulation (at the time step $t-1$), M is the eigenvalue calculation function whose input variable is the memristor state $x(t-1)$, therefore $M(x(t-1))$ is the scalar related to the memristor state, which is used to determine the appropriate modulation schemes, V is the piecewise memristive encoding scheme (condition function) responsible for generating current modulation signals $v(t)$, dx/dt is the derivative of the memristor state variable, and C is the state derivative function related to the memristor mechanism, current state and external modulation voltage for each i -th channel/memristor.

Notably, in our tactile experiments, the scalar used to determine to the memristive modulation schemes is the memristor resistance value, thus $M(x(t-1))$ refers to the observed memristor resistance value after the last modulation. V is a condition function based on current stimulus strength $f(t)$ and memristor resistance value $M(x(t-1))$ to generate the modulation voltages $v(t)$, as shown in Supplementary Table 1. In visual experiments, the memristor state is not used to stimulate the memristor, and V is a condition function based on the current frequency of light stimuli f_{light} as below:

$$v(t) = m \times f_{light} + b > 0 \quad f_{light} \subset f_{high}$$

$$v(t) = n \times f_{light} + c < 0 \quad f_{light} \subset f_{low}$$

where f_{high} and f_{low} represent high-frequency and low-frequency change features, respectively, and the remaining parameters are constant coefficients.

As explained, the proposed method provides a human-like information processing pipeline, which extracts key features of undergoing stimuli in real-time, opening the possibility for intelligent machines to operate in unstructured environments efficiently.”

And in Figure 6 (added):

Figure 6 (added). Method used to stimulate memristors in tactile and visual experiments. (a) Control schemes in tactile differential processing. The system initiates by entering the

initialization phase, where it performs an initial power-on reset on the device. Afterward, the system proceeds to the normal working cycle. Within each working cycle, the system first employs a DAC and an ADC to detect the resistance value of the piezoresistive film. It then waits for the system analog switch to activate after detection. Subsequently, it detects the resistance value of the memristor through the memristor read and write control circuit. Once both detections are completed, according to the logic, the DAC circuits generate the corresponding modulation voltage. This voltage is then used to modulate the resistance value of the memristor, achieving the differential processing of pressure information. (b) Circuit design of the visual differential process system. In this simulation, the external visual stimuli captured by the CMOS in-vehicle camera are utilized as the input signal for the system in the form of analog voltage signals. The visual information between two frames undergoes linear changes at a fixed time interval. Subsequently, the changes in visual information are extracted using filters. The extracted information is then translated into different modulation voltages through the analog operation circuit. Finally, the modulation voltage is applied to the memristor using the read and write control circuit. (c) Observed visual stimuli, intensity change extracted by the filter circuits and different modulation voltages applied to the memristor.

Comment #7

Reviewer #2 (Remarks on code availability):

I did not execute the code in the repository, however i examined the datasets photos.

The datasets and code in the repo concern only the vision use-case, not the robot hand control use-case. In the manuscript however, the authors claim that the code in the repo can be used to reproduce the results for both use-cases.

For the vision use-case there seems also to be too little scene action in the dataset by contrast to what the authors defend in the manuscript. I have a concern about how the research in the manuscript may have been produced based only on these datasets (although some parts would have).

Response:

We appreciate the reviewer checking the code provided in the repository.

We are glad to confirm that the repository actually includes the robot hand control code. We have highlighted it and added another example of communication code for use with the robotic platform using RS485. Additionally, we have uploaded more datasets for the vision use-case, which should be helpful for reproducing the results.

Reviewer #3 (Remarks to the Author):

I thank the authors for addressing all my comments in detail and with scientific rigor. In particular, I thank the authors for narrowing down the claims and focusing on the novel perception aspect. The newly provided abstract is clear, on-point, and well-aligned with the experimental hardware results (for tactile systems) and for the visual experiment based on simulation results.

The rewritten text in the new abstract, part of the introduction, and many other sections now position the article in the current state-of-the-art landscape.

Table R1 and Figures R14 and R13 highlight the novelty and impact.

Finally, the code in the GitHub repository enables the reproduction of the current results, allowing researchers to expand further based on the current work.

Response:

We are grateful to this reviewer for the time and effort dedicated in reviewing our manuscript. The insightful comments provided in earlier revisions have been instrumental to enhance the quality of our work.

The code contains a readme, explanations, and comments. I was able to download the dataset. Finally, the code allows researchers to reproduce the results in this paper.

Response:

Thanks again for reviewing our code and dataset.

References (Added)

58. Johansson, R. S. & Westling, G. Roles of glabrous skin receptors and sensorimotor memory in automatic control of precision grip when lifting rougher or more slippery objects. *Exp Brain Res* **56**, (1984).
59. Romeo, R. A. & Zollo, L. Methods and Sensors for Slip Detection in Robotics: A Survey. *IEEE Access* **8**, 73027–73050 (2020).
60. Van Wyk, K. & Falco, J. Calibration and Analysis of Tactile Sensors as Slip Detectors. in *2018 IEEE International Conference on Robotics and Automation (ICRA)* 2744–2751 (IEEE, Brisbane, QLD, 2018). doi:10.1109/ICRA.2018.8461117.
61. Muthusamy, R., Huang, X., Zweiri, Y., Seneviratne, L. & Gan, D. Neuromorphic Event-Based Slip Detection and Suppression in Robotic Grasping and Manipulation. *IEEE Access* **8**, 153364–153384 (2020).
62. James, J. W. & Lepora, N. F. Slip Detection for Grasp Stabilization With a Multifingered Tactile Robot Hand. *IEEE Transactions on Robotics* **37**, 506–519 (2021).
63. Kato, S. *et al.* An Open Approach to Autonomous Vehicles. *IEEE Micro* **35**, 60–68 (2015).
64. Liu, L. *et al.* Computing Systems for Autonomous Driving: State of the Art and Challenges. *IEEE Internet of Things Journal* **8**, 6469–6486 (2021).
65. Zhang, Y. *et al.* Three-dimensional perovskite nanowire array–based ultrafast resistive RAM with ultralong data retention. *Sci. Adv.* **7**, eabg3788 (2021).
66. Pazos, S. *et al.* Solution-processed memristors: performance and reliability. *Nat Rev Mater* 1–16 (2024). doi:10.1038/s41578-024-00661-6
67. Kumar, S., Wang, X., Strachan, J. P., Yang, Y. & Lu, W. D. Dynamical memristors for higher-complexity neuromorphic computing. *Nat Rev Mater* **7**, 575–591 (2022).
68. Goswami, S. *et al.* Charge disproportionate molecular redox for discrete memristive and memcapacitive switching. *Nat. Nanotechnol.* **15**, 380–389 (2020).
69. Lin, S.-C. *et al.* The Architectural Implications of Autonomous Driving: Constraints and Acceleration. in *Proceedings of the Twenty-Third International Conference on Architectural Support for Programming Languages and Operating Systems* 751–766 (ACM, Williamsburg VA USA, 2018). doi:10.1145/3173162.3173191.
70. Stachowsky, M., Hummel, T., Moussa, M., Abdullah, H. A. A Slip Detection and Correction Strategy for Precision Robot Grasping. *IEEE ASME Trans. Mechatron.* **21**, 2214–2226 (2016).
71. Romeo, R. A., Zollo, L. Methods and Sensors for Slip Detection in Robotics: A Survey. *IEEE Access* **8**, 73027–73050 (2020).

List of Supplementary Videos

Supplementary Video SV1 Grasping experiment 1: Sharp object

Supplementary Video SV2 Grasping experiment 2: Slippery object

Supplementary Video SV3 Visual information processing at night

Supplementary Video SV4 Visual information processing in daytime

Supplementary Video SV5 Visual information processing in various driving scenarios

Supplementary Video SV6 Introduction to differential neuromorphic computing